# Munc13-1 restoration mitigates presynaptic pathology in spinal muscular atrophy

**Mehri Moradi** [1] ✉, **Julia Weingart** [2,5], **Chunchu Deng** [1,3,5], **Mahoor Nasouti**[1,4], **Michael Briese** [1], **Sibylle Jablonka** [1], **Markus Sauer** [2] & **Michael Sendtner** [1] ✉

Degeneration of neuromuscular synapses is a key pathological feature of spinal muscular atrophy (SMA), yet cellular mechanisms underlying synapse dysfunction remain elusive. Here, we show that pharmacological stimulation with Roscovitine triggers the assembly of Munc13-1 release sites that relies on its local translation. Our findings show that presynaptic mRNA levels and local synthesis of Munc13-1 are diminished in motoneurons from SMA mice and hiPSC-derived motoneurons from SMA patients. Replacement of the Munc13-1 3'UTR with that of Synaptophysin1 rescues *Munc13-1* mRNA transport in SMA motoneurons and restores the nanoscale architecture of presynaptic Munc13-1 release sites. Restoration of Munc13-1 levels leads to functional synaptic recovery in cultured SMA motoneurons. Furthermore, SMA mice cross-bred with a conditional knock-in mouse expressing modified Munc13-1 with a heterologous 3'UTR display attenuated synapse and neurodegeneration and improved motor function. Identifying Munc13-1 as an SMA modifier underscores the potential of targeting synapses to mitigate neuromuscular dysfunction in SMA.

Spinal muscular atrophy (SMA) is a genetic neuromuscular disorder caused by mutations in the *SMN1* gene[1,2]. The loss of SMN function triggers a cascade of pathological events in SMA, including the degeneration of spinal motoneurons[3,4], axonal loss in the ventral roots[5], denervation of neuromuscular junctions (NMJs)[6,7], and loss of central proprioceptive synapses[8,9]. These abnormalities are accompanied by the presynaptic accumulation of neurofilaments[10], defective endplate maturation[11], and progressive muscle weakness and atrophy[6,12,13]. SMA mouse models show atrophic and smaller neuromuscular synapses, which are associated with impaired synaptic transmission and degeneration[14,15]. Importantly, SMA mice exhibit plasticity defects at NMJs[14,16,17], which significantly contribute to neurotransmission abnormalities. Current therapies for SMA primarily focus on increasing SMN protein levels by targeting the *SMN2* gene[18]. These therapies can effectively prevent disease progression when administered early. However, patients who miss the therapeutic window benefit less from such SMN-repletion therapies. Thus, there is an unmet need to understand the cellular mechanisms driving synaptic deficits in SMA to identify disease modifiers that can mitigate neuromuscular pathology. SMN plays a role in the assembly of small nuclear ribonucleoprotein (snRNP) particles involved in RNA splicing[19,20], including the assembly of U7 snRNP required for *Histone* mRNA processing[21,22] as well as in the assembly of mRNPs[23] involved in mRNA subcellular localization[24–26] and local translation[27]. Additionally, it directly interacts with ribosomes to regulate the translation of specific proteins[28]. Despite these known functions, a direct link between SMN function and the cellular targets that maintain synaptic integrity is only now emerging[29–31].

In presynaptic terminals, neurotransmitter release occurs at active zones (AZs)[32,33]. A tripartite Rab3A/α-RIM/Munc13-1 complex

[1]Institute of Clinical Neurobiology, University Hospital Wuerzburg, Versbacher Str. 5, Wuerzburg, Germany. [2]Department of Biotechnology and Biophysics, Biocenter, Julius-Maximilians-University Wuerzburg, Wuerzburg, Germany. [3]Department of Rehabilitation, Tongji Hospital, Tongji Medical College, Huazhong University of Science and Technology, Jiefang Avenue, Wuhan, China. [4]Department of Neurology and Experimental Neurology, Charité-Universitätsmedizin Berlin, Berlin, Germany. [5]These authors contributed equally: Julia Weingart, Chunchu Deng. ✉e-mail: E_Moradi_M@ukw.de; Sendtner_M@ukw.de

mediates the docking and priming of synaptic vesicles (SVs) onto AZs[34] and activates the SNARE/SM fusion machinery[35,36]. In addition, Munc13-1 regulates synaptic plasticity by modulating vesicle release probability and altering the fusion competence of the readily releasable pool[36–38]. Importantly, at NMJs synaptic transmission strongly depends on Unc13A protein[39–41] and mutations in the *Unc13A* gene impair the synaptic plasticity at NMJs[39,42,43]. Loss of Munc13-1 decreases both spontaneous and evoked synaptic release events in hippocampal neurons[37] and also evoked synaptic release events at NMJs[41,44]. This causes severe paralysis leading to early postnatal death in knockout (KO) mice[37]. In humans, mutations in the *UNC13A* gene cause microcephaly, cortical hyperexcitability, and fatal myasthenia[45].

Recently, the *UNC13A* gene has been identified as a survival modifier in patients with sporadic Amyotrophic lateral sclerosis (ALS) and Frontotemporal dementia[46–48]. Single nucleotide polymorphisms (SNPs) in the *UNC13A* gene and Tdp-43 loss of function promote the inclusion of a cryptic exon in *UNC13A* transcripts. This leads to its missplicing and cause UNC13A dysfunction[49–51].

Given the critical role of Munc13-1 in SV priming and fusion, we asked whether alterations in Munc13-1 synaptic functions contribute to defective neurotransmission and plasticity at NMJs in SMA.

Here, we investigated changes in the nanoscale supramolecular architecture of Munc13-1 clusters at axonal presynaptic membranes in spinal motoneurons by super-resolution microscopy. We demonstrate that locally translated Munc13-1 is involved in stimulus-dependent de novo assembly of Munc13-1 clusters that might represent SV release sites. Furthermore, we provide evidence that the axonal localization of *Munc13-1* mRNA is dependent on SMN, leading to diminished local translation of Munc13-1 in SMA. We generated a Munc13-1 rescue construct with modified 3'UTR to restore Munc13-1 protein and mRNA levels in axons of Smn KO motoneurons. Overexpression of this construct improved neurotransmitter release and enhanced the excitability in cultured Smn KO motoneurons. We generated a ROSA26 Cre/loxP conditional Munc13-1 knock-in rescue mouse model that harbors a *Munc13-1* allele with the modified 3'UTR. Of note, SMA mice crossbred with the knock-in rescue mice displayed diminished synapse and motoneuron degeneration as well as improved motor function, leading to increased lifespan of SMA animals. Thus, our results identify Munc13-1 as a modifier in SMA that can counteract NMJ dysfunction and highlight the role of Munc13-1 local translation for presynaptic function in spinal motoneurons.

## Results

### Axonal translocation and local translation of *Munc13-1* mRNA are diminished in SMA

Previous studies have demonstrated the distinct localization of transcripts for synapse-related proteins in neuronal dendrites[52,53], and axons[25,54,55]. Importantly, studies using RNA sequencing have revealed the localization of *Munc13-1* transcripts in axons of cultured mouse spinal motoneurons[25,54,56] as well as in axons and neuropils of neurons in the central nervous system[57–59]. The Smn protein plays a key role in the axonal transport and local translation of a variety of target mRNAs[24,28]. To explore whether *Munc13-1* mRNAs undergo Smn-dependent axonal translocation in spinal motoneurons, we utilized the severe Taiwanese-SMA mouse model on a C57BL/6 J background (C57BL/6 J.29P2-Smn1Hung<tm1Msd > /J)[60,61]. To assess *Munc13-1* mRNA levels, we isolated spinal motoneurons from E12.5 *Smn⁻/⁻,Hung^tg/+* (referred thereafter to as Smn KO), and *Smn⁺/⁻,Hung^tg/+* (referred thereafter to as control) mouse embryos, and cultured them for 7 days in compartmentalized chambers[27,54]. qRT-PCR analysis of RNA from the axonal compartment revealed an approximately 75% reduction in *Munc13-1* transcript levels in distal axons of Smn KO motoneurons compared to control (Fig. 1a). Next, we employed smFISH to investigate potential abnormalities in the subcellular localization of *Munc13-1* transcripts in cultured motoneurons from Smn KO

and control mice. As illustrated in Fig. 1b–d, we observed reduced levels of *Munc13-1* transcripts in the distal axon and axonal growth cones of Smn KO motoneurons, despite unchanged mRNA levels in the soma (Supplementary Fig. 1a). The specificity of the Munc13-1 FISH probe was validated in control experiments following Munc13-1 knockdown in motoneurons using an shRNA targeting Munc13-1 (Supplementary Fig. 1b–d). Furthermore, immunofluorescence analysis showed that impaired mRNA localization of Munc13-1 was associated with reduced levels of the corresponding protein in axonal growth cones of cultured Smn KO motoneurons (Supplementary Fig. 1e, f), as well as at NMJs of transverse abdominal muscle (TVA) from postnatal day 5 (P5) Smn KO mice (Fig. 1e–g). In contrast, Munc13-1 protein levels were not altered in somata of cultured Smn KO motoneurons (Supplementary Fig. 1g, h). Consistent with this, we observed a 25% reduction in total Munc13-1 protein levels in total lysates from Smn KO cultured motoneurons (Supplementary Fig. 1i, j). This indicates that the presynaptic protein and mRNA localization of *Munc13-1* are altered in SMA.

Next, we investigated whether diminished local translation of Munc13-1 might contribute to its reduced presynaptic levels. To this end, we performed puromycin proximity ligation analysis (Puro-PLA)[62], to detect Munc13-1 translation in axonal growth cones of cultured motoneurons in situ (Fig. 1h). We found decreased Puro-PLA signal in growth cones of Smn KO motoneurons, indicating decreased local protein synthesis in the distal axon (Fig. 1i, j). In contrast, no differences were detected in the Puro-Munc13-1-PLA signal in the soma, suggesting unaffected Munc13-1 protein synthesis in cell bodies of cultured Smn KO motoneurons (Supplementary Fig. 1k, l).

The specificity of the PLA signal was verified in control experiments using wildtype (wt) motoneurons, where only Munc13-1 or puromycin antibodies were used (Supplementary Fig. 1m). Moreover, pretreatment of motoneurons with anisomycin resulted in a near-complete loss of the Puro-Munc13-1-PLA signal (Supplementary Fig. 1n). This confirms that the detected signal reflects active translation. These results indicate that first, Munc13-1 becomes locally translated at axonal growth cones in cultured motoneurons, and second, its local translation is impaired in SMA.

In order to investigate the local translation of Munc13-1 in synapses in vivo, we performed RNA immunoprecipitation (RNA-IP) assays with ribosome pulldown using cortical synaptosome fractions isolated from P5 Smn KO and control littermates. As shown in Fig. 1k, RNA-IP revealed reduced *Munc13-1* mRNA levels bound to ribosomes in immunoprecipitated fractions of Smn KO compared to control. This reduction correlated with decreased *Munc13-1* mRNA levels in synaptic fractions, as shown by the assessment of input fractions (Fig. 1l). Interestingly, this phenotype was not caused by altered levels of ribosomes in synaptic fractions from Smn KO animals (Supplementary Fig. 2a–e). In control experiments, we conducted Western blot assays to examine the quality of the crude synaptosome fractions. In contrast to Histone H3, Synapsin levels were enriched in the Pellet 2 (synaptosome) fraction (Supplementary Fig. 2f). Together, these data show that axonal translocation of *Munc13-1* mRNAs for its subsequent local translation depends on Smn protein.

Collectively, these data suggest that reduced presynaptic levels of Munc13-1 may contribute to the synaptic defects observed in SMA.

### Axonal localization of *Munc13-1* mRNA depends on its 3'UTR and is regulated by Smn

Munc13-1 binds to hnRNP R via its 3'UTR[63], suggesting that impaired axonal localization of Munc13-1 is a consequence of reduced hnRNP R levels in axons of Smn KO motoneurons[64]. To validate that the axonal translocation of *Munc13-1* mRNA specifically depends on the Smn protein, we utilized other transcripts, such as *Synaptophysin1* (Syn-Phy), which are not reduced in axons of Smn-knockdown motoneurons as shown by RNA-seq[25] (Supplementary Table 1), and qRT-PCR

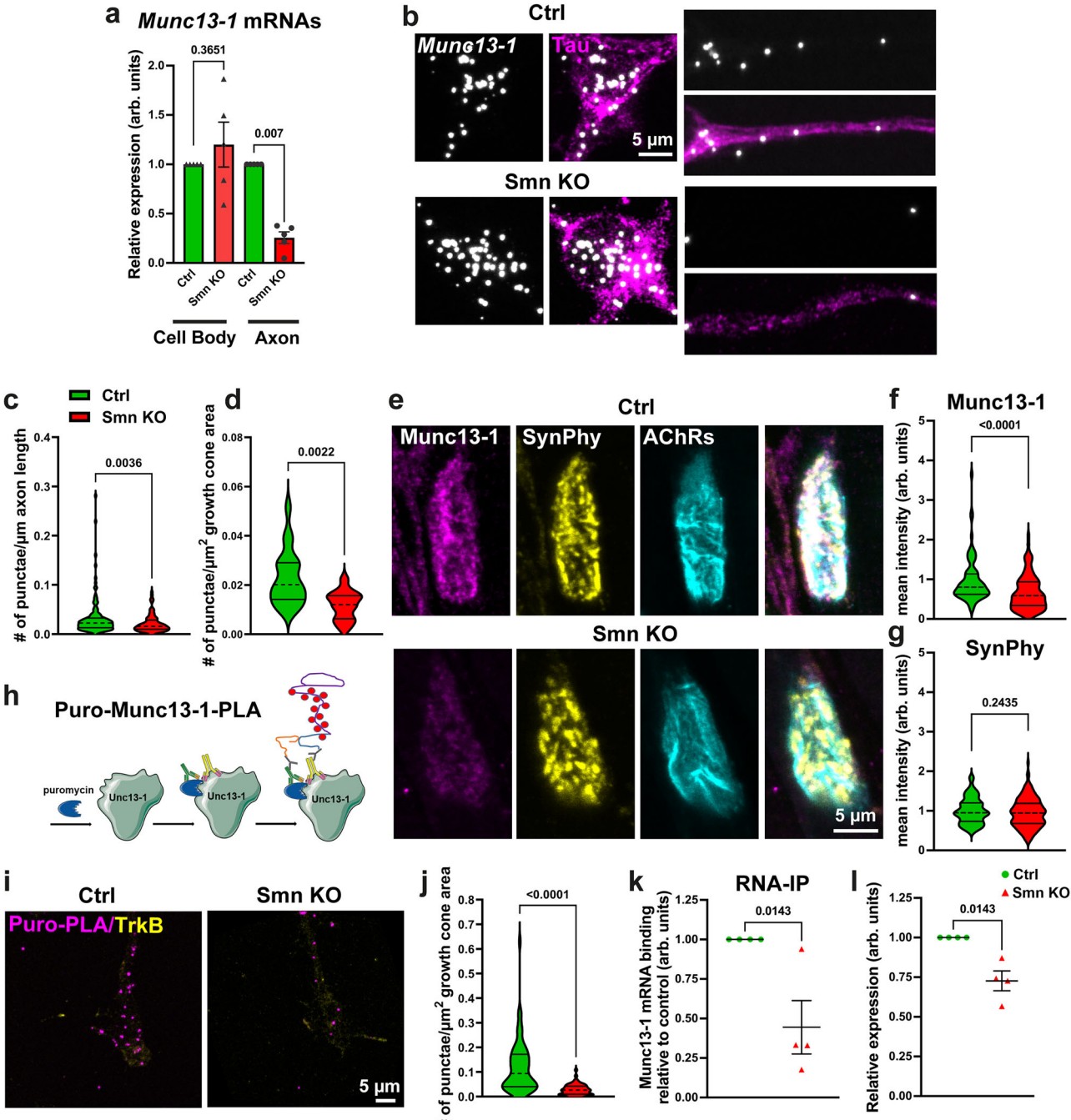

**Fig. 1 | Munc13-1 local translation is dysregulated in axon and presynaptic terminals in SMA.** **a** qRT-PCR reveals a significant reduction in *Munc13-1* mRNA levels in axons of cultured Smn KO motoneurons compared to control (Ctrl) (\*\**P* = 0.007; n = 5 biological replicates). **b** smFISH detects *Munc13-1* transcripts in somata (left panel) and distal axons of cultured motoneurons (right panel) in control and Smn KO motoneurons. **c, d** Graphs show reduced *Munc13-1* transcripts, as determined by quantitative analyses of smFISH punctae in axons (**c**) \*\**P* = 0.0036; *n* = 88-124 cells from *n* = 3 biological replicates) and in axonal growth cones (**d** \*\**P* = 0.0022; *n* = 13-20 cells from *n* = 3 biological replicates) in Smn KO neurons. **e** NMJs of TVA muscles from P5 littermates are stained against Munc13-1. Synaptophysin (SynPhy) antibody labels presynaptic membranes, and αBungarotoxin labels AChRs at postsynaptic membranes. **f, g** Quantification of fluorescent signals reveals decreased Munc13-1, but unaltered SynPhy (*n* = 132-159 NMJs from *n* = 5

mice/genotype) levels in NMJs of Smn KO littermates (\*\*\*\**P* < 0.0001; *n* = 80-101 NMJs from *n* = 3 mice/genotype). **h** Schematic of puromycin-Munc13-1-Proximity Ligation Assay (Puro-Munc13-1-PLA). **i** Locally translated Munc13-1 molecules are detected in axonal growth cones in cultured motoneurons by Puro-PLA. **j** Smn KO motoneurons exhibit reduced Puro-PLA signal for Munc13-1 in axonal growth cones (\*\*\*\**P* < 0.0001; n = 57-65 cells from n = 3 biological replicates). **k** RNA-immunoprecipitation (RNA-IP) shows reduced *Munc13-1* transcripts bound to translating ribosomes in cortical synaptosome fractions from Smn KO mice (\**P* = 0.0143 from *n* = 4 mice/genotype). **l** *Munc13-1* mRNA levels are reduced by 25% in the input of Smn KO compared to control mice (\**P* = 0.0143 from n = 4 mice/genotype). Mann-Whitney U test: one-tailed in (**a**), (**k**), (**l**) and two-tailed in (**c**–**j**). Bars represent mean ± SEM. Source data are provided as a Source Data file.

(Supplementary Fig. 2g-i). Consistent with this, Puro-PLA assays revealed unchanged levels of locally translated VAMP2 in axonal growth cones of Smn KO motoneurons compared to control (Supplementary Fig. 2j,k). These data suggest that the translocation of these transcripts either depends on transport proteins other than Smn or that Smn plays an inhibitory role in their translocation, which is removed upon Smn depletion.

Thus, we investigated whether the exchange of Munc13-1 3'UTR with the 3'UTR from *SynPhy* transcript could rescue defective Munc13-1 axonal localization in SMA. We generated three lentiviral rescue constructs: (i) a construct harboring the coding and the synonymous 3'UTR of Munc13-1 (wtMunc13-1), (ii) a construct harboring the coding region of Munc13-1 fused to SynPhy3'UTR (Rescue), and (iii) a construct lacking the axonal targeting 3'UTR (RescueΔ3'UTR) (Fig. 2a). To validate these constructs, we accomplished qRT-PCR with wt motoneurons grown in compartmentalized chambers. We found elevated *Munc13-1* transcript levels in axons of motoneurons following viral transduction of wtMunc13-1 and Rescue, but not RescueΔ3'UTR (Fig. 2b). This indicates that the 3'UTR of *Munc13-1* mRNA is necessary for its axonal localization and that the 3'UTR of *SynPhy* mRNA may similarly facilitate the axonal targeting of *Munc13-1* mRNA. Moreover, we could detect upregulated Munc13-1 protein levels in total lysates of cultured motoneurons after transduction with these constructs (Fig. 2c). To examine whether the Rescue construct with the modified 3'UTR could translocate *Munc13-1* mRNAs into distal axons in the absence of Smn, we performed smFISH with cultured Smn KO motoneurons. *Munc13-1* mRNAs were significantly increased in the soma of Smn KO motoneurons after viral transduction with both Rescue and RescueΔ3'UTR (Fig. 2d, e). Nevertheless, the localization of *Munc13-1* mRNA was restored in axons and axonal growth cones of Smn KO motoneurons following the expression of Rescue, but not the RescueΔ3'UTR construct (Fig. 2d, f, g). As expected, Munc13-1 protein levels were increased in the somata and axonal growth cones of Smn KO motoneurons transduced with either rescue constructs, as shown by immunostaining (Supplementary Fig. 2l–o).

These data demonstrate that axonal localization of *Munc13-1* mRNA requires Cis-elements within its 3'UTR, which are specifically recognized by Smn-dependent RNA binding proteins for axonal transport.

## Munc13-1 local translation is required for stimulus-dependent formation of its supramolecular clusters

To assess whether Munc13-1 undergoes local translation in response to stimulation, resulting in enhanced presynaptic $Ca^{2+}$ influx, we applied Roscovitine. This compound slows the deactivation of voltage-gated $Ca^{2+}$ channels ($Ca_v2.1/Ca_v2.2$) and prolongs their open state, leading to a transient enhancement of calcium influx[65–67]. The selection of Roscovitine for stimulation was based on previous findings demonstrating its ability to increase calcium influx through remaining $Ca_v2.2$ channels in *Smn* KO motoneurons[65], and its effects in models of Lambert-Eaton Syndrome[68], a disease characterized by autoimmune responses against voltage-gated $Ca^{2+}$ channels at motor nerve terminals. BDNF stimulation was ineffective in this context due to impaired TrkB signaling[25,69], and depolarizing agents such as KCl were avoided to prevent confounding effects associated with dysfunctional $Ca_v2.2$ channel clustering and activity in Smn KO motoneurons[15].

Remarkably, following a 5-minute Roscovitine pulse, a significant increase in Munc13-1 immunoreactivity was detectable in axonal growth cones of stimulated control motoneurons (Fig. 3a, b). This was impeded upon pretreatment of neurons with anisomycin, which blocks the protein synthesis (Supplementary Fig. 3a). In contrast, the increase of Munc13-1 immunoreactivity was not influenced by nocodazole treatment, which disrupts microtubules and thus the axonal transport of proteins being synthesized in the soma (Supplementary Fig. 3a). Similarly, stimulation of wt motoneurons with 40 nM BDNF for 1 min

(Supplementary Fig. 3b, c) or with 90 mM KCl for 5 min (Supplementary Fig. 3d, e) increased Munc13-1 levels in axonal growth cones. An effect that was blocked by treatment with anisomycin (Supplementary Fig. 3c, e), or ω-Conotoxin (CTX)/Tetrodotoxin (TTX) (Supplementary Fig. 3e), but not with nocodazole (Supplementary Fig. 3c).

Similar to the control group, Roscovitine stimulation elicited the local translation of Munc13-1 in axonal growth cones of Smn KO^Rescue motoneurons, but not in either stimulated Smn KO^RescueΔ3'UTR or stimulated Smn KO motoneurons (Fig. 3a, b). Thus, replacing the 3'UTR of Munc13-1 with that of SynPhy restores Munc13-1 local presynaptic synthesis in Smn KO motoneurons.

Next, we utilized super-resolution microscopy to assess the dynamic changes in the nanoscale architecture of Munc13-1 at axonal presynaptic membranes using structured illumination microscopy (Lattice-SIM). We identified ring-like Munc13-1 cluster architectures that tightly overlapped with the synaptic marker Snap25 (Fig. 3c). In cultured control motoneurons, we identified on average two ring-like Munc13-1 clusters in each presynaptic membrane, which displayed a diameter range between 170–1250 nm. Strikingly within 5 min stimulation, these clusters underwent a complete rearrangement to form new structures (Fig. 3d and Supplementary Fig. 3f). We propose that these highly organized clusters might be supramolecular assemblies of Munc13-1 containing release sites for SVs, since treatment with CTX completely blocked the formation of such clusters (Supplementary Fig. 3g).

Notably, Munc13-1 clusters were preserved after inhibition of the translation by anisomycin treatment; however, the formation of new clusters in response to stimulation was impeded (Supplementary Fig. 3h). While anisomycin does not specifically inhibit Munc13-1 synthesis or confirm its local translation, the observed impairment in cluster formation upon pharmacological stimulation supports a general requirement for protein synthesis in this process.

These data indicate that Munc13-1 translation might be required for the de novo formation of its supramolecular clusters. Moreover, motoneurons cultured on a non-muscle-specific laminin isoform (laminin111) did not form Munc13-1 clusters indicating that these structures are specific to presynaptic membranes (Supplementary Fig. 3i). Functional correlation experiments using mCLING-ATTO647N uptake (a dye specially developed for super-resolution microscopy[70]) showed that depolarization of the presynaptic membrane is concomitant with the endocytosis of mCLING at distinct membrane domains that clearly overlap with Munc13-1 clusters, and Synapsin (Supplementary Fig. 3j). This functional assay suggests that these clusters might represent Munc13-1 release sites for SVs.

Strikingly, similar to control, Roscovitine stimulation triggered a complete rearrangement of Munc13-1 clusters in Smn KO^Rescue motoneurons, leading to an increased number of clusters (Fig. 3d and Supplementary Fig. 3f). On the contrary, these dynamic changes were not evident in stimulated Smn KO and Smn KO^RescueΔ3'UTR motoneurons (Fig. 3d and Supplementary Fig. 3f). This clearly shows that Munc13-1 local synthesis contributes to the formation of new clusters in response to enhanced presynaptic $Ca^{2+}$ influx, which is impeded in SMA.

To gain detailed insights into individual Munc13-1 nanoassemblies within supramolecular clusters, we used Expansion Microscopy (ExM)[71]. Cultured wt motoneurons were 8-fold expanded[72], and axon terminals were imaged with lattice-SIM (Fig. 4a and Supplementary Movie 1 and 2). Such Munc13-1 nanoassemblies have been recently described in synapses of cultured hippocampal neurons[73,74]. In unstimulated control growth cones, we identified on average 6.18 individual Munc13-1 nanoassemblies within each cluster (Fig. 4a-c), which exhibited a diameter range between 18 and 113.6 nm (Fig. 4d). The average center-to-center distance between nearest neighboring nanoassemblies was 187.2 ± 5.9 nm (Fig. 4e, f). Intriguingly, upon stimulation, the average number of individual Munc13-1 nanoassemblies within clusters increased from 6.18 to 8.07 (Fig. 4a–c). As illustrated in

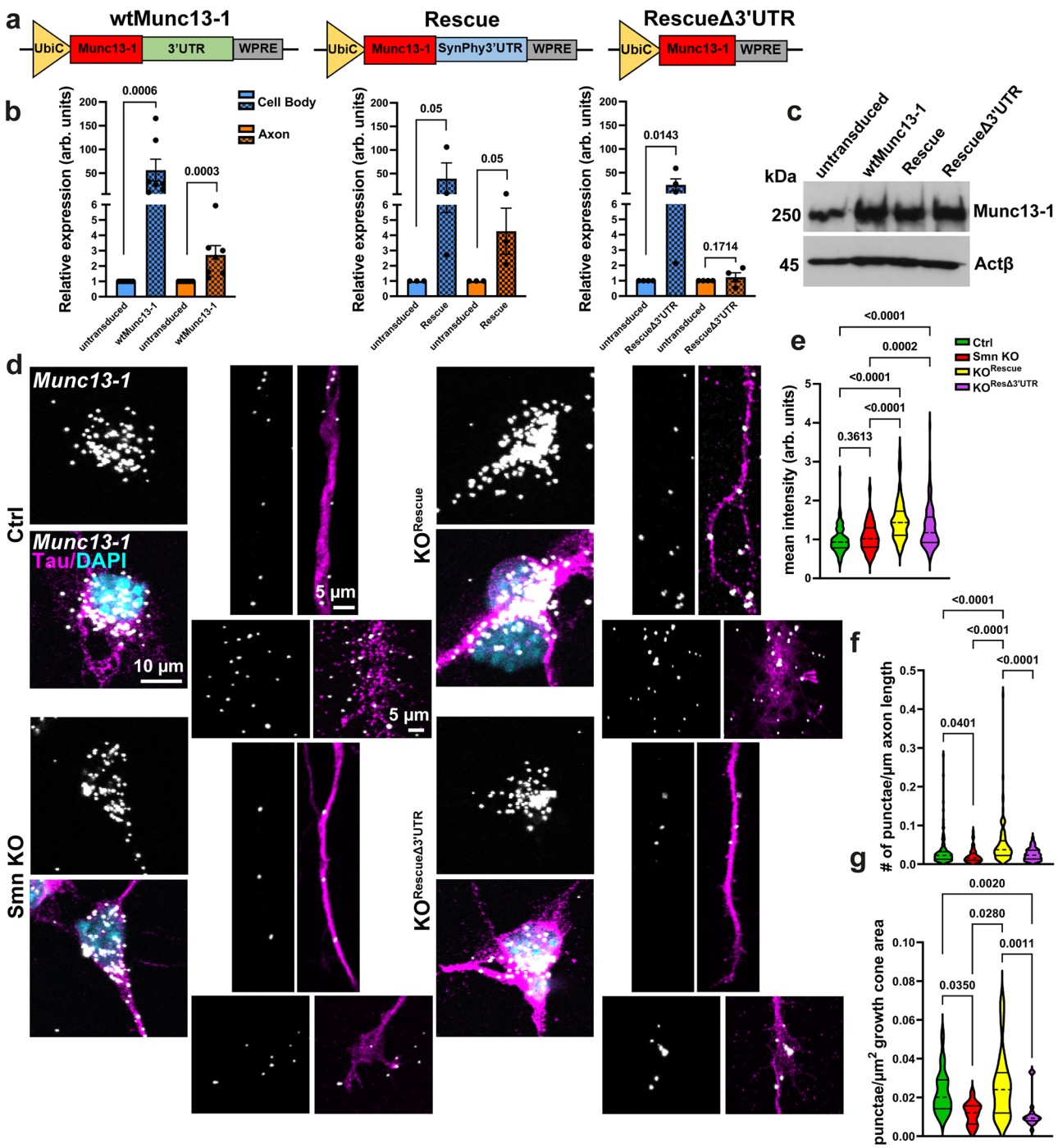

**Fig. 2 | Rescue construct of Munc13-1+SynPhy3'UTR drives axonal localization of _Munc13-1_ mRNAs in cultured Smn KO motoneurons. a** Scheme of three viral rescue constructs expressing Munc13-1 with synonymous 3'UTR (wtMunc13-1), the Synaptophysin (SynPhy) 3'UTR, or no 3'UTR (Δ3'UTR). **b** qRT-PCR reveals that the 3'UTR of Munc13-1 (***P = 0.0003, ***P = 0.0006 from _n_ = 7 biological replicates) and the 3'UTR of SynPhy (*P = 0.05, *P = 0.0286 from _n_ = 3 biological replicates) drive _Munc13-1_ transcripts into distal axons in wt motoneurons, while Munc13-1 construct lacking the 3'UTR does not (*P = 0.0143 from n = 4 biological replicates). **c** Immunoblot shows increased Munc13-1 protein levels in total lysates from motoneurons transduced with viral constructs shown in (**a**, **b**) (representative of _n_ = 3 biological replicates). **d** Representative images of smFISH in the soma, axon,

and axonal growth cones of cultured motoneurons. Nuclei are stained with DAPI. **e** _Munc13-1_ mRNAs are upregulated in the soma of Rescue and RescueΔ3'UTR-transduced Smn KO motoneurons (***P = 0.0002, ****P < 0.0001; _n_ = 134–186 cells from n = 2 biological replicates). **f, g** _Munc13-1_ mRNA levels are elevated in axons (**f**) *P = 0.0401, ****P < 0.0001; _n_ = 88–124 cells from _n_ = 3 biological replicates) and axonal growth cones (**g**) *P = 0.035, *P = 0.028, **P = 0.0011, **P = 0.002; _n_ = 13–27 cells from _n_ = 3 biological replicates) of Rescue, but not RescueΔ3'UTR-transduced Smn KO motoneurons. One-tailed Mann-Whitney U test in **b** and One-way ANOVA with Dunn's post-test in (**e**–**g**). Bars represent mean ± SEM. Source data are provided as a Source Data file.

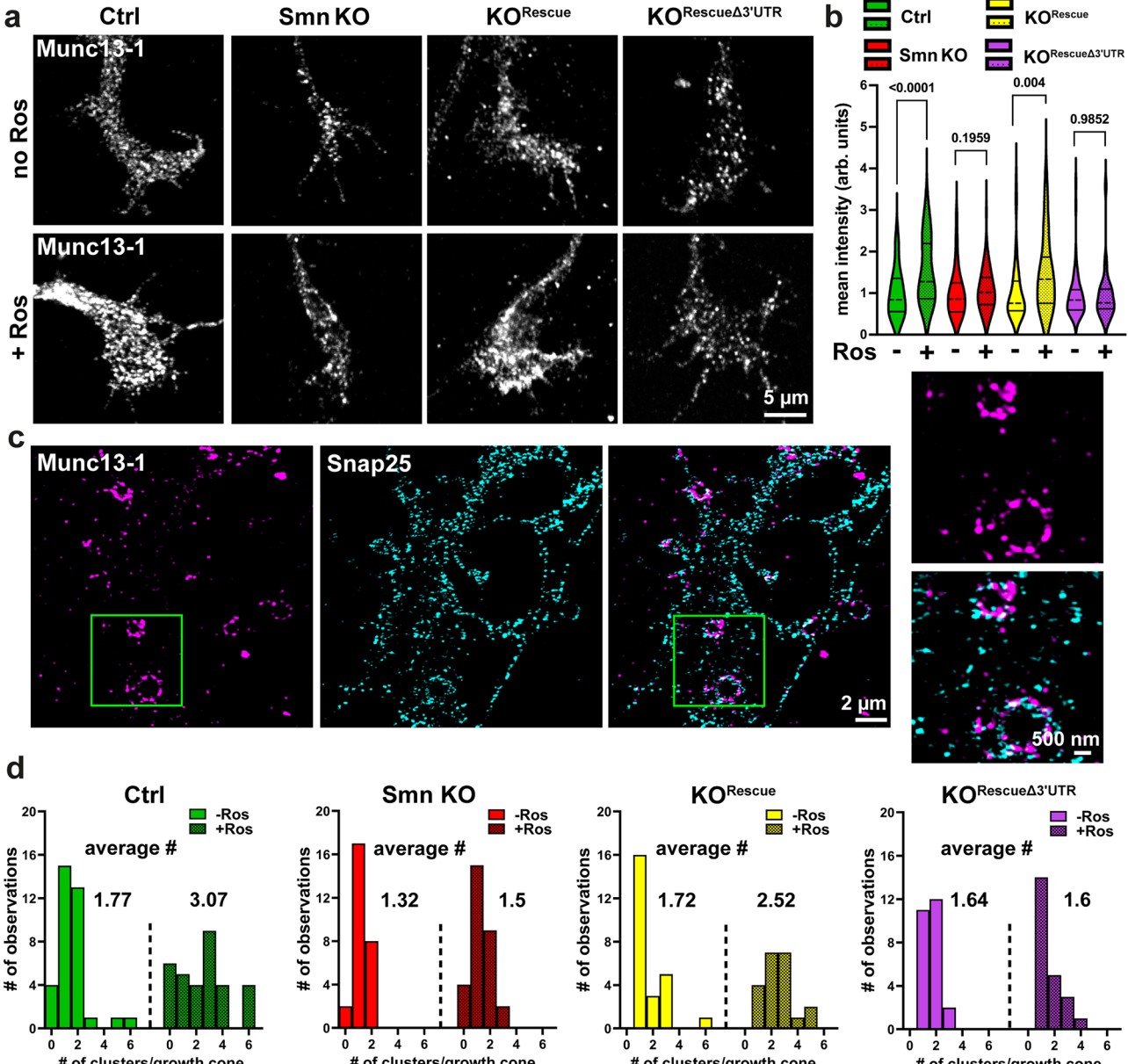

**Fig. 3 | Munc13-1 supramolecular clusters undergo rearrangement in response to stimulation. a** Representative confocal images of Munc13-1 in axonal growth cones of cultured motoneurons before and after Roscovitine (Ros) stimulation. **b** Graph shows increased Munc13-1 levels in stimulated control (Ctrl) (*n* = 4 biological replicates) and Smn KO^Rescue motoneurons (*n* = 3 biological replicates) (***P = 0.0004, ****P < 0.0001; *n* = 60-82 cells), but not in stimulated Smn KO (*n* = 4 biological replicates) and Smn KO^RescueΔ3'UTR motoneurons (*n* = 3 biological replicates). **c** Lattice-SIM shows Munc13-1 supramolecular clusters in presynaptic membranes at axonal growth cones of cultured motoneurons. Inset on the right side: Zoom-in shows colocalization of Munc13-1 with Snap25 in supramolecular clusters. **d** Upon Roscovitine stimulation, the number of Munc13-1 clusters per growth cone increases in control and Smn KO^Rescue motoneurons, but not in stimulated Smn KO and Smn KO^RescueΔ3'UTR motoneurons (from *n* = 3 biological replicates) (for statistical significance, refer to Supplementary Fig. 3f). Two-tailed Mann-Whitney U test in (**b**). Source data are provided as a Source Data file.

Fig. 4d, this coincided with a decrease in the average diameter of Munc13-1 nanoassemblies from 47.3 nm in unstimulated to 42.1 nm in stimulated neurons. Furthermore, we found a significant reduction in the center-to-center distance between nearest neighboring nanoassemblies in stimulated neurons to 119.6 ± 2.7 nm (Fig. 4e, f).

Of note, we identified similar Munc13-1 clusters in vivo at NMJs of P5 wt mice, using Airyscan microscopy (Fig. 4g). This super-resolution approach revealed that Munc13-1 is organized into discrete clusters arranged in a ring-like pattern. These clusters show substantial spatial overlap with RIM1, indicating that they are components of a shared presynaptic nano-architecture. This nanoscale organization closely resembles our observations using lattice-SIM imaging of axonal

growth cones in cultured motoneurons (Fig. 4c), further supporting the presence of spatially organized presynaptic Munc13-1 clusters at developing bona fide synaptic sites (Fig. 4g).

Collectively, our super-resolution microscopy data indicate that Munc13-1 clusters undergo a dynamic rearrangement upon Roscovitine stimulation, thus suggesting a role for Munc13-1 in modulation of presynaptic plasticity in motoneurons at the nanoscale.

Overexpression of Munc13-1 lacking the 3'UTR does not rescue the stimulus-dependent formation of new clusters in Smn KO neurons (Fig. 3d and Supplementary Fig. 3f). Likewise, stimulus-dependent formation of new clusters depends on protein synthesis (Supplementary Fig. 3h). Hence, we hypothesized that Munc13-1 is translated at

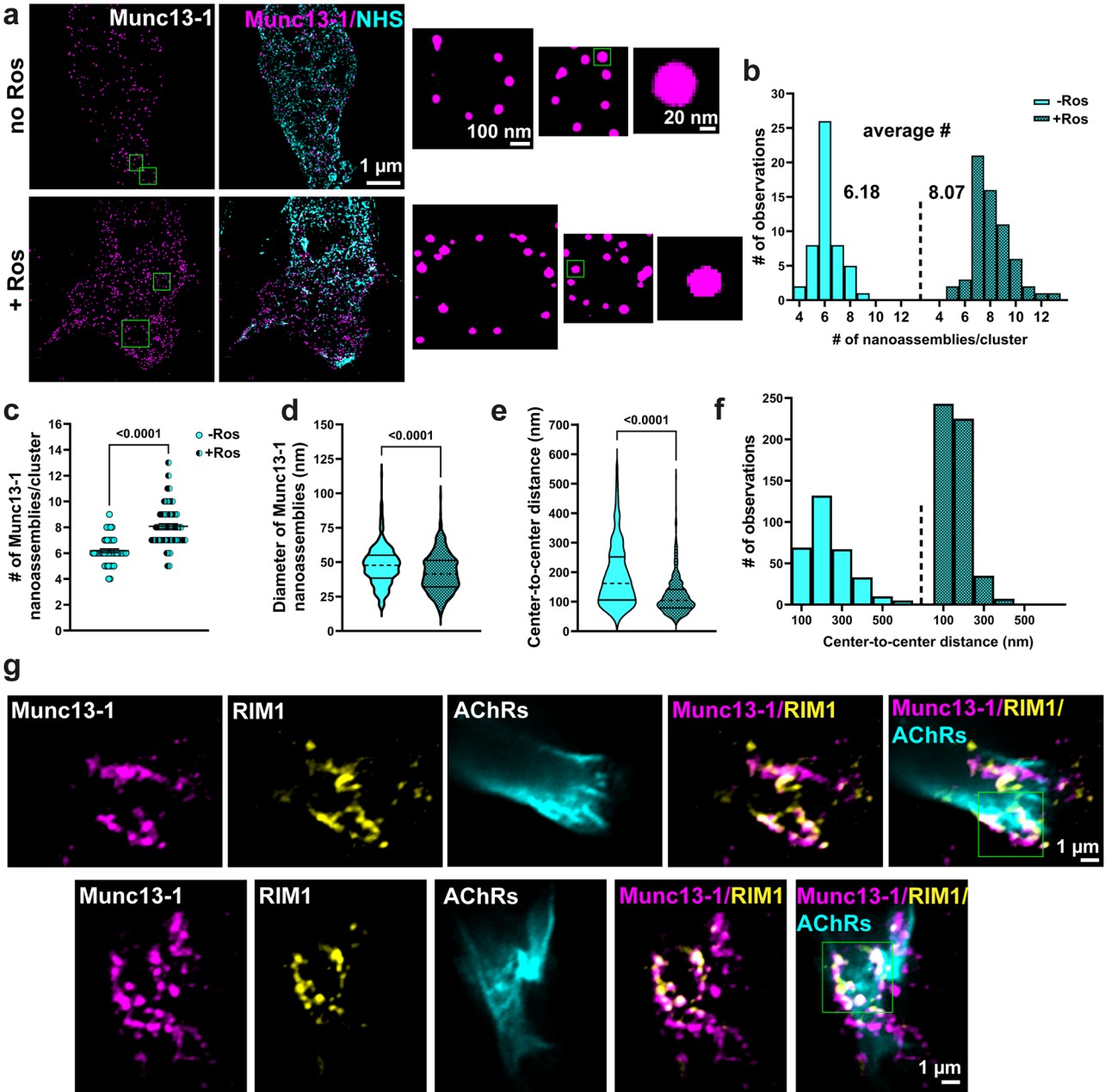

**Fig. 4 | Expansion microscopy shows the nanoscale transition of Munc13-1 assemblies after activation of presynaptic Ca²⁺ channels. a** Representative single optical sections from expansion microscopy images of axonal growth cones in cultured wt motoneurons, before and after stimulation of presynaptic voltage-gated Ca²⁺ channels with Roscovitine (Ros). Inset: Zoom-in shows Munc13-1 nanoassemblies within supramolecular clusters. See also Supplementary Movie 1 and 2. **b** and **c** The number of Munc13-1 nanoassemblies within clusters increases (**c**) ****$P < 0.0001$) upon stimulation (**b**, **c**) $n = 50$–63 clusters; $n = 18$–20 cells from $n = 3$ biological replicates). **d**–**f** Upon stimulation, the diameter of Munc13-1 nanoassemblies as well as their center-to-center distance decreases (**d**, **e**)

****$P < 0.0001$; $n = 313$–505 nanoassemblies; $n = 18$–20 cells from $n = 3$ biological replicates). **g** Representative single optical sections from Airyscan super-resolution images of neuromuscular junctions (NMJs) from P5 wt mice, showing immunolabeling for Munc13-1, RIM1, and AChRs. Munc13-1 is organized in discrete clusters within the presynaptic membrane, forming ring-like structures that show spatial overlap with RIM1 ($n = 45$ NMJs from $n = 8$ mice). Insets in merged images show Munc13-1/RIM1 clusters within the presynaptic membrane at NMJs. In a, scale bars are corrected to indicate pre-expansion dimensions. Two-tailed Mann-Whitney U test in c-e. Bars represent mean ± SEM. Source data are provided as a Source Data file.

ribosomes and immediately recruited into new presynaptic clusters. Intriguingly, upon stimulation with BDNF, Roscovitine, or KCl in wt motoneurons, we observed increased colocalization between Munc13-1 and the ribosomal marker Y10B, which was accompanied by enhanced colocalization between Munc13-1 and the synaptic protein marker Ca$_v$2.2 (a subtype of voltage-gated Ca²⁺ channels) (Supplementary Fig. 4a–f). In line with this, Roscovitine stimulation resulted in enhanced colocalization between Munc13-1 and the ribosomal markers RPL8 (Supplementary Fig. 5a, b) and Y10B (Supplementary Fig. 6a, b),

as well as with the synaptic protein markers Snap25 (Supplementary Fig. 5a, c) and Ca$_v$2.2 (Supplementary Fig. 6a, c) in both control and Smn KO$^{Rescue}$ motoneurons. In contrast, Smn KO and Smn KO$^{RescueΔ3'UTR}$ motoneurons failed to respond to the stimulation, as indicated by unaltered colocalization between Munc13-1/RPL8/Snap25 (Supplementary Fig. 5a–c), as well as between Munc13-1/Y10B/Ca$_v$2.2 (Supplementary Fig. 6a–c). As expected, motoneurons pretreated with anisomycin did not exhibit increased colocalization between Munc13-1 and RPL8 (Supplementary Fig. 6d).

These data provide evidence of Munc13-1 local translation in response to pharmacological stimulation, which might contribute to the modulation of SV release sites and plasticity, highlighting the relevance of this function for disease pathogenesis in SMA.

## Munc13-1 restoration rescues neuronal excitability in cultured motoneurons

Finally, we assessed whether reduced Munc13-1 synaptic levels contribute to disease pathogenesis in SMA and whether its restoration ameliorates neuromuscular dysfunction in SMA. To investigate this, we conducted a series of in vitro and in vivo experiments, including a SV recycling assay using an antibody targeting the luminal domain of Synaptotagmin1[75]. As shown in Fig. 5a, b, this assay revealed a significant increase in SV recycling, which is indicative of enhanced SV release events, in cultured Smn KO motoneurons after transduction with the Rescue viral construct and a partial rescue in RescueΔ3'UTR-transduced Smn KO motoneurons. These data suggest that while restoring Munc13-1 protein levels alone may partially rescue neurotransmission defects in spinal motoneurons, a complete rescue is only achieved when the local translation of Munc13-1 is also restored. This finding highlights the critical importance of locally translated Munc13-1. Treatments with CTX and TTX completely blocked the Synaptotagmin1 antibody uptake in cultured wt motoneurons (Supplementary Fig. 7a). Intriguingly, this improved neurotransmission was accompanied by increased levels of presynaptic AZ proteins, including RIM1/2, Piccolo, and Bassoon (Fig. 5c, d), in both Smn KO[Rescue] and Smn KO[RescueΔ3'UTR] motoneurons. This indicates a nonredundant role of Munc13-1 in SV priming and fusion, as recently shown[76]. At presynaptic AZs, RIM[77,78] and RIM-binding protein (RIM-BP)[79,80] mediate clustering of voltage-gated $Ca^{2+}$ channels[36,39], and a complex consisting of Liprin-α and PTPσ regulates the priming site assembly for SVs[81]. The calcium coupling by tethering SVs near voltage-gated $Ca^{2+}$ channels is crucial for ensuring the precise timing of SV release events in coordination with these channels[43]. Interestingly, SMA mice exhibit deficiencies in clustering of voltage-gated $Ca^{2+}$ channels, which in turn results in reduced excitability[15,65,82]. Therefore, we sought to analyze the potent rescue effect of Munc13-1 overexpression on presynaptic levels of voltage-gated $Ca^{2+}$ channels and their functionality. Intriguingly, the expression of both Rescue and RescueΔ3'UTR viruses increased the levels of $Ca_v2.2$ channels in axonal growth cones of cultured Smn KO motoneurons (Fig. 6a, b).

Consistent with this, $Ca^{2+}$ imaging revealed an increase in spontaneous $Ca^{2+}$ transients within axonal growth cones in both Smn KO[Rescue] and Smn KO[RescueΔ3'UTR] motoneurons (Fig. 6c,d). The $Ca^{2+}$ transients measured by $Ca^{2+}$ imaging reflect spontaneous events in cultured motoneurons, triggered by both action potentials and local $Ca^{2+}$ currents. While the frequency of $Ca^{2+}$ transients was increased in both KO[Rescue] and Smn KO[RescueΔ3'UTR] motoneurons, the amplitude remained significantly lower in Smn KO[RescueΔ3'UTR] motoneurons (Fig. 6e). Moreover, measuring the evoked response after membrane depolarization revealed a decreased response in Smn KO and Smn KO[RescueΔ3'UTR] neurons (Fig. 6f-h). In control experiments, motoneurons were treated with CTX for 1 h prior to $Ca^{2+}$ imaging. Under these conditions, no spontaneous $Ca^{2+}$ transients were observed (Supplementary Fig. 7b).

From these data, we conclude that diminished Munc13-1 local translation is accompanied by reduced calcium influx in Smn KO and Smn KO[RescueΔ3'UTR] motoneurons. However, the underlying cause, whether upstream or local, cannot be determined from these experiments.

## Munc13-1 restoration recovers motor function in SMA mice

To establish whether restoring Munc13-1 function could rescue motor defects in SMA in vivo, we generated a Cre/loxP conditional Munc13-1 knock-in rescue mouse model. The rescue cassette harboring a conditional Munc13-1+SynPhy3'UTR cassette (Fig. 7a) was inserted into the ROSA26 locus. Due to the presence of loxP-flanked stop sequences, we cross-bred SMA mice with conditional Munc13-1 knock-in and a Nestin-Cre driver line[83] to obtain the following genotype: $Smn^{-/-},Hung^{tg/+},R26Unc13-1^{tg/+},Cre^{tg/+}$ (referred thereafter to as Smn KO[R26Unc13-1tg/+]). The selection of Nestin-Cre was based on the observation that Smn loss of function in all neurons and glia affects the neuromuscular synaptic transmission, thereby leading to animal death[84]. As depicted in Fig. 7b, the expression of $R26Unc13-1$ rescue allele increases with postnatal age in cortical tissues from Munc13-1 knock-in mice cross-bred with Nestin-Cre. Then, we cross-bred Munc13-1 knock-in mice with $Munc13-1^{-/-}$ mice and prepared cultured motoneurons from E12.5 embryos. We could detect the R26Unc13-1-derived transgenic protein by Western blot in lysates from $Munc13-1^{-/-}$ motoneurons transduced with a Cre-expressing virus (Fig. 7c).

Denervation of motor units in skeletal muscles is a hallmark of SMA that occurs before spinal motoneuron degeneration in SMA patients[85], as well as in SMA animal models[3,5]. Therefore, we investigated potential signs of ameliorated synapse degeneration at NMJs of vulnerable TVA muscles in our SMA rescue mouse model at the disease end stage (P10) (Fig. 7d). We observed that approximately 15% of the TVA endplates were fully denervated in Smn KO mice, compared to 3% denervation in Smn KO[R26Unc13-1tg/+] mice and 1% in control mice. Fully denervated endplates were defined as AChR+ NMJs lacking any presynaptic SynPhy/NFH coverage. This attenuated denervation in Smn KO[R26Unc13-1tg/+] mice correlated with an upregulation of Munc13-1 and $Ca_v2.1$ in presynaptic membranes (Supplementary Fig. 7c, e). Together, these data indicate diminished synapse degeneration of NMJs in vulnerable TVA muscles in Smn KO[R26Unc13-1tg/+] mice (Fig. 7d, e). These findings align with previous studies reporting endplate denervation in Taiwanese-SMA mice used in this study[86–91], but contradict a recent study that examined the NMJ denervation of the axial quadratus lumborum and distal tibialis anterior muscles[92].

In SMA, the proximal and axial skeletal muscles are particularly prone to weakness[93], and the motoneurons innervating these muscles are significantly affected, leading to their preferential loss[9,94]. This includes motoneuron pools residing in the lumbar L1 and L2 segments of the spinal cord[92,95], which have been reported to exhibit neurodegeneration in the Taiwanese-SMA mouse model[86,87,89–91,96], except for one study[92]. The varying levels of vulnerability within spinal motoneurons correspond to transcriptional alterations across motoneuron pools that are differentially affected in SMA[97]. Accordingly, we examined the number of ChAT+ motoneurons within the ventral horn of L1/L2 segments of the lumbar spinal cords in P10 mice to assess the possible neuroprotective effect of Munc13-1 on motoneuron degeneration in SMA (Fig. 7f).

Of note, the motoneuron counts within the L1/L2 segments of the spinal cords revealed 28% loss in Smn KO compared to 5% loss in Smn KO[R26Unc13-1tg/+] littermates (Fig. 7g).

Next, we performed behavioral motor assessments. As demonstrated in Fig. 8a–c, the data showed improved motor function and attenuated muscle weakness in P10 Smn KO[R26Unc13-1tg/+] littermates compared to Smn KO. This was evidenced by measurements of time to right (Fig. 8a) and assessments of muscle strength in both forelimbs (Fig. 8b) and hind limbs (Fig. 8c). Finally, the average survival was significantly increased from 10 days in Smn KO to 14 days in Smn KO[R26Unc13-1tg/+] littermates (Fig. 8d, e). Notably, 100% of Smn KO[R26Unc13-1tg/+] pubs survived until P10, which is the average survival of Smn KO mice, whereas only 67% of their Smn KO littermates survived until P10.

We did not observe any overt phenotypic differences in Ctrl[R26Unc13-1tg/+] pubs (Ctrl[Rescue]). These mice exhibited no noticeable abnormalities or behavioral changes compared to control mice (Fig. 8a–c).

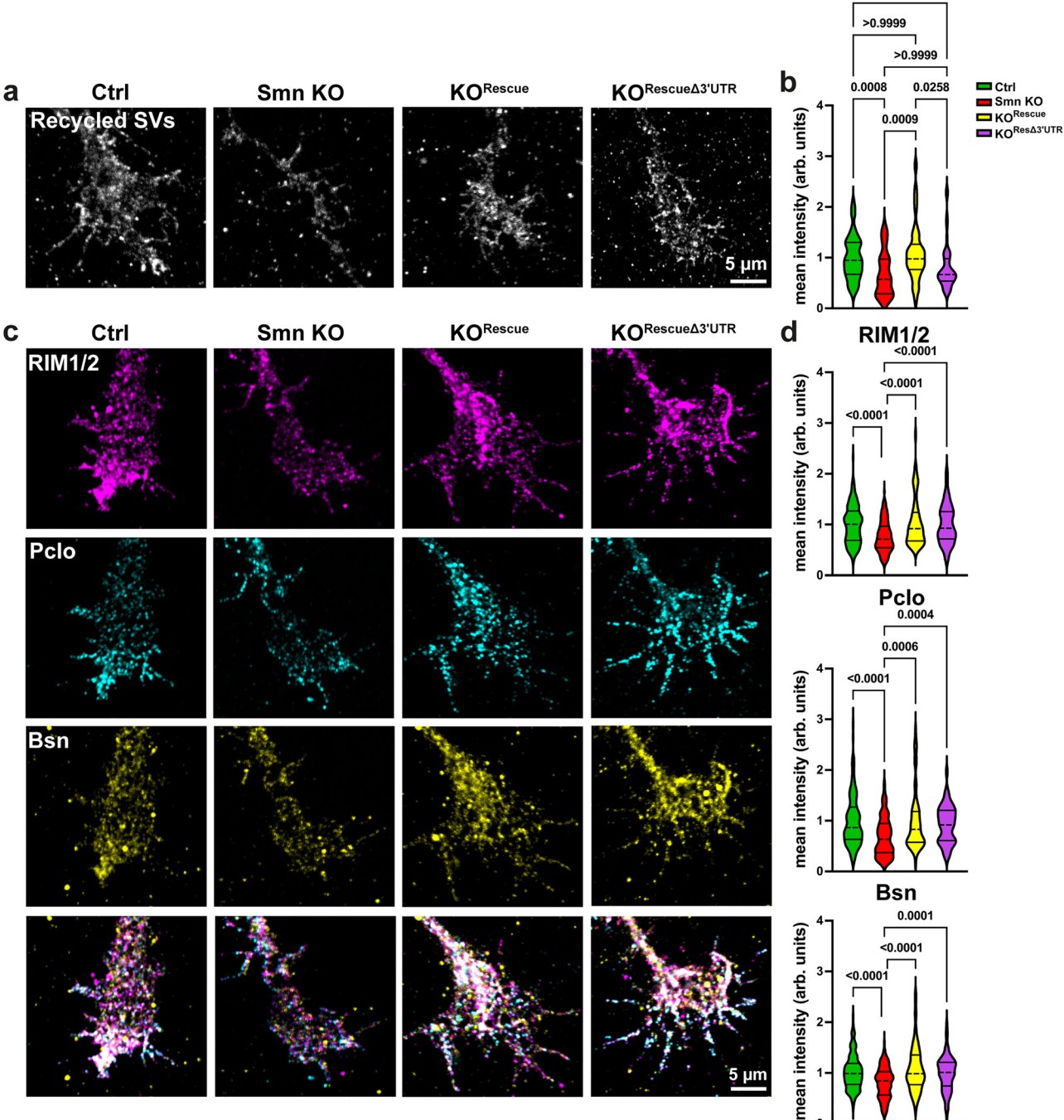

**Fig. 5 | Munc13-1 overexpression increases SV release events in cultured Smn KO motoneurons. a** Representative images of axonal growth cones indicating the uptake of Synaptotagmin antibody in cultured motoneurons. **b** SV recycling assay reveals improved neurotransmitter release in Smn KO^Rescue and a partial rescue in Smn KO^RescueΔ3'UTR motoneurons (***P = 0.0008, ***P = 0.0009, *P = 0.0352, *P = 0.0258; n = 35–66 cells from n = 4 biological replicates). **c** Representative images of axonal growth cones of cultured motoneurons transduced with Munc13-1 rescue constructs, stained against AZ markers. (**d**) RIM1/2 (****P < 0.0001; n = 69–184 cells from n = 4 biological replicates), Piccolo (****P < 0.0001, ***P = 0.0004, ***P = 0.0006; n = 52–145 cells from n = 5 biological replicates), and Bassoon (****P < 0.0001, ***P = 0.0001; n = 82–176 cells from n = 5 biological replicates) are restored in axonal growth ones of Smn KO^Rescue and Smn KO^RescueΔ3'UTR motoneurons. One-way ANOVA with Dunn's post-test. Source data are provided as a Source Data file.

We included both male and female littermates in our motor assessment experiments. However, we did not observe any sex-specific differences in the rescue outcomes in the experiments shown in Fig. 8. Therefore, the data suggest that the results are consistent across different sexes.

Collectively, these data demonstrate that restoring Munc13-1 synaptic levels beneficially affects synapse degeneration, motor function, and neurodegeneration in SMA mice.

## Axonal localization of *UNC13A* mRNA and protein are perturbed in hiPSC-derived motoneurons from SMA patients

To understand the importance of UNC13A synaptic function on SMA pathophysiology, we investigated the role of SMN in axonal translocation and translation of UNC13A in human induced pluripotent stem cell-derived (hiPSC) motoneurons from two type I and one type II SMA patients (Supplementary Fig. 8a–c). To this end, we cultured hiPSC-derive motoneurons in compartmentalized chambers and analyzed

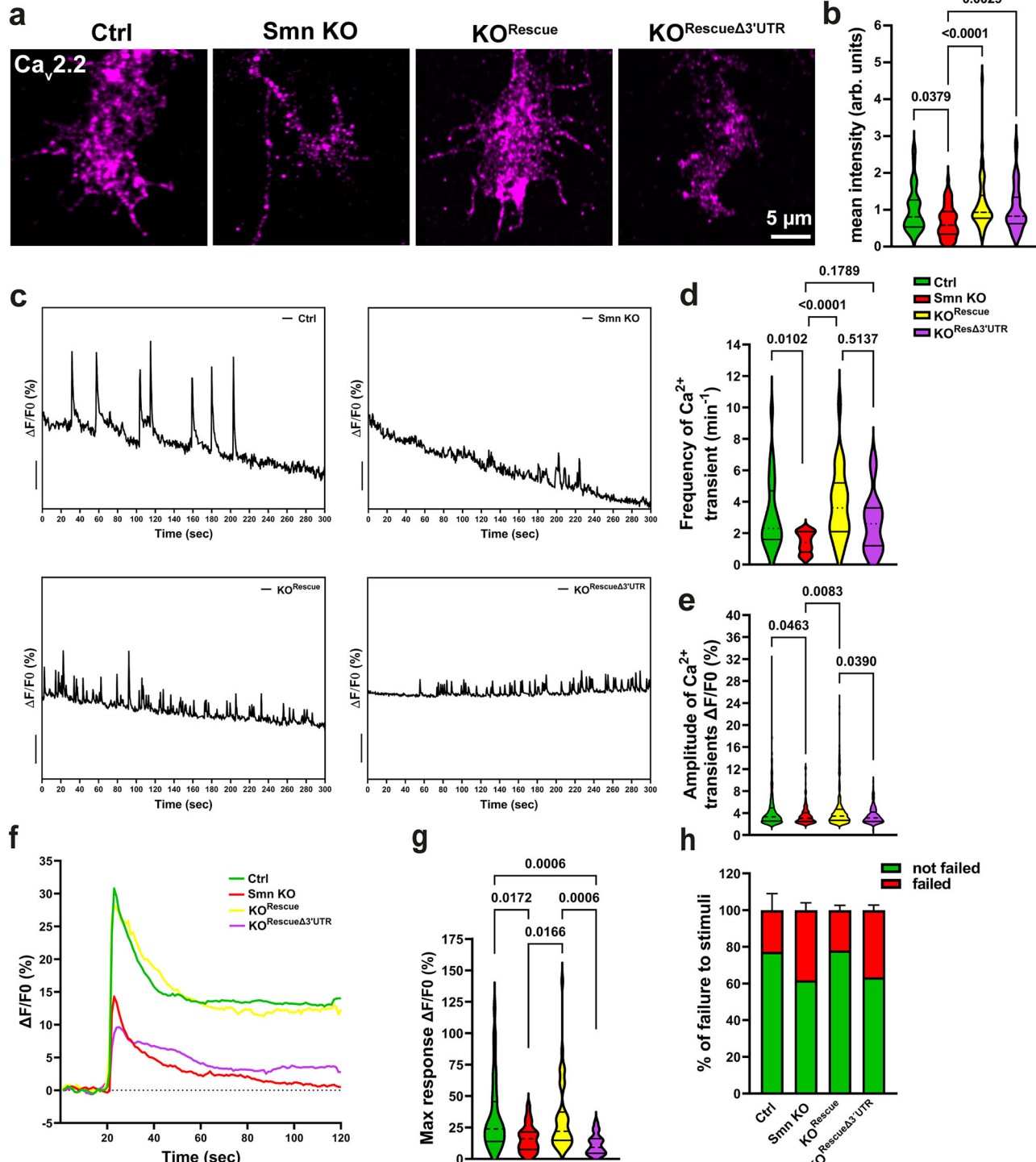

**Fig. 6 | Munc13-1 local translation rescues neuronal excitability in cultured Smn KO motoneurons. a** Representative images of axonal growth cones of cultured motoneurons stained against voltage-gated Ca²⁺ channels (Ca$_V$2.2). **b** Quantification indicates that the levels of Ca$_V$2.2 are restored in axonal growth cones of both Smn KO$^{Rescue}$ and Smn KO$^{Rescue\Delta3'UTR}$ motoneurons (*$P$ = 0.0379, **$P$ = 0.0029, ****$P$ < 0.0001; $n$ = 42–91 cells from $n$ = 3 biological replicates). **c**, **d** Ca²⁺ imaging shows increased frequency of spontaneous Ca²⁺ transients in axonal growth cones of Smn KO$^{Rescue}$ and Smn KO$^{Rescue\Delta3'UTR}$ motoneurons compared to Smn KO motoneurons (*$P$ = 0.0102, ****$P$ < 0.0001; $n$ = 15–26 cells from $n$ = 3 biological replicates). **e** The amplitude of spontaneous Ca²⁺ transients is significantly elevated in

Smn KO$^{Rescue}$ motoneurons, but only slightly increased in Smn KO$^{Rescue\Delta3'UTR}$ (*$P$ = 0.039, *$P$ = 0.0463, **$P$ = 0.0083; $n$ = 172–489 transients in n = 15-26 cells from $n$ = 3 biological replicates). **f** Growth cones were depolarized by 90 mM KCl and Ca²⁺ transients were measured. Graph shows the average maximum response to depolarization. **g** Maximum response is lower in Smn KO and Smn KO$^{Rescue\Delta3'UTR}$ motoneurons (*$P$ = 0.0172, 0.0166, ***$P$ = 0.0006; $n$ = 17–46 cells from $n$ = 4 biological replicates). **h** The percentage of failure response to membrane depolarization is higher in Smn KO and Smn KO$^{Rescue\Delta3'UTR}$ motoneurons ($n$ = 4 biological replicates). One-way ANOVA with Dunn's post-test. Bars represent mean ± SEM. Source data are provided as a Source Data file.

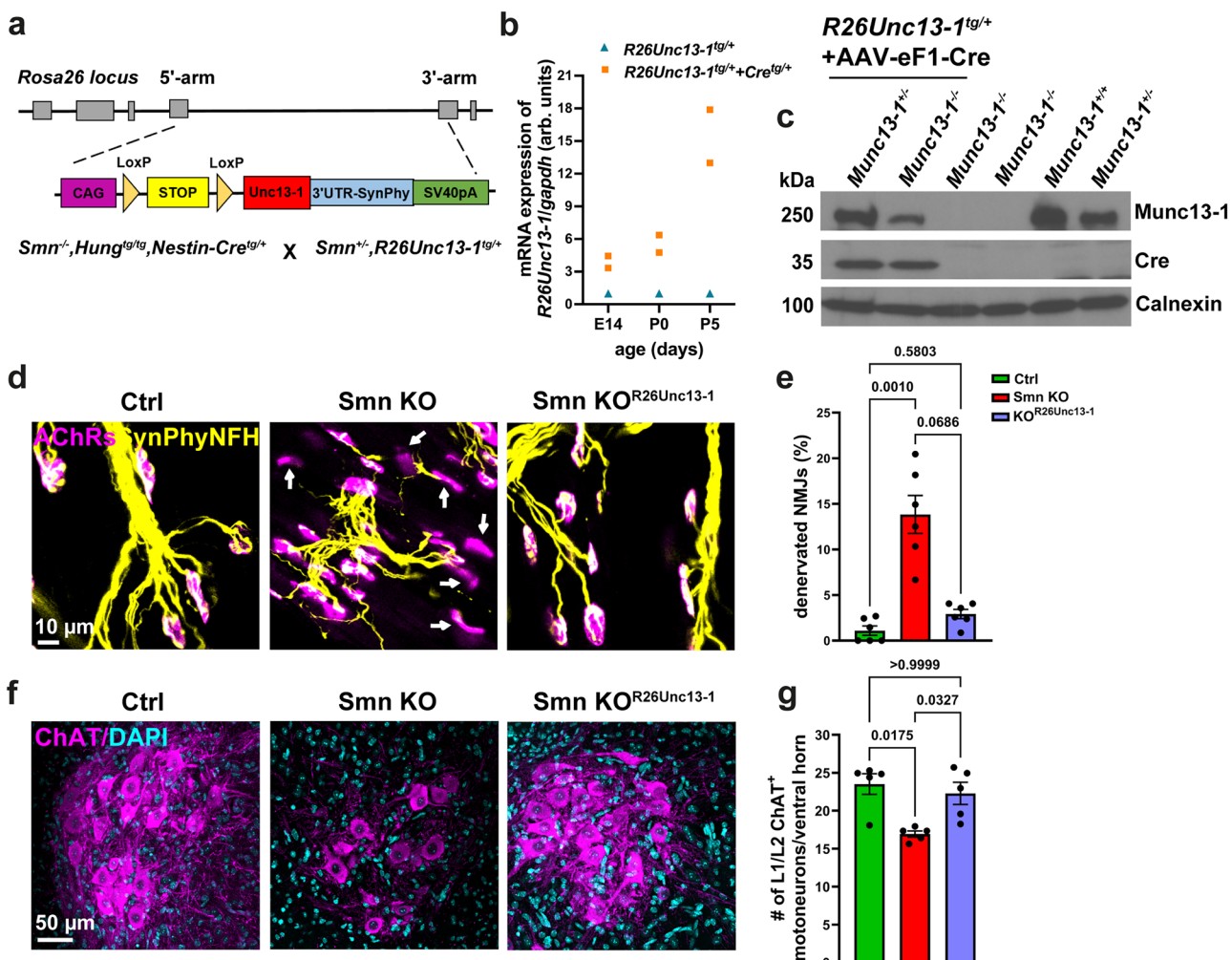

**Fig. 7 | Restoration of Munc13-1 attenuates synapse degeneration in SMA mice.**
**a** Scheme of the Cre/loxP conditional Munc13-1 rescue cassette inserted into the mouse *ROSA26 locus* and the breeding with Smn KO mice. **b** *R26Unc13-1^tg/+* mice were cross-bred with *Nestin-Cre^tg/+*. qRT-PCR indicates the expression of *Munc13-1* rescue allele in the brain of *R26Unc13-1^tg/+* animals (*n* = 2 mice/genotype/age). **c** Motoneurons with *Munc13-1^+/-,R26Unc13-1^tg/+* and *Munc13-1^-/-,R26Unc13-1^tg/+* genotypes were transduced with adeno-associated virus-eF1-Cre (AAV-eF1-Cre) and cultured. Protein product of the *Munc13-1* rescue allele is detectable in *Munc13-1^-/-,R26Unc13-1^tg/+* motoneurons by Western blot (representative of *n* = 1 biological replicate). **d** Representative images of NMJs of the TVA muscle from P10 littermates stained against Synaptophysin (SynPhy), Neurofilament H (NFH), and AChRs. White arrows indicate denervated NMJs. **e** Graph shows increased percentage of fully innervated NMJs in Smn KO^R26Unc13-1tg/+^ compared to Smn KO littermates (**P* = 0.001, *P* = 0.0686; *n* = 453–810 NMJs from *n* = 6 biological replicates). **f** Representative images of ChAT+ spinal motoneurons within L1/L2 lumbar segments of the spinal cord from P10 littermates. Motoneurons are stained with ChAT antibody and nuclei are stained with DAPI. **g** Graph represents increased number of ChAT+ spinal motoneurons within L1/L2 lumbar segments of the spinal cord in Smn KO^R26Unc13-1tg/+^ littermates compared to Smn KO (**P* = 0.0175, **P* = 0.0327 from *n* = 5 biological replicates). One-way ANOVA with Dunn's post-test in **e** and **g**. Bars represent mean ± SEM. Source data are provided as a Source Data file.

the mRNA levels of UNC13A in axonal compartments by qRT-PCR (Fig. 9a). Importantly, marked reduction in *UNC13A* mRNA levels in axons of cultured hiPSC-derived motoneurons was present in all three SMA patient lines, while *UNC13A* mRNA levels were not significantly altered in the somata (Fig. 9b). Then, we generated a Rescue lentivirus construct expressing the human UNC13A coding region fused to the 3'UTR of human *Synaptophysin1* mRNA (UNC13A + SYP3'UTR). Western blot analysis revealed increased UNC13A expression in total lysates of transduced cultured hiPSC-derived motoneurons (Supplementary Fig. 8d). In agreement with our data from SMA mice, transduction of SMA-hiPSC-derived motoneurons with the UNC13A Rescue lentivirus restored the attenuated translocation of *UNC13A* mRNA in axons (Fig. 9b).

Next, we assessed the protein levels of UNC13A as well as $Ca_v2.2$ in axonal growth cones of cultured SMA-hiPSC-derived motoneurons (Fig. 9c and Supplementary Fig. 8e). As illustrated in Fig. 9d, SMA-hiPSC-derived motoneurons showed reduced levels of UNC13A and

$Ca_v2.2$, which is consistent with our observations with SMA mice (Supplementary Fig. 1e, f). Similarly, expression of the UNC13A Rescue construct resulted in increased levels of both UNC13A and $Ca_v2.2$ in the axonal growth cones in cultured SMA-hiPSC-derived motoneurons (Fig. 9d and Supplementary Fig. 8e).

Collectively, these data indicate that similar to the mouse model, in human motoneurons, axonal mRNA transport and local translation of UNC13A depend on SMN and are diminished in SMA.

## Discussion

In motoneuron diseases like ALS[98,99] and SMA[14], defective synaptic transmission contributes to the degeneration of NMJs. In SMA, however, a direct link between SMN cellular functions and the maintenance of synaptic activity is only now emerging. SMN plays a role in the SNARE complex assembly at NMJs and the expression of a G470R variant of the chaperon protein Hspa (Hspa8^G470R^) rescues the defective SNARE complex assembly and NMJ function in SMA mice[29,30].

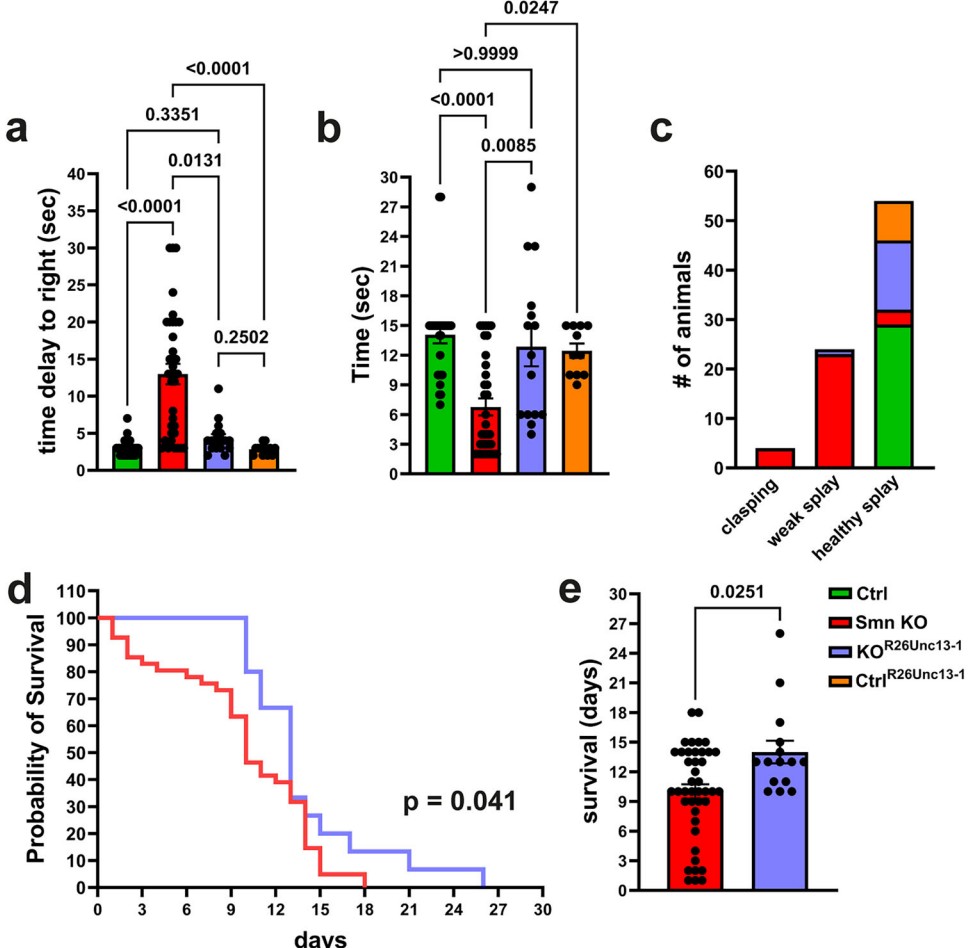

**Fig. 8 | Restoration of Munc13-1 rescues motor function in SMA mice. a** Graph shows decreased time taken to self-right in Smn KO$^{R26Unc13-1tg/+}$ compared to Smn KO littermates (*$P = 0.0131$, ****$P < 0.0001$ from $n = 34$ Control (Ctrl), $n = 36$ Smn KO, $n = 17$ Smn KO$^{R26Unc13-1tg/+}$, and $n = 12$ Ctrl$^{R26Unc13-1tg/+}$ mice). **b** Smn KO$^{R26Unc13-1tg/+}$ litter-mates show improved forelimb motor performance compared to Smn KO (*$P = 0.0247$, **$P = 0.0085$, ****$P < 0.0001$ from $n = 29$ Ctrl, $n = 34$ Smn KO, $n = 15$ Smn KO$^{R26Unc13-1tg/+}$, and $n = 11$ Ctrl$^{R26Unc13-1tg/+}$ mice). **c** Smn KO$^{R26Unc13-1tg/+}$ littermates show improved hind limb clasping posture compared to Smn KO ($n = 8–30$ mice/ genotype). **d** Kaplan-Meier curve indicates survival of Smn KO$^{R26Unc13-1tg/+}$, control and Smn KO mice (*$P = 0.041$ from $n = 42$ Smn KO, and $n = 15$ Smn KO$^{R26Unc13-1tg/+}$ mice). **e** Survival of Smn KO$^{R26Unc13-1tg/+}$ mice is increased compared to Smn KO (*$P = 0.0251$ from $n = 42$ Smn KO, and $n = 15$ Smn KO$^{R26Unc13-1tg/+}$ mice). Two-tailed Mann-Whitney U test in (**a**, **b**, and **e**). Log-rank (Mantel-Cox) test in (**d**). Bars represent mean ± SEM. Source data are provided as a Source Data file.

Moreover, SMN function for the assembly of U7 snRNP is required for NMJ integrity and the restoration of U7 snRNP assembly through co-expression of Lsm10 and Lsm11 proteins rescues NMJ denervation and restores synaptic transmission in SMA mice[31]. Our study demonstrates that presynaptic Munc13-1 protein levels are reduced at NMJs in SMA mice and in hiPSC-derived motoneurons from SMA patients, suggesting that synaptic plasticity defects in SMA may stem from Munc13-1 deficiency. Interestingly, this reduction correlates with perturbed axonal localization of *Munc13-1* transcripts, indicating that these transcripts undergo SMN-dependent transport mechanisms. Importantly, iCLIP assays have shown that *Munc13-1* mRNA binds to hnRNP R via its 3'UTR[63]. hnRNP R is an RNA-binding protein that interacts with Smn to facilitate the mRNA transport for various axonal targets[63,64]. Interestingly, hnRNP R levels are reduced in axons of Smn KO spinal motoneurons[64], suggesting that diminished hnRNP R axonal levels might lead to the mislocalization of *Munc13-1* transcripts in SMA.

Evidence for the role of locally translated proteins in the activity-dependent regulation of synaptic plasticity and their contribution to synapse degeneration is emerging[100,101]. Local translation is implicated in synaptogenesis, plasticity, and axon regeneration through rapid modulation of the local proteome in response to extracellular cues[102]. Alterations in local protein synthesis contribute to the pathogenesis of diverse neurodegenerative disorders such as Fragile X Syndrome, Alzheimer's disease, Parkinson's disease, ALS, and SMA[103,104]. A central mechanism of impaired local translation involves perturbed mRNA localization including transcripts encoding synaptic components[25,104,105]. Loss of dendritic synthesis of Calcium/calmodulin-dependent protein kinase II α (CaMIIKα) abolishes the induction of long-term plasticity (LTP), thereby affecting learning and memory[106]. While most previous studies focused on the role of de novo synthe-sized proteins in dendritic presynaptic functions and postsynaptic plasticity[107], elucidating the specific function of locally translated synaptic proteins at axonal presynapses remains challenging. In our study, we could detect Munc13-1 local translation at axonal growth cones in cultured spinal motoneurons and in synaptosome fractions from mouse cortical neurons. This aligns with previous studies show-ing Munc13-1 mRNA localization and translation in axons and neuropils[108,109] and its mRNA localization in axonal growth cones[57,58] and neuropils[59] of neurons in the brain[109]. However, these studies report relatively low enrichment of *Munc13-1* mRNAs in these com-partments, suggesting a low rate of local translation in these neurons. Unlike central neurons, local translation is particularly important for spinal motoneurons due to their long axons, which can extend up to one meter in length in humans, the high number of axon terminals, and

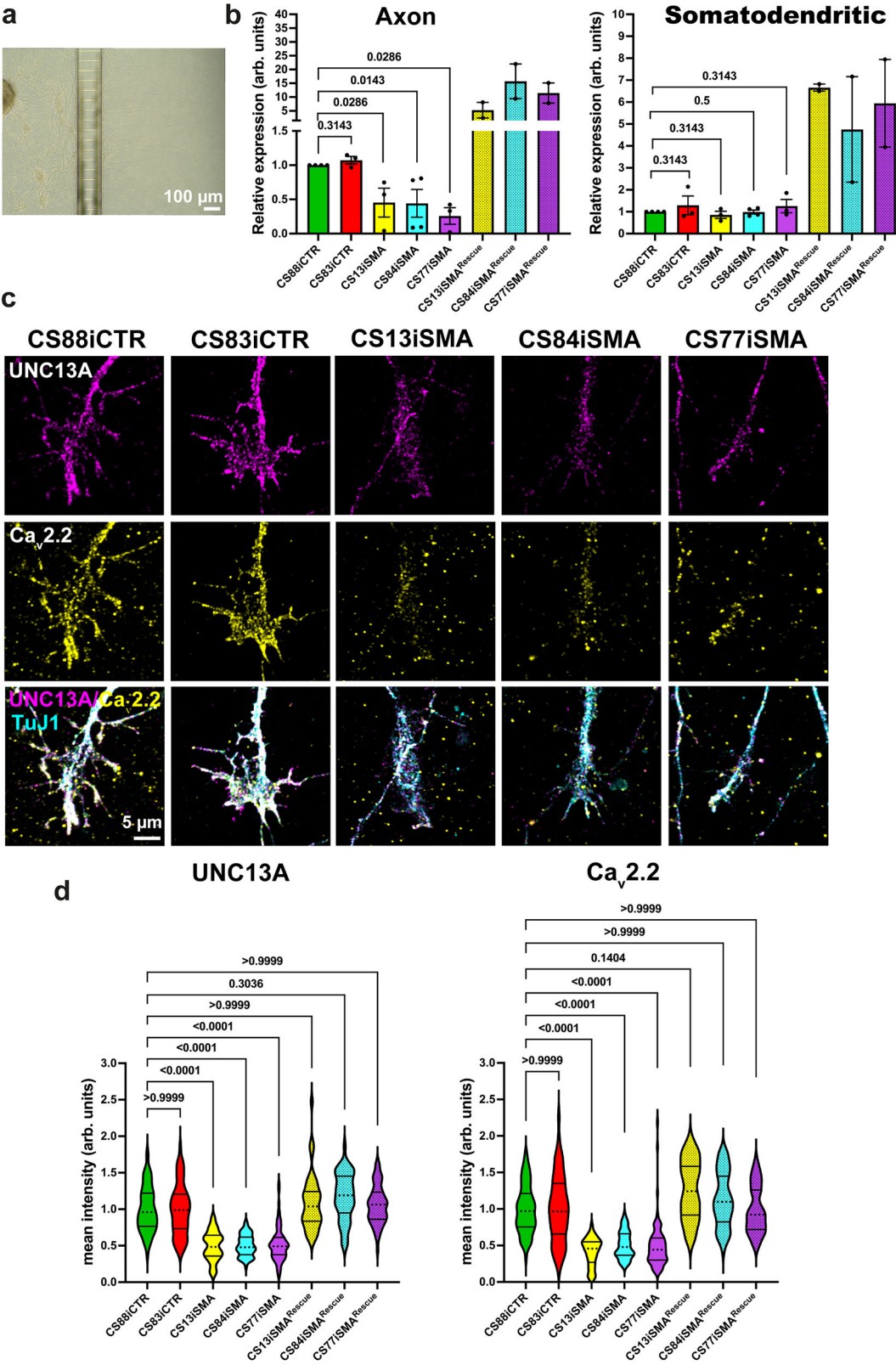

their distinct anatomical and morphological features[110]. These unique characteristics of spinal motoneurons distinguish their synapses from central synapses[111], rendering them particularly vulnerable to degeneration in response to alterations in protein synthesis, especially within the axon. Moreover, despite a conserved SV exocytosis machinery, key differences exist between the central synapses and NMJs as well as between excitatory and inhibitory synapses[111]. These differences, including the number and size of AZs, the molecular composition of SV release machinery[111], the high frequency of voltage-gated $Ca^{2+}$ channel openings in response to continuing tetanic excitation of motoneurons in tonic muscle groups and the spatial coupling of voltage-gated $Ca^{2+}$ channels to the AZ[112], may explain why, in contrast to NMJs, local translation of Munc13-1 occurs at reduced rates in central synapses.

**Fig. 9 | Axonal localization of the *UNC13A* mRNA and protein are impaired in hiPSC-derived motoneurons from SMA patients. a** Image shows growing of axons of hiPSC-derived motoneurons cultured in compartmentalized chambers. **b** qRT-PCR indicates reduced mRNA levels of UNC13A in distal axons (left panel) of hiPSC-derived motoneurons from two type I and one type II SMA patients compared to two healthy individuals (*$P = 0.0286$ for C88iCTR versus C13iSMA and C77iSMA, *$P = 0.0143$ for C88iCTR versus C84iSMA from $n = 3$ biological replicates for C88iCTR, C83iCTR, C13iSMA, C77iSMA lines, and $n = 4$ biological replicates for C84iSMA line). *UNC13A* mRNA levels are increased in both somatodendritic and axonal compartments in SMA-hiPSC-derived motoneurons transduced with Rescue lentivirus expressing UNC13A + SYP3'UTR ($n = 2$ biological replicates). **c** Representative images of axonal growth cones of cultured DIV25 hiPSC-derived motoneurons from SMA patients and control individuals stained against TuJ1, UNC13A, and Ca$_v$2.2. Representative images of SMA-hiPSC-derived motoneurons transduced with UNC13A + SYP3'UTR Rescue virus are included in Supplementary Fig. 8e. **d** UNC13A and Ca$_v$2.2 protein levels are reduced in axonal growth cones of cultured SMA-hiPSC-derived motoneurons compared to control (****$P < 0.0001$; $n = 48$-$82$ cells from $n = 3$ biological replicates). In SMA-hiPSC-derived motoneurons, UNC13A and Ca$_v$2.2 levels are restored in axonal growth cones upon expression of UNC13A + SYP3'UTR Rescue construct. One-tailed Mann-Whitney U test in (**b**) and One-way ANOVA with Dunn's post-test in (**d**). Bars represent mean ± SEM. Source data are provided as a Source Data file.

In most synapses, AZs contain one or more release sites, with discrete domains believed to mediate the fusion of a single SV[113]. As demonstrated recently, within these ~29 nm nanodomains 6 Munc13-1 molecules assemble under a single SV[73]. In cultured hippocampal neurons, Munc13-1 molecules form multiple and discrete supramolecular self-assemblies that serve as independent vesicular release sites by recruiting syntaxin-1[74]. Within these supramolecular self-assemblies, the number of Munc13-1 molecules directly determines the quantal release, enabling a stable synaptic weight on neuronal circuits. Similarly, at *Drosophila* neuromuscular synapses, Unc13A is responsible for stabilizing AZ positioning and creating new release sites at specific sub-AZ positions, a critical function for maintaining synaptic plasticity at NMJs[114]. Here, using 8-fold ExM in combination with lattice-SIM, we resolved the molecular assemblies of Munc13-1 within presynaptic membranes at the nanoscale and provided evidence that pharmacological stimulation increases the Munc13-1 nanoassemblies from 6 in unstimulated neurons to 8, and decreases the distance between the neighboring nanoassemblies. This remodeling of the presynaptic supramolecular clusters in response to pharmacological stimulation contributes to synaptic plasticity and relies on the Munc13-1 local translation. This might occur directly at ribosomes within these supramolecular clusters, leading to the recruitment of neosynthesized Munc13-1 molecules into newly assembled clusters. This aligns with previous studies demonstrating that monosomes actively translate synaptic mRNAs within neuronal processes[108]. Thus, perturbed local translation of Munc13-1 due to its mRNA mislocalization may contribute to the plasticity defects observed in SMA mouse models[14,16].

As demonstrated recently, pharmacological induction of neuronal excitability shows neuroprotective effects on ALS-hiPSC-derived motoneurons[115]. In SMA mice, treatment of animals with a potassium channel blocker that increases neuronal activity improves motor function[116], and treatment with Roscovitine that enhances Ca$^{2+}$ influx and transmitter release beneficially affects the survival of SMA mice[65]. Continuous firing of motoneurons at ~50 Hz is essential for sustained muscle contraction and relies on repeated activation of presynaptic voltage-gated Ca$^{2+}$ channels[117], a mechanism enhanced by Roscovitine stimulation[67]. This pathway is pathophysiologically relevant, as demonstrated in disorders such as Lambert-Eaton Syndrome, where voltage-gated Ca$^{2+}$ channel dysfunction impairs neurotransmission, underscoring the therapeutic potential of voltage-gated Ca$^{2+}$ channel modulators, including Roscovitine, in neuromuscular diseases[68].

Our data show that restoring presynaptic Munc13-1 levels in cultured Smn KO motoneurons rescues synaptic plasticity and neurotransmission defects. Notably, this rescue is accompanied by a restoration of key AZ proteins, including RIM1/2, Piccolo, Bassoon, and voltage-gated Ca$^{2+}$ channels at presynaptic terminals. RIM1 is known to activate Munc13-1 by disrupting its autoinhibitory homodimerization[118] and to recruit voltage-gated Ca$^{2+}$ channels to the presynaptic membrane, thereby coupling vesicle priming to calcium entry[119]. Loss of RIM and ELKS leads to destabilization of Munc13-1, Piccolo, and Bassoon, and RIM-BP, resulting in disassembly of the AZ and impaired SV docking[120]. While Munc13-1 functions downstream of RIM in AZ assembly, the observed restoration of RIM and voltage-gated Ca$^{2+}$ channel levels upon Munc13-1 overexpression suggests a potential reciprocal stabilization mechanism, the molecular details of which remain to be elucidated. Importantly, our experiments with different rescue constructs show that full rescue requires restoring *Munc13-1* mRNA localization for its local translation, as restoring protein levels alone only achieves partial rescue. In vivo rescue experiments with SMA mice cross-bred with conditional Munc13-1 knock-in mice with modified 3'UTR demonstrate that these mice exhibit improved motor function, ameliorated NMJ pathogenesis, attenuated neurodegeneration, and increased survival. This mRNA editing thus indicates that therapies aimed at mitigating neuromuscular pathology in SMA may need to focus not only on increasing Munc13-1 protein levels but also on correcting mRNA transport and local translation processes to fully rescue synaptic functions.

## Methods
### Animals
SMA litters, *Smn$^{-/-}$,Hung$^{tg/+}$*, (referred to as Smn KO in the text), and control litters, *Smn$^{+/-}$,Hung$^{tg/+}$*, (referred to as control in the text) were offspring of two mouse strains (I) Smn$^{+/-}$ that is hemizygote for the Smn$^{tm1Hung}$ targeted mutation, and (II) *Smn$^{-/-}$,Hung$^{tg/tg}$* that is homozygote for the Smn$^{tm1Hung}$ targeted mutation as well as for the transgenic Hung allele, Tg(SMN2)2Hung[60]. The *Smn$^{-/-}$,Hung$^{tg/tg}$* line was generated by crossing transgenic mice carrying the human *SMN2* gene with mice heterozygous for the targeted Smntm1Hung mutation. Mice heterozygous for the Smn1tm1Hung knockout allele are phenotypically normal. The number of human *SMN2* transgene copies is strongly correlated with the severity of the neurodegenerative phenotype. Mice hemizygous for Tg(SMN2)2Hung carry approximately two copies of the transgene, whereas mice homozygous for Tg(SMN2)2Hung carry approximately four copies. Mice homozygous for the Smn1tm1Hung knockout allele and hemizygous for Tg(SMN2)2Hung exhibit a Type 1 SMA-like phenotype and die at around 10 days of age. In contrast, mice homozygous for both the Smn1tm1Hung knockout allele and the Tg(SMN2)2Hung transgene display a diverse Type 3 SMA phenotype, are viable, fertile, and exhibit shortened and thickened tails[60]. Both mouse lines were obtained from Jackson repository and maintained on a C57BL/6 J background (C57BL/6 J.29P2-Smn1Hung< tm1Msd >/J). C57BL/6 J mice (referred to as wt in the text) were used for all control experiments as well as for expansion microscopy and were obtained from Charles River repository. Munc13-1 KO mice (*Munc13-1$^{-/-}$*)[37,121] were originally obtained from Goettingen, Germany and cross-bred in-house. Nestin-Cre transgenic mice (C57BL/6 J.Cg(Nes-cre)1Kln/J)[122] were cross-bred in-house. The *R26Unc13-1$^{tg/+}$* knock-in mouse model was designed and cloned by M. Moradi and generated at the Czech Centre for Phenogenomics in Prague, Czech Republic (https://www.phenogenomics.cz). *Smn$^{+/-}$,R26Unc13-1$^{tg/+}$* and *Smn$^{-/-}$,Hung$^{tg/tg}$,Nestin-Cre$^{tg/+}$* mice were cross-bred from parents and generated in-house. Animal sex as a biological variable was not considered in the study design.

## Primary mouse motoneuron culture and viral transduction

Primary mouse motoneuron culture was applied as previously described[27,123,124]. Pregnant mice were euthanized by cervical dislocation, and E12.5 mouse embryos were isolated. Motoneurons were enriched via p75[NTR] antibody panning, transduced with lentiviral particles for 10 min at RT, and plated onto precoated polyornithine and laminin211/221 (Biolamina, LN211-0501, and LN221-0501) cell culture dishes. This muscle-specific laminin isoform induces the differentiation of axonal growth cones into presynaptic structures in cultured motoneurons[125–127]. Cells were grown in presence of 3.5 ng/ml BDNF for 6 days. For immunofluorescence, SmFISH, and ExM, motoneurons were plated onto glass coverslips, for Western blot and qRT-PCR onto 24-well plates, for Ca$^{2+}$ imaging on μ-dishes (Ibidi, 81156), and for lattice-SIM onto 8-well chambers with 1.5 high-performance cover glasses (Cellvis, C8-1.5H-N). Culturing of motoneurons in compartmentalized microfluidic chambers was performed as previously described[54]. In cultured motoneurons, growth cones were identified based on their characteristic morphology, including their position at the distal tip of axons, their size, and the presence of distinct filopodia-like structures. Additionally, growth cones were identified as regions devoid of DAPI staining, ensuring that they were not part of the neuronal soma or nuclei.

## R-Roscovitine stimulation and immunocytochemistry

For stimulation experiments, DIV6 motoneurons received a 5 μM R-Roscovitine (referred to as Roscovitine in the text) (Merck, R7772) pulse for 5 min at 37 °C using a hot plate. BDNF stimulation was carried out as previously described[123]. Briefly, motoneurons were deprived of BDNF overnight and then exposed to a 40 nM pulse of BDNF for 1 min. Following stimulation, cells were fixed with 4% Paraformaldehyde (PFA) (ThermoFisher Scientific, 28908) for 10 min at RT and permeabilized with 0.1% Triton X-100 for 5–10 min. Cells were incubated with block solution (2% BSA, 100 μg/ml saponin, and 0.25% sucrose in PBS) for 1 h at RT. Primary antibodies were diluted in block solution and incubated at 4 °C overnight. After 3 × washing with TBST, secondary antibodies diluted 1:500 in PBS were added and incubated for 1 h at RT. Coverslips were embedded in Aqua Poly/Mount (Polysciences, 18606-20). For immunostaining of hiPSC-derived motoneurons, cells were fixed with 4% PFA for 15 min at RT, subsequently permeabilized and blocked as described above. For β-actin (Actβ) immunostaining, motoneurons were first exposed to ice-cold methanol for 5 min at -20 °C and then permeabilized with 0.1% Triton X-100 for 5 min at RT. In control experiments, prior to as well as during Roscovitine stimulation, cells were treated with 10 μM nocodazole for 2 h, or 100 ng/ml anisomycin for 1 h to inhibit axonal transport and local translation, respectively. For mCLING labeling assay, motoneurons were first incubated with 0.2 nmol mCLING-ATTO 647 N (Synaptic Systems, 710006AT1) for 1 min followed by a depolarization step with 90 mM KCl for 7 min. Neurons were then fixed with 4% PFA, 0.2% glutaraldehyde for 20 min on ice followed by a 10 min incubation at RT. The fixation buffer was quenched in 100 mM glycine solution for 20 min at RT and cells were subsequently immunostained against Munc13-1 and Synapsin1/2. In no-pulse control group, cells were immediately fixed after 1 min incubation with mCLING and treated as described above. Following primary antibodies were used: rabbit polyclonal anti-Tau (Sigma-Aldrich, T6402, 1:1000), mouse monoclonal anti-α-Tubulin (Sigma-Aldrich, T5168, 1:1000), mouse monoclonal purified IgG anti-Bassoon (Synaptic Systems, 141011, 1:500), guinea pig polyclonal anti-serum anti-Piccolo (Synaptic systems, 142104, 1:500), rabbit polyclonal purified anti-RIM1/2 (Synaptic Systems, 140213, 1:500), rabbit polyclonal anti-Munc13-1 (Synaptic System, 126103, 1:500), guinea pig polyclonal purified anti-Ca$^{2+}$ channel N-type alpha-1B (Ca$_v$2.2) (Synaptic System, 152305, 1:250), goat polyclonal anti-ribosomal protein L8 (RPL8) (Sigma-Aldrich, SAB2500882, 1:500), mouse monoclonal rRNA (Y10B) antibody (ThermoFisher Scientific, MA1-16628), guinea pig

monoclonal recombinant IgG anti-Snap25 (Synaptic systems, 111308, 1:250), guinea pig polyclonal antiserum anti-Synapsin1/2 (Synaptic systems, 111308, 1:500), goat anti-Choline Acetyltransferase (Millipore, AB144P, 1:250), anti-TuJ1 (Neuromics, MO15013, 1:1000), and polyclonal goat anti-TrkB (Bio-Techne Sales Corp, AF1494, 1:500). Secondary antibodies are as followed: donkey anti-mouse IgG (H + L) (Alexa Fluor 488, Jackson ImmunoResearch, 715-545-150), donkey anti-rabbit IgG (H + L) AffiniPure (Alexa Fluor 488, Jackson ImmunoResearch, 711-545-152), donkey anti-rabbit IgG (H + L) AffiniPure (Cy3, Jackson ImmunoResearch, 711-165-152), donkey anti-guinea pig IgG (H + L) AffiniPure (Cy5, Jackson ImmunoResearch, 706-175-148), and donkey anti-goat IgG (H + L) AffiniPure (Alexa Fluor 647, Jackson ImmunoResearch, 705-605-003).

## Single-molecule fluorescence in situ hybridization (smFISH)

smFISH was conducted as previously described[27] and following the manufacturer's instructions (ThermoFisher Scientific). Motoneurons were fixed with paraformaldehyde lysine phosphate (PLP) buffer (4% PFA, 5.4% glucose, and 10 mM sodium metaperiodate, pH 7.4) for 10 min at RT and permeabilized with a supplied detergent solution for 4 min at RT. mRNAs were unmasked by proteinase K digestion, which was applied for 4 min at 1:8000 dilution. Hybridization probes specific to the *Munc13-1* mRNA coding region were diluted 1:100 in the hybridization buffer and incubated at 40 °C overnight. For the amplification of FISH signal, preamplifier, amplifier, and label probe oligonucleotides (diluted 1:25 in respective amplification buffers) were incubated each for 1 h at 40 °C. After the washing steps, cells were immunostained against Tau for visualization of the neurite boundaries.

## Immunohistochemistry

For immunofluorescence of NMJs, mice were euthanized at P5 or P10 by decapitation, and TVA muscles were collected in an extracellular physiological solution (135 mM NaCl, 12 mM NaHCO3, 5 mM KCl, 1 mM MgCl$_2$, 2 mM CaCl$_2$, 20 mM glucose). Muscles were fixed with 4% PFA at 4 °C for 90 min, incubated with 0.1 M glycine on a shaker for 30 min, and permeabilized with PBS-T (1% Triton X-100) twice for 5 min, twice for 10 min, and twice for 30 min. Muscles were then blocked with 5% BSA in PBS-T (0.1% Triton X-100) at RT for 3 h, and then with primary antibodies diluted in block solution for two nights at 4 °C on a shaker. Then, preparations were washed with PBS-T (0.1% Triton X-100) at RT 3 × 15 min on a shaker, and secondary antibodies along with α-Bungarotoxin (ThermoFisher Scientific, B13422, 1:1000) were incubated at RT for 1 h. After 3 × wash with PBS-T, preparations were rinsed in water and embedded using Aqua-Poly/Mount. Postsynaptic membranes in the NMJs were labeled with Alexa Fluor 488-conjugated α-Bungarotoxin, which binds to the α-subunits of nicotinic acetylcholine receptors (AChRs). For immunofluorescence staining of motoneurons within the spinal cord, naive spinal cords were isolated from P10 mice and fixed in 4% PFA overnight. L1-L2 segments of the spinal cord were embedded in warm 5% Agar and serial sections of 50 μm were cut on a Vibratome. Sections were first incubated in 0.1 M glycine for 15 min and blocked in 5% Donkey serum, 0.3% Triton X-100 at RT for 2 h. The following primary and secondary antibodies were used: guinea pig polyclonal anti-Synaptophysin1 (Synaptic Systems, 101004, 1:1000), rabbit polyclonal anti-Ca$^{2+}$ channel P/Q-type specific against the alpha-1A subunit (Ca$_v$2.1) (Synaptic Systems, 152203, 1:500), rabbit polyclonal anti-Munc13-1 (Synaptic System, 126103, 1:500), guinea pig polyclonal antiserum anti-Munc13-1 (Synaptic Systems, 126104, 1:500), guinea pig polyclonal anti-RIM1 (Synaptic systems, 140005, 1:500), chicken polyclonal anti-Neurofilament H (Merck, AB5539, 1:1000), goat anti-Choline Acetyltransferase (Millipore, AB144P, 1:250), donkey anti-rabbit IgG (H + L) AffiniPure (Cy3, Jackson ImmunoResearch, 711-165-152, 1:500), donkey anti-guinea pig IgG (H + L) AffiniPure (Cy5, Jackson ImmunoResearch, 706-175-148, 1:500), donkey anti-chicken IgY (H + L)

AffiniPure (Cy5, Jackson ImmunoResearch, 703-175-155, 1:500), and donkey anti-goat IgG (H + L) AffiniPure (Cy3, Jackson ImmunoResearch, 705-165-147, 1:500).

## Cloning and generation of Munc13-1 rescue constructs and virus production

For cloning of Munc13-1/UNC13A lentivirus rescue constructs, plasmids harboring the coding region (cDNA) of endogenous mouse Munc13-1 and human UNC13A were purchased from GenScript. The 3'UTR of endogenous mouse Munc13-1 was synthesized and purchased from GenScript. The 3'UTR of mouse and human SynPhy were amplified by PCR using cDNA from mouse or human motoneurons as template and PfuUltra II Fusion HotStart DNA Polymerase (Agilent, 600670). The coding regions of Munc13-1/UNC13A were fused to the 3'UTR of mouse/human *Synaptophysin* mRNAs or to the 3'UTR of mouse *Munc13-1* mRNAs using NEBuilder® HiFi DNA Assembly Cloning Kit (New England Biolabs, E5520S) and inserted into a lentivirus backbone vector with the Ubiquitin promotor. For RescueΔ3'UTR, only the coding region of Munc13-1 was inserted into the lentivirus backbone vector. For generation of Munc13-1 knockdown construct, shRNA-targeting mouse Munc13-1 was cloned into a pSIH-H1 vector, as previously described[27]. The sequences of the antisense oligo used for Munc13-1 shRNA cloning are as follows; Munc13-1: 5'-TCCCGTGTGAAACAAAGGT-3'. For knockdown of Smn, a previously described and validated shRNA lentiviral construct was used[54]. eF1-Cre expressing vector was a gift from Philip Tovote. The expression of all rescue constructs was validated in cultured wt motoneurons by Western blot and qRT-PCR. Lentiviruses and AVVs were packaged in HEK[293T] cells using TransIT-293 (Mirus, MIR2706) for transfection[128,129]. For lentivirus packaging, pCMV-VSVG and pCMVΔR8.91 helper plasmids, and for AAV packaging, Rep/Capin, pAAV-mGly, and pHGTI-adeno1 AVV helper plasmids were used. Viral supernatants were harvested by ultracentrifugation at 60–72 h post-transfection. Virus titer was determined in NSC[34] cells (Cedarlane, cat. no. CLU140), using standard methods with serial dilutions.

## Generation of Cre/loxP conditional Munc13-1 rescue mouse model

The coding sequence of endogenous mouse Munc13-1, the 3'UTR of endogenous mouse SynPhy, and the SV40pA sequence were first amplified by PCR using available plasmids as templates (see the cloning section). PCR products were then assembled into one fragment and inserted into an expression vector using NEBuilder® HiFi DNA Assembly Cloning Kit. Next, the assembled fragments were excised from the expression vector by XhoI restriction enzymes and inserted into a backbone vector harboring the CAG-loxP-Stop-loxP cassette. The resulting cassette including CAG-loxP-Stop-loxP-Munc13-1+SynPhy3'UTR-CV40pA was excised from this vector by SalI and ligated into a SalI linearized ROSA26 donor vector. The Cre-dependent expression of Munc13-1 was validated by qRT-PCR and Western blot in HEK[293] cells, which were transfected with the vector expressing the targeting cassette and an eF1-Cre expressing vector. For the generation of $R26Unc13-1^{tg/+}$ knock-in mice, the targeting cassette was inserted into the ROSA26 locus through CRISPR-Cas9 technology at Czech Centre for Phenogenomics in Prague, Czech Republic (https://www.phenogenomics.cz). Three founders were obtained, and the transgenic cassette was validated by sequencing as well as genotyping PCR. The Cre-dependent expression of the transgenic Munc13-1 allele was verified by qRT-PCR using a Nestin-Cre driver line (Fig. 7b), as well as by Western blot in $Munc13-1^{-/-}$ mice (Fig. 7c). Following primers were used for genotyping of R26Unc13-1 knock-in allele: ROSA26ext-forward: 5'-TGCCATGAGTCAAGCCAGTC-3', SynPhy3'UTR-reverse 5'-CTCTGCTGTGTCTGTGACGT-3', and for ROSA26 wt allele: ROSA26-reverse: 5'-GGCTCAGTTGGGCTGTTTTG-3'.

## Ca$^{2+}$ imaging and data quantification

Ca$^{2+}$ imaging was performed using the calcium indicator Oregon Green™ 488 BAPTA-1, AM, cell-permeant (ThermoFisher Scientific, O6807). Calcium indicator was dissolved in Pluronic F-127/DMSO in an ultrasonic bath for 2 min to prepare a 5 mM stock solution. Motoneurons were first washed twice with prewarmed Ca$^{2+}$ imaging buffer (135 mM NaCl, 6 mM KCl, 1 mM MgCl$_2$, 1 mM CaCl$_2$, 10 mM HEPES, and 5.5 mM glucose) and incubated with 5 μM Ca$^{2+}$ indicator diluted in the Ca$^{2+}$ imaging buffer for 15 min at 37 °C in a CO$_2$ incubator. Cells were washed again twice with Ca$^{2+}$ imaging buffer and imaged in 2 ml of Ca$^{2+}$ imaging buffer in the presence of 3.5 ng/ml BDNF. For time-lapse imaging, a TE2000 Nikon inverted epifluorescence microscope was used that was equipped with a 60× 1.4-NA objective, a perfect focus system, Orca Flash 4.0 V2 camera (Hamamatsu Photonics), an LED fluorescence light for excitation at 470 nm, and Nikon Element image software. Cells were imaged at 37 °C in the presence of 5% CO$_2$ using a TOKAI HIT CO, LTD heated stage chamber. Cells were imaged at 500 ms intervals over a total period of 7 min for spontaneous Ca$^{2+}$ spikes and over 2 min for KCl pulse experiments. 16-bit images of 1.024 × 1.024-pixel resolution were acquired with a 2 × 2 binning. For pulse experiments, 10 μl of 90 mM KCl was applied to the imaging cell at 1 min post-imaging. For the quantification of Ca$^{2+}$ spikes, first, a region of interest (ROI) was defined within growth cones using Fiji. Next, intensity values were generated from all time-lapse frames using dynamic Z-axis profile. For spontaneous Ca$^{2+}$ spikes, the average of the first 10 frames before a Ca$^{2+}$ spike was considered F0, and in pulse experiments, the average of the first 20 frames immediately before KCl application was considered F0. For data normalization, all intensity values were divided to F0 and plotted (F/F0). BAR Plugin of Fiji was used for counting the Ca$^{2+}$ spikes. In control experiments, motoneurons were treated with 30 μM ω-conotoxin (ω-CTX-MVIIC), a selective blocker of Ca$_v$2.1/Ca$_v$2.2, 1 hour prior to Ca$^{2+}$ imaging. No spontaneous Ca$^{2+}$ spikes were detected.

## Synaptic vesicle recycling assay

To investigate the synaptic vesicle recycling in motoneurons, a Cy3-conjugated monoclonal antibody directed against the intravesicular domain of Synaptotagmin1 (Synaptic Systems, 105103C3) was used[75]. DIV5 cultured motoneurons were incubated with the antibody overnight (diluted 1:400 in the cell culture medium) in a CO$_2$ incubator. On the next day, cells were washed twice with pre-warmed PBS and fixed with 4% PFA for 5 min at RT. Neurons were imaged using a standard confocal microscope. In control experiments, cells were cultured for 6 days in presence of 30 nM CTX as well as 60 nM TTX, a selective blocker of voltage-gated Na$^+$ channels, and fed with Synaptotagmin1 antibody afterward.

## Puromycin proximity ligation assay (Puro-PLA)

Proximity ligation assay was conducted as previously described with minor modifications[69]. Shortly, 10 μg/ml puromycin (Merck, 540222-25MG) and 100 μg/ml cycloheximide (Merck, 01810-1 G) were added to the cells and incubated for 5 min at 37 °C. The puromycylation reaction was stopped through washing with PBS-MC (1 × PBS pH 7.4, 1 mM MgCl$_2$, 0.1 mM CaCl$_2$) and cells were fixed with 4% PFA, 4% sucrose in PBS-MC buffer for 10 min at RT. Cells were then permeabilized with 0.2% Triton X-100 for 10 min at RT and incubated with Duolink blocking solution for 1 h at 37 °C. PLA assay was conducted using Duolink® In Situ Detection Reagents Orange (Sigma-Aldrich, DUO92007). Primary antibodies, rabbit polyclonal anti-Munc13-1 (Synaptic System, 126103, 1:500), rabbit polyclonal anti-Synaptobrevin2 (VAMP2) (Synaptic System, 104008, 1:500), and mouse monoclonal anti-puromycin (Merck Millipore, MABE343, 1:1000) were diluted in the Duolink antibody diluent and incubated with cells for 1 h at RT. Cells were washed first several times and then

2 × 5 min with 1 × Wash Buffer A (Sigma-Aldrich, DUO82049) at RT. PLA probes; anti-mouse MINUS (DUO92004) and anti-rabbit PLUS (DUO92002); were diluted 1:50 in Duolink antibody diluent and incubated with cells for 1 h at 37 °C. After multiple washing steps with 1 × Wash Buffer A, ligase (diluted 1:40 in ligation buffer) was added to the cells and incubated for 30 min at 37 °C. Cells were washed again several times with 1 × Wash Buffer A and incubated with polymerase (diluted 1:80 in amplification buffer) for 100 min at 37 °C. The amplification step was stopped by 2 × 10 min wash at RT with 1 × Wash Buffer B (Sigma-Aldrich, DUO82049). Finally, cells were washed for 1 min with 0.01 × Wash Buffer B and mounted using Duolink mounting medium (Sigma-Aldrich, DUO82040). All incubation steps at 37 °C were carried out in a dark/humid chamber using a dry incubator (Binder BD 23). In the control PLA experiments, motoneurons were incubated with either Munc13-1 or puromycin antibodies, followed by TrkB immunostaining. In additional control experiments, motoneurons were pretreated with 100 ng/ml anisomycin for 1 h before performing PLA assay, followed by TrkB immunostaining. For TrkB staining, the permeabilization and blocking steps were omitted.

### Cortical synaptosome fractionation and RNA immunoprecipitation (RNA-IP)

Crude synaptosome fractions were prepared from cortices of P5 control and Smn KO mice by sequential centrifugation steps in a sucrose buffer (0.32 M sucrose, 5 mM HEPES, 1 × protease inhibitor cocktail (Roche)) as described earlier[130]. In brief, P5 littermates were euthanized by decapitation. Cortices were isolated and mechanically homogenized in 500 µl cold lysis buffer and centrifuged at 1000 × g for 10 min at 4 °C. Pellets (P1) containing the nuclei were discarded and supernatants (S1) were centrifuged at 12000 × g for 20 min at 4 °C. Resulting supernatants (S2) containing the light membrane fraction and soluble enzymes were discarded and pellets (P2) containing crude synaptosomes were resuspended in 700 µl IP buffer (20 mM Tris pH 7.5, 2 mM MgCl₂, 150 mM KCl, 0.1% Nonidet P-40, 1 × protease inhibitor cocktail, 100 µg/ml cycloheximide) and incubated on ice for 15 min. To prevent the disassembly of ribosomal 80S complexes with their bound transcripts, 100 µg/ml cycloheximide was added into the fractionation buffer as well as into the IP buffer for all the sequential IP steps. For the pulldown of 80S ribosomal complexes, a mouse monoclonal rRNA (Y10B) antibody (ThermoFisher Scientific, MA1-16628) and normal mouse IgG control (Santa Cruz Biotechnology, sc-2025) were used. First, 1 µg Y10B or IgG control antibodies and 10 µl protein G magnetic dynabeads (ThermoFisher Scientific, 10003D) were added to 100 µl IP buffer and incubated with rotation for 1 h at RT. Synaptosome fractions were then added into pre-washed magnetic protein G/antibody beads and incubated with rotation for 2 h at 4 °C. Resulting immunocomplexes were washed twice for 10 min with rotation at 4 °C. For qRT-PCR, RNAs were eluted from magnetic beads with ethanol precipitation and purified using PicoPure™ RNA Isolation Kit (ThermoFisher Scientific, KIT0204) following the manufacturer's instructions. Purified RNA was resuspended in 20 µl RNase-free water and 10 µl was reverse transcribed with random primers using RevertAid First Strand cDNA Synthesis Kit (ThermoFisher Scientific, K1621). Relative binding of *Munc13-1* transcripts to ribosomes as well as 18srRNA levels were determined by qRT-PCR. To determine ribosome levels in the input and IP fractions, proteins were eluted with 1 × Laemmli buffer (125 mM Tris, pH 6.8, 10% SDS, 50% glycerol, 25% β-mercaptoethanol, and 0.2% bromophenol blue) and subsequently analyzed by Western blot. In control experiments, equal volumes of lysates from all centrifugation steps (P1, S1, S2, and P2) were loaded onto a 12% SDS-PAGE gel and analyzed by Western blot using antibodies against Synapsin I and Histone H3.

### RNA extraction and quantitative RT-PCR (qRT-PCR)

For RNA extraction from microfluidic chambers, PicoPure™ RNA Isolation Kit (ThermoFisher Scientific, KIT0204) was used. qRT-PCR was performed on a LightCycler 1.5 thermal cycler (Roche) using Luminaris HiGreen qPCR Master Mix (ThermoFisher Scientific, K0992). The relative expression of target genes was measured according to the ΔΔCt method. Gapdh was used as internal control and for data normalization. The following equation was used to determine the relative number of *Munc13-1* transcripts bound to ribosomes in RNA-IP experiments using 18srRNA as reference:

$$\text{Ratio} = \left(E_{target}\right)^{-(IP_{KO} - input_{KO}) - (IP_{wt} - input_{wt})} / \left(E_{ref}\right)^{-(IP_{KO} - input_{KO}) - (IP_{wt} - input_{wt})}$$

Following primers were used for qRT-PCR: mouse Gapdh (forward) 5'-AACTCCCACTCTTCCACCTTC-3' and (reverse) 5'-GGTCCAGGGTTTCTTACTCCTT-3', mouse Munc13-1 coding region (forward) 5'-CACCACGCCCACCTACTGCTA-3' and (reverse) 5'-TTGCGCTCGCGGATCT-3', mouse 18srRNA (forward) 5'-CGCGGTTCTATTTTGTTGGT-3' and (reverse) 5'-AGTCGGCATCGTTTATGGTC-3', mouse Snap25 (forward) 5'-CCTAGGAAAATTCTGCGGGC-3' and (reverse) 5'-CTGCTCCAGGTTCTcATCCA-3', mouse VAMP2 (forward) 5'-GTGGATGAGGTGGTGGACAT-3' and (reverse) 5'-CCACCAGTATTTGCGCTTGA-3', mouse SynPhy (forward) 5'-TGGCCACCTACATCTTCCTG-3' and (reverse) 5'-TCCCTCAGTTCCTTGCATGT-3', mouse Synaptotagmin I (forward) 5'-GGTGACATCTGCTTCTCCCT-3' and (reverse) 5'-TGGATTTGCTCGAACGGAAC-3', human GAPDH (forward) 5'-GCAAATTCCATGGCACC-3' and (reverse) 5'-CGCCAGTGGACTCCACGAC-3', human UNC13A (forward) 5'- GGACGTGTGGTACAACCTGG-3' and (reverse) 5'- GTGTACTGGACATGGTACGGG-3'.

### Western blotting

For Western blotting with primary mouse motoneurons or human iPSC-derived motoneurons, 200,000 cells were plated and grown for 7 days and 25 days, respectively. After a short wash with pre-warmed PBS, cells were lysed in 1 × Laemmli buffer, lysates were boiled at 99 °C for 5 min and centrifuged briefly. Protein extracts were loaded onto 4-12% gradient SDS-PAGE gels and blotted onto PVDF membranes. For Western blot with hiPSCs, 300,000 cells were plated on Matrigel, until 90% confluent. Cells were lysed in RIPA buffer (10 mM Tris-HCl, pH 8.0, 1 mM EDTA, 0.5 mM EGTA, 1% Triton X-100, 0.1% Sodium Deoxycholate, 0.1% SDS, 140 mM NaCl), protein concentration was determined using Pierce BCA Protein Assay Kit (ThermoFisher Scientific, A55860) and 20 µg total protein was loaded into 10% SDS gels. For Western blot with the IP obtained from synaptosome fractions, equal volumes of protein extracts from input, IP, and IgG fractions were loaded. Calnexin was used as loading control. Primary antibodies were incubated overnight at 4 °C, washed the next day and secondary antibodies were incubated for 1 h at RT. Membranes were developed using ECL systems (GE Healthcare). The following antibodies were used: rabbit polyclonal anti-Calnexin (Enzo Life Sciences, ADI-SPA-860-F, 1:6000), rabbit polyclonal anti-Histone H3 (Abcam, ab1791, 1:10000), guinea pig polyclonal antiserum anti-Synapsin1/2 (Synaptic systems, 111308, 1:500), goat polyclonal anti-ribosomal protein L8 (Sigma-Aldrich, SAB2500882, 1:5000), rabbit polyclonal anti-Munc13-1 (Synaptic Systems, 126103, 1:4000), mouse monoclonal anti-SMN (BD Biosciences, 610646, 1:5000), rabbit polyclonal anti-Cre (Merck, 69050, 1:5000), mouse monoclonal anti-β-actin (GeneTex, GTX26276, 1:5000), peroxidase AffiniPure donkey anti-goat IgG (H + L) (Biozol, 705-035-003, 1:10000), peroxidase AffiniPure donkey anti-mouse IgG (H + L) (Biozol, 715-035-151, 1:10000), and peroxidase AffiniPure goat anti-rabbit IgG (H + L) (Biozol, 111-035-144, 1:10000).

### Motor assessments

All motor tests were carried out with P10 mice. For the righting reflex, mice were placed on their back and were hold for 5 s. The time that they took to turn to the prone position was recorded. An average of

three attempts with a maximum of 30 s designated the "time delay to right" score. For the grip strength test of forelimbs, mice were placed on a thin glass tube with their forelimbs and the time it took for them to fall was measured. For the hind limb clasping test, mice were put on a 50 ml falcon and the splay was recorded when the animals climbed into the falcon. The observed splay was scored between 0 and 4. A score of 4 was assigned to a healthy splay of both hind limbs. A score of 3 was given to a weak splay of both hind limbs. A score of 2 was assigned to a clasping. A score of 1 was assigned when no splay was observed, and a score of 0 received a pub that crossed both hind limbs over each other. Investigators who carried out the motor tests were blinded to the animal genotypes. No animals were excluded from motor assessments. Two litters, out of 29, were excluded from the survival assessment, wherein the female stopped to suckle the litter leading to the death of all pubs. In addition, pubs, which died at P0, were not considered for the survival assessment.

### Lattice-SIM

For lattice-SIM, DIV6 motoneurons were first stimulated with 5 µM Roscovitine for 5 min and then fixed with 4% PFA for 10 min at RT. Following three washing steps with PBS, cells were permeabilized for 10 min at RT with 0.1% Triton X-100 in PBS. Afterwards, cells were blocked for 1 h at RT with 5% BSA in PBS and incubated overnight at 4 °C with following primary antibodies: polyclonal rabbit anti-Munc13-1 (Synaptic Systems, 126103), monoclonal guinea pig anti-Snap25 (Synaptic Systems, 111308), polyclonal goat anti-Ribosomal Protein L8 (RPL8) (Sigma Aldrich, SAB2500882), and guinea pig polyclonal antiserum anti-Synapsin1/2 (Synaptic systems, 111308, 1:500) diluted 1:250 in the blocking solution. On the next day, cells were washed again trice with PBS. Thereafter the following secondary antibodies were incubated for 1 h at RT in blocking solution: Alexa Fluor 647 donkey F(ab')2 anti-rabbit IgG (H + L) (Abcam, ab181347, 1:300), Alexa Fluor 488 donkey anti guinea pig (Dianova, 140967, 1:300) and CF568 donkey anti-goat IgG (H + L) (Biotium, 20106, 1:500). The samples were post-fixed for 15 min at RT with 4% PFA and washed 3 × with PBS.

### Expansion microscopy (ExM)

Wt motoneurons cultured on glass coverslips were first stimulated with 5 µM Roscovitine for 5 min. In unstimulated control group, cells were treated with neurobasal medium without Roscovitine. Immediately after stimulation, cells were incubated in a humidified chamber for 5 h at 37 °C in 0.7% formaldehyde and 1% acrylamide diluted in PBS. Thereafter, coverslips with cultured motoneurons were transferred upside down to a 60 µl droplet of the TREx monomer solution[72] on parafilm in a humidified chamber on ice. The monomer solution was incubated for at least 2 h at RT for polymerization. Afterwards, gels were incubated for 1 h at 95 °C in denaturation buffer (200 mM SDS, 200 mM NaCl and 50 mM Tris, pH 6.8) and washed 2 × with PBS. Gels were then incubated overnight at RT in 5% BSA in PBS with a 1:200 dilution of polyclonal rabbit anti-Munc13-1 (Synaptic Systems, 126103). On the next day, after 3 × 15 min washing steps with PBST (0.1% Tween-20), gels were incubated for 3 h at 37 °C with Alexa Fluor 647 donkey F(ab')2 anti-rabbit IgG (H + L) (Abcam, ab181347, 1:100) secondary antibody diluted in 5% BSA in PBS. Gels were washed again 3 × for 15 min with PBST (0.1% Tween-20) and incubated for 1 h at RT with Alexa Fluor 488 NHS diluted in 100 mM NaHCO3 (ThermoFisher, A20000, 20 µg/ml). The nuclei were stained by adding Hoechst 34580 (ThermoFisher, H21486, 1:500) to the pan staining for 25 min at RT. For expansion, gels were placed in petri dishes. Water was exchanged every 20 min until the gel reached its final expansion factor. For imaging with lattice-SIM, gels were transferred to Lab-Tec™ chambers coated with PDL.

### Differentiation of human induced pluripotent stem cells into motoneurons

The hiPSC lines used in this study were purchased from Cedars-Sinai Biomanufacturing Center and are as followed: CS84iSMA (type I), CS77iSMA (type I), CS13iSMA (type II), CS83iCTR (healthy control), and CS88iCTR (healthy control). The cell lines were originally generated from biospecimens donated to the NIGMS Human Genetic Cell Repository (Coriell Institute for Medical Research) under informed consent (NIGMS Informed Consent Form, Form 1401-63 Rev E – 072015). Donors provided consent for use of their samples in research, including derivation of hiPSCs. All procedures for collection and use of biospecimens were approved and are overseen by the Coriell Institutional Review Board (Coriell IRB; Protocol #R116; contact: NIGMS@coriell.org, Tel. 800-752-3805). Motoneurons were differentiated according to Reinhardt et al.[131], with few modifications as described earlier[132]. Briefly, hiPSCs were grown on Matrigel-coated (Corning, 356234, 1:100) dishes and expanded in mTeSR Plus medium (Stemcell Technologies, 05825). For the induction of Embryoid Bodies (EBs), hiPSCs were seeded into non-coated low adherent 12-well plates (Greiner, M9187) and grown in mTeSR Plus medium supplemented with small molecules; SB431542 (AdooQ BioScience, A10826-50, 10 µM), dorsomorphin homolog 1 (DMH1) (R&D Systems, 4126, 1 µM), CHIR99021 (Cayman Chemical Company, 13122, 3 µM), and Purmorphamine (PMA) (Cayman Chemical Company, 10009634, 0.5 µM). On day 2, mTeSR Plus medium was replaced with neuronal medium (Neurobasal medium (Gibco, 21103049), Dulbecco's modified Eagle's medium F-12 (DMEM/F-12) (Gibco, 21331046), N-2 Supplement (Gibco, 17502048), Penicillin/Streptomycin/Glutamax (Gibco, #10378016, 100 µg/mL)) supplemented with the same small molecules as above. On day 4, medium was replaced with expansion medium (neuronal medium supplemented with 3 µM CHIR99021, 0.5 µM PMA, and 150 µM Ascorbic acid (AA) (Sigma, A92902)). For induction of Neuronal Progenitor Cells (NPCs), EBs were collected from the suspension on the day 6, dissociated and plated on Matrigel-coated dishes. In order to achieve pure NPC cultures, cells were split for at least 20 passages once a week using Accutase (Thermo Fisher, 07920). For NPC differentiation into motoneurons, cells were seeded on Matrigel-coated dishes and expanded for two days in neuronal medium supplemented with 1 µM PMA. On day 2, medium was exchanged with neuronal medium supplemented with 1 µM PMA and 1 µM Retinoic acid (Stemcell Technologies, 72264). For motoneuron differentiation, on day 9, NPCs were harvested and plated on polyornithine/laminin211/221-coated coverslips or microfluidic chambers. Motoneurons were differentiated for 25 days in neuronal medium supplemented with 5 ng/mL glia-derived neurotrophic factor (GDNF) (Alomone Labs, G-240), 5 ng/mL brain-derived neurotrophic factor (BDNF) (Institute of Clinical Neurobiology, University Hospital of Wuerzburg, Germany), and 500 µM dibutyryl-cAMP (dbcAMP) (Stemcell Technologies, 73886). For all steps, medium was exchanged every other day. The culturing and differentiation of hiPSCs into motoneurons in microfluidic chambers were performed as previously described[54] with the following modifications. In these chambers, 5 ng/mL BDNF and 5 ng/mL GDNF were utilized in the somatodendritic compartment, while the axonal compartments received 20 ng/mL BDNF, 20 ng/mL GDNF, and 20 ng/mL Ciliary neurotrophic factor (CNTF) were used for the axonal compartments. Additionally, cAMP was applied to both compartments at a concentration of 500 µM.

### Image acquisition and processing

Confocal images were acquired as 16-bit images with 800 × 800–pixel resolution with an Olympus Fluoview 1000 microscope equipped with a 60× 1.35-NA oil objective. For cultured motoneurons, confocal images of a single z-stack were taken. For NMJs, confocal images of 6-8 z-stacks of 0.5 µm were taken and maximum projection images were

shown for representative images. For Lattice-SIM and expansion microscopy, images were acquired using an ELYRA 7 SIM Zeiss equipped with either a Plan-Apochromat 63 × NA-1.4 oil objective (for unexpanded samples) or a C-Apochromat 63 × 1.2-NA water immersion objective (for expanded samples) and 405 nm diode (50 mW), 488 nm OPSL (500 mW), 561 nm OPSL (500 mW) and 642 nm diode (500 mW) excitation lasers. The laser power was adjusted between 2 and 5% with an integration time of 200 ms. For lattice-SIM, five z-stack images of 110 nm were taken, and maximum projection images were shown as representative images. For lattice-SIM with expanded samples (ExM), 20-60 z-stack images of 124 nm were taken, and single stack images were shown as representative images. The acquired 16-bit raw images were processed with a commercial software package from Zeiss (ZEN 3.0 SR FP2 black) to reconstruct super-resolution images. For Airyscan microscopy, mounted mouse TVA tissue slices were imaged with a LSM 900 with Airyscan 2 (Zeiss) in SR imaging mode utilizing a Plan-Apochromat 63 × 1.35-NA oil immersion objective (Zeiss). The excitation wavelengths and filter settings for the respective dyes suitable for the samples were selected via the integrated dye presets in the ZEN 2 blue software (Zeiss, version 3.5). Each stack was acquired with a 0.15 μm distance to the next slice and processed in the standard strength mode of 3D Airyscan processing. On average, 5 to 7 stacks were acquired per NMJ. For proper overlay of the different channels, a channel alignment was performed on the reconstructed images using fiducial markers (ThermoFisher Scientific, TetraSpeck™ Microspheres, 0.2 μm, fluorescent blue/green/orange/dark red, T7280). Images were processed and analyzed with Fiji. For better visibility, linear contrast enhancement was implemented to all representative images using Adobe Photoshop 24.2.0.

## Data analysis

For quantification of immunofluorescence signal in growth cones and somata as well as quantification of smFISH in somata, mean gray values of unprocessed raw images were measured using Fiji following background subtraction. For the quantification of immunofluorescence signal in NMJs, first average projections of multiple z-stacks images were created, and then mean gray values were measured within Syn-Phy positive area. Signal intensities of SynPhy were comparable between Smn KO and control and were therefore used for normalization of Munc13-1 intensities. All intensities were normalized to the average intensity of the control group within the same experiment. To assess muscle innervation, at least 450 randomly selected endplates were quantified per muscle sample per mouse for each genotype. Endplates labeled with AChRs that showed no presynaptic synaptophysin/NFH coverage were considered as fully denervated NMJs. For stimulation experiments, all signal intensities were normalized to the mean intensities of the non-stimulated corresponding genotype. For quantification of the PLA signal, the number of punctae was calculated within axonal growth cones or cell bodies using Analyze Particle Plugin of Fiji, and the number of punctae was normalized to μm² growth cone area. For quantification of the smFISH data, the punctae were manually counted in the axon, and the number of punctae was normalized to μm axon length. For colocalization analysis of lattice-SIM data, single optical sections of raw 16-bit images were used. Regions of interest (ROIs) were defined within axonal growth cones, and the Pearson R-value was calculated using Fiji. For expansion microscopy, single optical sections of 16-bit raw images were analyzed with Fiji. To minimize potential bias in the identification and quantification of Munc13-1 ring-like structures in expansion specimens, we employed several strategies. First, ROIs were defined using objective structural criteria such as size, intensity, and their characteristic ring-like structure rather than subjective selection. These criteria were validated using our lattice-SIM data, as shown in Fig. 3c,d, and Supplementary Fig. 3f-j. The number of Munc13-1 nanoassemblies within each ring-like cluster was determined manually, with an average of 6-8 nanoassemblies per

cluster. The center-to-center distance between neighboring nanoassemblies was measured using the freehand line tool in Fiji. Although ring-like arrangements were identified based on the Munc13-1 signal, we used a consistent thresholding approach across all images, applied uniformly using Fiji. Two independent observers performed manual counting to ensure reproducibility. We acknowledge the limitation that Munc13-1 was both the marker of interest and the basis for structural selection. All immunostaining experiments, including smFISH and PLA, except for super-resolution microscopy, were carried out and analyzed blindly. In Fig. 2e, data corresponding to control and Smn KO are reproduced from Supplementary Fig. 1a. In Fig. 2f, g, data corresponding to control and Smn KO are reproduced from Fig. 1d, e.

## Statistical analysis

GraphPad Prism 10 was used for graph illustration and statistical analyses. Data are depicted as bar graphs or scatter dot plots, with error bars representing mean ± SEM, or as violin plots, with median indicated as dashed lines. To determine the statistical significance between two groups Mann Whitney t-test, and between multiple groups Mann Whitney t-test or one-way analysis of variance (ANOVA) Kruskal-Wallis test with Dunn's Multiple Comparison post-hoc test were used. Statistical tests were performed with pooled data from at least three independent biological replicates, except otherwise stated.

## Ethics statement

All experimental procedures involving mice were conducted in accordance with the German federal regulations on animal protection and the guidelines of the Association for Assessment and Accreditation of Laboratory Animal Care. This study was approved under protocol number 55.2.2-2532.2-924-14. The work was approved by and carried out under the supervision of the local veterinary authority. Laboratory mice were housed in the animal facility of the Institute of Clinical Neurobiology at the University Hospital Wuerzburg, a certified local facility compliant with all relevant regulations and standards. Mice were housed under controlled conditions on a 12 h light/12 h dark cycle at 20–22 °C and 55–65% relative humidity, with ad libitum access to food and water.

## Reporting summary

Further information on research design is available in the Nature Portfolio Reporting Summary linked to this article.

## Data availability

All datasets are available in the publicly accessible repository "Figshare" under the following accession code: (https://doi.org/10.6084/m9.figshare.29971474). Source data are provided with this paper.

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

## Acknowledgements

We thank Regine Sendtner and Franziska Wagner for support with animal husbandry and performance of behavioral motor tests. Michaela Kessler for support with stem cells and immunohistochemistry. Hildegard Troll for virus production, Philip Tovote for providing eF1-Cre expression vector, Saeede Salehi, Patrick Lueningschroer, Lena Saal-Bauernschubert, Janna Eilts, Abdolhossein Zare, and Neha Jadav-Giridhar for their intellectual contributions and technical assistance. We also thank Annamaria Musti for her constructive input on the manuscript preparation. We are grateful to Xinsheng Nan and Yves-Alain Barde (Cardiff University) for providing plasmids with CAG-loxP-Stop-loxP cassette and ROSA26 donor, as well as Noa Lipstein (Leibniz-Institute, Berlin) and Nils Brose (Max-Planck-Institute, Göttingen) for providing a rat Munc13-1-gfp expressing plasmid that we used for our preliminary experiments. We also thank Petr Kašpárek (Institute of Molecular Genetics, Prague) for generation of the Cre/loxP conditional Munc13-1 rescue mouse model. This work was funded by grants from "Deutsche Forschungsgemeinschaft (DFG)", SE 697/7-1 to M. Sendtner, and SA 829/17-1 and SA829/19-1 to M. Sauer, as well as by a grant from Pico-Quant to M. Sendtner and the European Research Council (ERC) under the European Union's Horizon 2020 research and innovation program (grant agreement No 835102) to M. Sauer.

## Author contributions

M.M. and M. Sendtner. conceived the project, designed and conducted the experiments, and prepared the manuscript. J.W. and M. Sauer. performed lattice-SIM, Airyscan and ExM, C.C.D. helped with motoneuron culture and immunohistochemistry. M.N. performed Ca$^{2+}$ imaging and smFISH. M.B. helped with human motoneuron cultures. S.J. helped with mice crossing.

## Funding

## Competing interests

M. M. and M. Sendtner are inventors on two patents that have been applied for by the Julius-Maximilians-Universität Würzburg and were recently published as WO 2025026896A1 and WO 2025026772A1. The patent application with the number WO 2025026772A1 was designed to protect the use of the Munc13-1 hybrid molecule described in this manuscript for the treatment of Spinal Muscular Atrophy and other motoneuron disorders. The patent application with the number WO 2025026896A1 refers to the local presynaptic modulation of Munc13-1 expression for the treatment of motoneuron disorders. The remaining authors declare no competing interests.
