## [Transparent Peer Review file · Nature Communications]

Munc13-1 Restoration Mitigates Presynaptic Pathology in Spinal Muscular Atrophy

Corresponding Author: Dr Mehri Moradi

Version 0:

Reviewer comments:

Reviewer #1

(Remarks to the Author)

The paper by Moradi et al describes a role for localization and translation of Munc13-1 mRNA in spinal motor atrophy. The authors mostly use a mouse model for SMA and study neurons in vitro or in vivo. They note a reduction in the localization of Munc-13 mRNA in the mouse model – which they attribute to the loss of the RNA-binding protein HnRNPR which binds to the 3'UTR of Munc-13 mRNA and localizes it to axon terminals (previous work from the Sendtner lab). They then carry out a series of rescue experiments- in which the localization of Munc-13 is restored by expression a version with a different 3'UTR. They show, with a variety of methods, that this restores Munc-13 levels and restores compromised presynaptic function and also some of the behavioral phenotypes in the mutant mice. Overall, this is a very nice study- my specific comments and suggestions are listed below.

1. Re: the qRT-PCR analysis of RNA from the axonal compartment of the SMN KO, what about other transcripts? It would be nice to see a few that are not affected- and it is also desirable to see the real (unnormalized) expression level of the Munc13 mRNA in the control mice.
2. minor point: the authors nicely state at the beginning of the paper that they will refer to the SMN-Hung transgenic mouse as KO- but then they do not do this. It is very challenging for readers and reviewers to digest the very small differences between +/- and -/- superscripted in all of the figures.
3. I find it surprising that the Munc-13 protein content is unaltered in the SMN cultured motoneurons and also find the assertion that the “somata represent the main part of these neurons”... The volume of the axon is not insignificant.
4. For the Puro-PLA experiments- is the antibody epitope for Munc13 N-terminal? What does the puro-PLA signal in the cell body look like and how far is the axonal terminal studied from the cell body? Is the puro-PLA signal blocked by anisomycin?
5. minor point: “dots”- the authors might want to adopt the term “punctae” or “puncta” instead.
6. In the synaptosome experiments, are the synaptosomes free of nuclear contaminants?
7. re: the levels of ribosomes in synaptic fractions from the SMN KO animals- the data shown in supplementary figure 2c are not convincing.
8. Re: fig 3b. The KO delta-3'UTR data- in particular in the images shown, do not seem to fit with the authors idea. It looks like there is indeed a rescue.
9. Re, the anisomycin experiments in figure 3 (supplement). The statement that begins “these data indicate that Munc13-1 local translation is required for the de novo formation of SV release sites” is not supported by these experiments. The anisomycin experiments do not specifically block synthesis of Munc-13 and there is nothing about the anisomycin application that addresses whether the translation is local.
10. the co-localization of Munc-13 and RPL8. This experiment does not convincingly show that ribosomes are colocalized with Munc-13. There are 79 ribosomal proteins and there is the possibility of extra-ribosomal functions. Perhaps a better

experiment would be to use the y10B antibody which recognizes ribosomal RNA. The discussion section on this point is also over-stated.

11. Re: Fig 5 a,b. This partial rescue by the delta mutant is not tested statistically. Are the rescue and delta 3'UTR really different? The analysis in B does not appear to support this statement. In addition, in figure 6f-h, the results with the delta 3'UTR mutant are not fully explained or interpreted.

12. Re fig 8. Can the authors provide reasons why the rescue delays death, but does not fully restore to WT lifetimes?

Reviewer #2

(Remarks to the Author)

This study by Moradi and colleagues investigates the expression and levels of Munc13-1 in spinal muscular atrophy (SMA) and how modulating Munc13-1 expression may improve degenerative phenotypes in this disease. They demonstrate that the levels of Munc13-1 and other presynaptic proteins are reduced in the axons of neurons cultured from an SMA mouse model and in axons from neurons derived from patients with SMA. They also show that a portion of Munc13-1 mRNA is present in, and locally translated at, axons, and that this process is impaired in the SMA mouse model and in human neurons. Additionally, they show that the 3'UTR region of Munc13-1 mRNA is important for targeting this transcript to axons, and that replacing it with the 3'UTR of a different gene (Synaptophysin) restores the levels of this mRNA in the axons of the SMA mouse model. Finally, they demonstrate that increasing the levels of Munc13-1 decreases signs of axonal degeneration, improves motor coordination, and leads to a slight increase in the survival of the mutant mice.

Addressing the relationship between synaptic proteins/functions and degenerative disorders is timely. In that context, this study is somewhat relevant. However, the current manuscript is problematic, mainly because the experiments corresponding to the assembly and ultrastructure of release sites and neuronal excitability (Figures 3, 4, 5 and 6) are not well designed, executed or conceptualized. In consequence, no conclusions can be reliably drawn at the synaptic level. The paper would require substantial revision. Here are the details of why this part is concerning:

- The culture system is not sufficient to study presynaptic architecture. The cultured neurons used in this study develop presynaptic-like structures instructed by laminin present on the dish. By definition, these are not presynapses, as they lack postsynaptic partners. It is very unclear how this culture system reflects the true structure and the physiology of neuromuscular junctions. Any finding using this system must be corroborated in real synapses.
- Non-physiological stimulation is used (figure 3). Stimulating axons by adding Roscovitine for five minutes is very far from physiological. Keeping calcium channels open for five minutes will lead to a very strong influx of calcium. This will likely lead to the activation of intracellular pathways that may not be activated during physiological stimulation. Hence, it is very unclear what the increase in Munc13-1 levels in this figure truly reflects.
- Poor data quality. The images corresponding to immunostainings are heavily processed and, in most cases, broadly saturated. Such images cannot be analyzed.
- Lack of clarity in the analysis.

o In Figure 3, the number of Munc13 puncta per growth cone is reported. How are growth cones identified, and how is their confinement defined, given that no markers are expressed?

o Similarly, in figure 4, the number of puncta per release site is reported. How are release sites defined here?

o The authors provide no comprehensive information about how puncta are detected. Were they manually counted?

o Extended data figure 4: SNAP25 is not an active zone marker. It is a protein that is expressed throughout the axons. Hence, quantifying colocalization between SNAP25 and Munc13 is not meaningful.

• The actual contribution of local translation of Munc13-1 to the phenotypes of SMA is very unclear. The authors show in Figure 2 that deletion of the 3'UTR portion of Munc13-1 mRNA results in decreased levels of this transcript in axons. However, expressing the Δ 3'UTR Munc13-1 mRNA is sufficient to restore Munc13-1 levels in axons (Extended Data Figure 2), as well as the levels of RIM, Piccolo, Bassoon (Figure 5), and Cav2.2 (Figure 6). This suggests that local translation is not essential for maintaining presynaptic protein levels in axons. Given that the in-vivo rescue experiments (Figure 8) cannot distinguish between axonal and global translation, and that the stimulation experiment (Figure 3) is not interpretable, it is not possible to assess the role of local translation of Munc13-1.

• Overall, the most likely explanation and the most accurate conclusion based on the data presented in this paper is that axonal degeneration is slowed when Munc13 is overexpressed (regardless of whether translation occurs in axons or soma). This is consistent with the observation that the immunoreactivity of several axonal proteins (RIM, Bass, Munc13, Cav2.2) is restored by Munc13 mRNA (including the Δ 3'UTR), and with the fact that Munc13 acts downstream of RIM (PMID 21262469). A way to address the concerns of the structural and physiological data would be to focus on the angle of axonal degeneration rescued by Munc13-1 expression and significantly de-emphasize those structural and functional experiments

(figures 3 through 6).

- Frequent conceptual errors. The wrong terminology is often used. The authors should carefully review all statements in their manuscript to ensure they are both scientifically and conceptually accurate. I will provide some examples, though there are likely many more.
 - o Line 215. Supramolecular assemblies cannot be identified using imaging. The authors should refer to puncta or clusters.
 - o Similarly, the authors often call Munc13 clusters release sites. While there is a relationship between those two, it is unknown whether every Munc13 cluster is a release site. They should refer to them as puncta.
 - o Line 288-290: "At presynaptic AZs, a complex consisting of Munc13-1, RIM and RIM-binding protein (RIM-BP) regulates calcium coupling by tethering SVs near voltage-gated Ca²⁺ channels." This statement is not up to date. Clustering of calcium channels and priming via Munc13 is instructed by different sub-machineries (PMID 39160372). Update text accordingly.
 - o Line 292: coupling primed vesicles to a calcium source is the basis for submillisecond neurotransmitter release under basal conditions, not necessarily critical for plasticity.
 - o Caption of figure 5: "Munc13-1 overexpression restores neurotransmission and AZ assembly in cultured Smn KO motoneurons". Neurotransmission is not measured here, only levels of proteins.
- The calcium imaging experiment (Figure 6) is not interpretable for two reasons. First, it is unclear what the calcium transients represent. Neuronal networks are not formed in their cultures, as the axons are separated from the soma and dendritic tree. Hence, these transients do not reflect synaptic function. Most likely, the changes measured here are simply indicative of axonal degeneration. Second, the measurements are not ratiometric, making quantifying intensities unreliable (Figures 6f-g).

Other points to consider include:

- Line 124. A better description of the mouse model is needed. Non-experts will not know the genetics behind this model
- Figure 1B. Why is there no cell body and axon fraction for the controls? Comparing axonal fractions and cell body fractions in the mutant versus whole cell levels in the control is a misleading comparison.
- Quality controls for the FISH experiment are necessary. Given that they have a Munc13-1 KO mouse, the authors should validate their probe using this knockout mouse.
- The total levels of Munc13-1, measured by western blot, are similar between control and mutant neurons. If the axonal levels of this protein are truly decreased in the mutant, this should be reflected in a western blot. The explanation given by the authors (Lines 142-144) is that this is dominated by the levels in the soma. This is probably not right given that Munc13-1 is a presynaptic protein. The authors should investigate this further.

Reviewer #3

(Remarks to the Author)

The Survival Motoneurone (SMN) protein is ubiquitously expressed in healthy subjects and reduced throughout the organism in the autosomal recessive disease, spinal muscular atrophy (SMA). Synaptic and, in particular, neuromuscular dysfunction in this paediatric condition therefore remains to be fully explained. In this report, Moradi and colleagues address this conundrum by identifying and implicating a synaptic protein, Munc13-1, in the pathophysiology of SMA. The authors show that whilst total motoneuronal levels of Munc13-1 are not significantly altered, axonal localization and local translation of the gene's transcripts are markedly reduced under conditions of reduced SMN. This affects axonal Munc13-1 protein levels and, since Munc13-1 is critically important in the process of neurotransmission, synaptic activity in SMA is diminished. Identifying and implicating synaptic proteins as mediators of the SMA phenotype is important and would cast substantially new light on neuromuscular dysfunction in this disease. To this extent, the article by Moradi et al is interesting and expansive. Yet, the data in support of the authors' conclusions is sometimes counterintuitive and equivocal. Specific concerns and detailed comments follow:

Lines 87-88: The assertion that "Munc13-1 arrests spontaneous...at NMJs" is not entirely accurate. The study (Ref. 40) clearly found an increase in spontaneous quantal content release in Munc13 knockout NMJs. Please edit accordingly.

Figure 1b: This analysis has been incorrectly carried out. If mRNA levels are assessed separately in axons and soma of mutant cells, transcripts should similarly be assessed in the two compartments of motoneurons of control mice. Instead, it appears that transcripts from control neuronal compartments are combined and then compared to axonal and soma-localized Munc13-1 of mutant cells. This is erroneous and ought to be remedied.

Extended Figure 1a: The assertion made based on data provided here is inconsistent with the data in Fig. 1b wherein it is claimed that Munc13-1 transcripts are INCREASED in mutant soma. On a related note, I am not sure if one should be surprised that there is less Munc13-1 staining in mutant growth cones. This might merely be a consequence of the smaller mutant growth cone. Were Munc13-1 RNA levels normalized to tubulin RNA levels in the respective growth cones?

Figures 1j, k: Decreased signal here might also be explained by the smaller mutant growth cones. Is the local translation of transcripts universally reduced in mutant growth cones or is it more specific to Munc13-1? Quantifying a "control" transcript would be instructive and is recommended.

Extended Figure 2a: The data presented here is not convincing or representative of the assertion being made; control IP has more RPL8 than does mutant IP.

Lines 168-170: This is most confusing. The authors claim that certain transcripts are not reduced in SMA axons, citing the table (Table 1) from a previously conducted study. They then go on to state that transport of these transcripts, including Syp is not SMN-dependent. Whilst true that Syp does not decrease in mutant axons, on the other hand, the transcript INCREASES, suggesting that SMN inhibits its transport and that knockdown of the protein (SMN) releases such inhibition. This section needs to be completely re-worded for the sake of clarity. Moreover, why was the 3'UTR of Syp and not the 3'UTR of any other similarly regulated gene transcript used for the study? The authors never explain the specific choice of the Syp 5'UTR.

Figure 2b: Comparisons should be made between the same compartments in untransduced and transduced cells. If what is shown in the graphs for untransduced cells is data from the entire cell, the comparison is erroneous.

Extended Figures 2d-g: Why do the authors "expect" Munc13-1 protein in knockout axons transduced with the RESCΔ3'UTR construct to be normalized if Munc13-1 RNA in the axons of the cells is not restored and, moreover, is subject to local translation? Would the paucity of axonally localized Munc13-1 RNA not result in reduced Munc13-1 protein?

Figures 3a, b: Why does treatment with Ros fail to increase Munc13-1 signal in knockout growth cones? Whilst true that they have less Munc13-1 transcript, the residual transcript should nevertheless be subject to modest increased translation following Ros treatment. The same argument could be made for mutant neurones transduced with the RESCΔ3'UTR construct.

Extended Figure 3f: The overlap referred to is, in reality, quite non-specific; Munc13-1 coverage is much more extensive than is the mCLING dye signal. Please alter the sentence accordingly.

Figure 3d: Please analyze this data statistically to confirm the significance of the increase in release sites.

Line 256: Figure 3d does not depict the information referenced in this sentence. Please correct.

Extended Figure 3d: Please present statistics to support the stated conclusion.

Lines 273-306: This section is simply not intuitive. If transport and local translation of Munc13-1 is essential for synaptic activity and if the RESCΔ3'UTR construct fails to restore these, it is unclear how the construct restores pre-synaptic active zone architecture, i.e., CaV2.2, RIM, Bassoon etc., as well as MEPP frequency but not MEPP amplitude or EPPs. Reductions in MEPP amplitude in SMA neurons would suggest aberrant loading of neurotransmitter into synaptic vesicles, a function that is not attributed to Munc13-1 or implicated in this study.

Figure 5b: It is unclear why the RESCΔ3'UTR construct is deemed to partially correct the phenotype in the SV recycling assay. Indeed, the result is inconsistent with the failure of this construct to restore levels of axonal Munc13-1 (e.g., see Fig. 2b). I am also puzzled by the assertion that the RESCΔ3'UTR construct provides partial benefit, as this is not depicted in Fig. 5b. I am also quite confused by the allegation that the RESC construct restores neurotransmission – based on assay outcomes – and the subsequent statement in lines 280-282. The two sentences seem to contradict one another, as the inference made in the second sentence does not logically follow from observations made in the first.

Figure 6h: Please provide statistical analyses.

Lines 346-347: The authors' results do not support this conclusion. They restricted their observations to a single time point (P10) at which both denervated NMJs and reduced motoneurons were observed. To conclusively assert what is being expressed in the sentence, mutants would have to be analyzed at multiple time points. Else, please alter the sentence.
Lines 356-357: The assertion that no overt phenotype was seen in "rescued" mice is difficult to reconcile with a mere 40% increase in survival and 100% mortality of all such mutants by P26. What were the weights of the 4 cohorts of mice? What did the "rescued" mutants succumb to? What did muscle morphology in the mutants look like? Was neurotransmission at NMJs of mice truly restored? In this regard, parallel analysis of a transgenic line expressing a RESCΔ3'UTR construct would have been very useful.

Extended Figure 6e: Please include photomicrographs of controls and untransduced mutant cells.

Lines 468-471: What purpose would intra-axonal translation of Munc13-1 be expected to serve? Can the authors comment?

Version 1:

Reviewer comments:

Reviewer #1

(Remarks to the Author)

The authors have adequately addressed my concerns and I have no further issues with the manuscript.

Reviewer #2

(Remarks to the Author)

Moradi and colleagues have made clear improvements to the previous version of the manuscript. The clarity of the text has increased, and conclusions are now more accurately framed. The main message remains that increasing the axonal levels of Munc13 in a model of spinal muscular atrophy improves motor coordination and increases the survival of mutant mice, which is a sufficiently significant finding.

In some parts of the manuscript, the authors keep attempting to establish a broader connection between neuronal activity, active zone protein levels, local translation and active zone assembly that continues to be insufficiently supported by the data. I appreciate the effort to address these complex mechanisms, but this is still based on assumptions and overinterpretation. I believe these claims need to be interpreted more factually. Here are the three main concern in this regard:

- Reliance on incubation with Roscovitine for 5 minutes as their means to establish changes driven by neuronal activity. As pointed out, this approach does not constitute physiological activation and therefore does not support conclusions about activity dependence. The revised version now includes evidence that other forms of non-physiological excitation (such as application of KCl for 5 minutes) also result in increased protein levels in growth cones, which is not sufficient. I appreciate the effort to validate their pharmacological approach, and I understand that it may be hard to bypass the confounding of the different levels of Cav2 across genotypes. However, not being able to present a way to address neuronal stimulation physiologically does not allow taking conclusions at this level. Furthermore, if Roscovitine operates through opening Cav2 channels -as the authors state in line 215- it is unclear how this pharmacological treatment would bypass the reduced Cav2 levels in Smn null motoneurons. The authors should replace every claim regarding activity-dependence for "pharmacological treatment with Roscovitine for 5 minutes" in the abstract, introduction and results sections. Any interpretation about how this pharmacological treatment may reflect a plasticity pathway triggered by neuronal activity should only be present in the discussion.
- Suggestion that Munc13 mediates assembly of release sites. They should better incorporate studies that concluded that Munc13 is downstream of active zone assembly (for instance PMIDs 16052212, 21262469, 31530643, or 27537483) and discuss how those studies impact their findings. Similarly, remove any explicit claims that Munc13 drives assembly in motoneurons (for instance, line 82).
- Interpretation of calcium imaging experiments. I appreciate the inclusion of controls, such as calcium channel blockers, in the revised version, which addresses some of my previous concerns. However, the data remain insufficient to support conclusions about active zone assembly or Cav2 clustering. This is because the observed effects could stem from upstream factors affecting presynaptic excitability rather than local protein assembly. For instance, a reduced ability of the neurons to fire action potentials or decreased synaptic input from the network could also lead to diminished calcium signals, independent of changes at the active zone. The authors should limit their conclusion to a 'reduction in calcium influx' and clearly state that the underlying cause—whether upstream or local— cannot be determined from these experiments.

Also, in the revised version, the authors added several assumptions regarding the amplitude of Cav2 signals and its connection to action potentials (lines 342-351). This seems very speculative as it is not backed by literature or any measurement shown. This should be removed.

Furthermore, three additional issues that remain are:

- The representative microscopy images remain below standards: This revised version still includes many examples that are saturated. See Figures 3c, 5c, or Extended Data Figure 5, as examples, although there may be more. Given that the correlation between intensities are measured, non-saturated images should be shown as examples.

Furthermore, some experiments lack representative examples (for instance, measurements of Cav2 in Extended Data Figure 5). All measurement should be accompanied by example images.

- Conceptual inaccuracies remain:
 - o Line 331: Munc13 does not cluster Cav2
 - o Line 23: I would revisit the statement that loss of Munc13-1 arrest spontaneous release. I would say decrease. Arrest may sound like full stop
 - o Line 272: Munc13-1 clusters are not found on dendrites, but on axons that make synapses onto dendrites. Important distinction
 - o Furthermore, the statements regarding image resolution (lines 271 and 283) seems to be based on the assumption that they are getting close to the best resolution that the microscopes can provide. Unless they have conducted precise measurements of the XY resolution that they can get in their samples, such statements should be removed.
- Potential bias in their structural analysis. Since the ring-like structures are selected based on the arrangement of Munc13 and Munc13 clusters are counted manually, it is essential that the authors clearly explain how they minimized bias. Ideally, such measurements should be performed using a marker different from the protein of interest and analyzed using automated methods to eliminate bias.

Version 2:

Reviewer comments:

Reviewer #2

(Remarks to the Author)

No further comments.

RESPONSE TO REVIEWER COMMENTS

Reviewer #1 (Remarks to the Author):

The paper by Moradi et al describes a role for localization and translation of Munc13-1 mRNA in spinal motor atrophy. The authors mostly use a mouse model for SMA and study neurons in vitro or in vivo. They note a reduction in the localization of Munc-13 mRNA in the mouse model – which they attribute to the loss of the RNA-binding protein HnRNPR which binds to the 3'UTR of Munc-13 mRNA and localizes it to axon terminals (previous work from the Sendtner lab). They then carry out a series of rescue experiments- in which the localization of Munc-13 is restored by expression a version with a different 3'UTR. They show, with a variety of methods, that this restores Munc-13 levels and restores compromised presynaptic function and also some of the behavioral phenotypes in the mutant mice. Overall, this is a very nice study- my specific comments and suggestions are listed below.

1. Re: the qRT-PCR analysis of RNA from the axonal compartment of the SMN KO, what about other transcripts? It would be nice to see a few that are not affected- and it is also desirable to see the real (unnormalized) expression level of the Munc13 mRNA in the control mice.

Response: We thank the reviewer for this helpful suggestion. To address this point, we selected several transcripts that were not significantly altered in Smn-depleted motoneurons as determined by our RNA-Seq data (Supplementary Table 1), including Vamp2, Snap25, Synaptophysin, and Synaptotagmin. We then assessed their expression levels by qRT-PCR in axonal RNA from Smn-knockdown compartmentalized cultures. Consistent with our RNA-Seq data, we did not observe significant alterations in the axonal levels of these transcripts upon Smn knockdown (see Extended Data Fig. 2h,i). These transcripts serve as internal negative controls and support the specificity of the reduction observed for Munc13-1 mRNAs.

Regarding the request for the unnormalized expression levels of Munc13-1 mRNA in control motoneurons, we have included these data (absolute Ct-values) exclusively in this response letter for reference only.

Munc13-1		Gapdh	
Somatodendritic compartment	28.45 25.22 24.35 25.78 25.2 25.19	Somatodendritic compartment	21.88 17.4 17.75 18.36 21.18 16
Axonal compartment	32.43 32.13 24.29 30.82 25.15 31.71	Axonal compartment	30.1 31.72 25.46 25.56 27.73 31.1

Table 1: unnormalized expression levels of Munc13-1 and Gapdh mRNAs in control mouse spinal motoneurons.

2. minor point: the authors nicely state at the beginning of the paper that they will refer to the SMN-Hung transgenic mouse as KO- but then they do not do this. It is very challenging for

readers and reviewers to digest the very small differences between +/- and -/- superscripted in all of the figures.

Response: To improve consistency throughout the manuscript, we have revised the text, figure panels, and figure legends accordingly. As stated in the beginning of the result part, the term "KO" is now used consistently to refer to the Smn-Hung transgenic and superscripted genotypes have been minimized to avoid confusion.

3. I find it surprising that the Munc-13 protein content is unaltered in the SMN cultured motoneurons and also find the assertion that the “somata represent the main part of these neurons”... The volume of the axon is not insignificant.

Response: We thank the reviewer for this important comment. In our original experiments, we performed Western blot analysis of total Munc13-1 protein levels from cultured Smn KO motoneurons. However, we did not carry out quantitative analysis at that point, since the amount of the protein that can be extracted from such cultures is very low. To address this concern, we have now conducted additional quantifications using lysates from multiple independent SMA motoneuron cultures. These new data reveal a modest but significant ~25% reduction in total Munc13-1 protein levels in Smn KO motoneurons compared to control (see Extended Data Fig. 1i,j).

4. For the Puro-PLA experiments- is the antibody epitope for Munc13 N-terminal?

Response: Yes, we used an antibody that detects Munc13-1, specifically targeting amino acids 3-317 at the N-terminus.

What does the puro-PLA signal in the cell body look like

Response: We have performed additional experiments and quantified our previous data to assess the Puro-PLA signal in cell bodies in Smn KO cultures. No differences were detected in the somatic Puro-Munc13-1-PLA signal between Smn KO and the control (see Extended Data Fig. 1k,l).

and how far is the axonal terminal studied from the cell body?

Response: We examined the axonal growth cone, which is located at the tip of the axon. In motoneurons cultured on laminin 211/221, the average axonal length typically measures around 260 μm (see the quantification graph below).

Is the puro-PLA signal blocked by anisomycin?

Response: To address this point, we conducted additional control experiments involving anisomycin treatment and found that the Puro-PLA signal is indeed abolished in its presence (see Extended Data Fig. 1n). This confirms that the detected signal reflects active translation.

5. minor point: “dots”- the authors might want to adopt the term “punctae” or “puncta” instead.

Response: For consistency, we have replaced the term "dots" with "punctae" throughout the text, figure panels, and figures.

6. In the synaptosome experiments, are the synaptosomes free of nuclear contaminants?

Response: To address this concern, we performed additional control Western blot assays to examine the quality of the crude synaptosome fractions. In contrast to the nuclear marker Histone H3, Synapsin levels were specifically enriched in the Pellet 2 fraction, indicating that the synaptosomes were largely free of nuclear contaminants (see Extended Data Fig. 2f).

7. re: the levels of ribosomes in synaptic fractions from the SMN KO animals- the data shown in supplementary figure 2c are not convincing.

Response: We have repeated this experiment (n = 3 mice/genotype) and updated the representative blot accordingly. Additionally, we quantified the blots to ensure accuracy (see Extended Data Fig. 2a-c).

8. Re: fig 3b. The KO delta-3'UTR data- in particular in the images shown, do not seem to fit with the authors idea. It looks like there is indeed a rescue.

Response: We have updated this figure by including better and representative images accordingly (see Fig 3b).

9. Re: the anisomycin experiments in figure 3 (supplement). The statement that begins “these

data indicate that Munc13-1 local translation is required for the de novo formation of SV release sites” is not supported by these experiments. The anisomycin experiments do not specifically block synthesis of Munc-13 and there is nothing about the anisomycin application that addresses whether the translation is local.

Response: We appreciate the reviewer’s comment on the anisomycin experiments shown in original Extended Data Fig.3. We agree that anisomycin does not specifically block the synthesis of Munc13-1 and does not directly address the question of whether its translation is local. This experiment was intended to complement our data from the KO^{Δ3’UTR} model, which specifically assesses local translation of Munc13-1 in the axonal growth cones of motoneurons. In order to address this concern, we have revised the statement in the manuscript to avoid over interpretation and ensure it accurately reflects the scope of the anisomycin experiments.

10. the co-localization of Munc-13 and RPL8. This experiment does not convincingly show that ribosomes are colocalized with Munc-13. There are 79 ribosomal proteins and there is the possibility of extra-ribosomal functions. Perhaps a better experiment would be to use the y10B antibody which recognizes ribosomal RNA. The discussion section on this point is also over-stated.

Response: We agree that using RPL8 alone may not support sufficient evidence of ribosome colocalization with Munc13-1. We have therefore performed additional experiments using the Y10B antibody to detect ribosomal RNA for more specific analysis. These new data are included in the Extended Data Fig. 4a-f as well as Extended Data Fig. 5c in the revised manuscript.

11. Re: Fig 5 a,b. This partial rescue by the delta mutant is not tested statistically. Are the rescue and delta 3’UTR really different? The analysis in B does not appear to support this statement.

Response: We have now included the statistical analysis in the revised manuscript and figure panel. This new analysis shows that the uptake of the Synaptotagmin antibody (indicating recycled SVs) in Smn KO^{Δ3’UTR} motoneurons is only slightly, but significantly, lower compared to both control and Smn KO^{Rescue} conditions, and is not significantly different from Smn KO motoneurons. We have revised the manuscript to provide a more detailed explanation and interpretation of the results involving the KO^{Δ3’UTR} mutant in Fig. 5a,b (These data suggest that while restoring Munc13-1 protein levels alone may partially rescue neurotransmission defects in spinal motoneurons, a complete rescue is only achieved when the local translation of Munc13-1 is also restored).

In addition, in figure 6f-h, the results with the delta 3’UTR mutant are not fully explained or interpreted.

Response: The Ca²⁺ transients presented in Figure 6f-h reflect spontaneous events in cultured motoneurons, which we interpret as being triggered either by action potentials (typically associated with transients of higher amplitude) or by local calcium currents. High-amplitude transients usually represent Ca²⁺ entry through VGCCs that are clustered at functional release sites within the active zone. Although the frequency of Ca²⁺ transients was increased in both the Rescue and Δ3’UTR mutant conditions, the amplitude remained significantly lower in the Δ3’UTR mutant. This suggests that although some local calcium activity is restored in the Δ3’UTR mutant, action potential-dependent transients, requiring fully functional release sites with integrated VGCCs, remain impaired. This difference likely

*reflects the reduced capacity of the $\Delta 3'$ UTR construct to support the formation and maturation of functional release sites. Thus, the lack of local Munc13-1 synthesis in *Smn KO* and *Smn KO ^{$\Delta 3'$ UTR}* likely impairs the formation of VGCC-enriched active zones, leading to reduced transient amplitudes. This interpretation is further supported by our depolarization-evoked Ca^{2+} imaging experiments, which revealed reduced responses in *Smn KO* and *Smn KO^{Rescue $\Delta 3'$ UTR}* neurons (Fig. 6f–h). We have clarified this mechanistic interpretation in the revised manuscript text accordingly.*

12. Re fig 8. Can the authors provide reasons why the rescue delays death, but does not fully restore to WT lifetimes?

*Response: The partial rescue of lifespan, despite Munc13-1 restoration, is likely due to the ubiquitous expression of SMN and its loss affecting multiple tissues, including NMJs and muscles. Full restoration of lifespan would require complete recovery of SMN levels across these different organs and tissues. While SMN-based therapies improve survival, muscle weakness and fatigue remain challenging. Munc13-1 restoration can rescue NMJ function and muscle weakness but does not rescue other tissues such as the liver and heart. In future experiments, we will explore whether combining Munc13-1 restoration with SMN-targeted therapies can further improve survival. Thus, *Smn* loss predominantly impacts motoneurons, muscles, and NMJs, but it could also affect other tissues due to its widespread expression in the body. See these reviews: PMID: 23228902, and DOI: [10.2174/1566524016666161128113338](https://doi.org/10.2174/1566524016666161128113338)).*

Reviewer #2 (Remarks to the Author):

This study by Moradi and colleagues investigates the expression and levels of Munc13-1 in spinal muscular atrophy (SMA) and how modulating Munc13-1 expression may improve degenerative phenotypes in this disease. They demonstrate that the levels of Munc13-1 and other presynaptic proteins are reduced in the axons of neurons cultured from an SMA mouse model and in axons from neurons derived from patients with SMA. They also show that a portion of Munc13-1 mRNA is present in, and locally translated at, axons, and that this process is impaired in the SMA mouse model and in human neurons. Additionally, they show that the 3'UTR region of Munc13-1 mRNA is important for targeting this transcript to axons, and that replacing it with the 3'UTR of a different gene (Synaptophysin) restores the levels of this mRNA in the axons of the SMA mouse model. Finally, they demonstrate that increasing the levels of Munc13-1 decreases signs of axonal degeneration, improves motor coordination, and leads to a slight increase in the survival of the mutant mice.

Addressing the relationship between synaptic proteins/functions and degenerative disorders is timely. In that context, this study is somewhat relevant. However, the current manuscript is problematic, mainly because the experiments corresponding to the assembly and ultrastructure of release sites and neuronal excitability (Figures 3, 4, 5 and 6) are not well designed, executed or conceptualized. In consequence, no conclusions can be reliably drawn at the synaptic level. The paper would require substantial revision. Here are the details of why this part is concerning:

- The culture system is not sufficient to study presynaptic architecture. The cultured neurons used in this study develop presynaptic-like structures instructed by laminin present on the dish. By definition, these are not presynapses, as they lack postsynaptic partners. It is very unclear how this culture system reflects the true structure and the physiology of

neuromuscular junctions. Any finding using this system must be corroborated in real synapses.

Response: We appreciate the reviewer's point regarding the potential limitations of in vitro culture systems for studying presynaptic architecture. While it is true that cultured motoneurons form presynaptic-like structures on laminin2-coated substrates, that lack postsynaptic partners, previous studies have shown that specific laminin isoforms (merosin/laminin- α 2) play a critical instructive role in organizing presynaptic components, including the clustering of VGCCs, in a manner that closely mimics early stages of neuromuscular synapse formation, see references 119-121 in the revised manuscript and PMID: 25556799, DOI: [10.1038/nature03112](https://doi.org/10.1038/nature03112), DOI: [10.1038/35097557](https://doi.org/10.1038/35097557).

To address this concern more directly and ensure the physiological relevance of our findings, we have now performed additional super-resolution imaging of neuromuscular junctions in vivo using Airyscan microscopy (achieving an x-y resolution of ~120 nm). These new experiments enabled us to corroborate the presynaptic phenotypes observed in cultured motoneurons with those present at bona fide synapses. The results, presented in Figure 4g, demonstrate that key structural and molecular features identified in our cultures are also observed at NMJs, supporting the validity of our in vitro system for investigating aspects of presynaptic organization. We have also added the corresponding discussion to the revised manuscript.

Moreover, we attempted to apply Expansion Microscopy to muscle preparations for super-resolution imaging of NMJs. However, the resulting image quality was suboptimal, as the protocol requires extensive optimization for thick tissue samples. Given the time-consuming nature of this process and the scope of the current revision, we have not pursued this approach further at this stage.

- Non-physiological stimulation is used (figure 3). Stimulating axons by adding Roscovitine for five minutes is very far from physiological. Keeping calcium channels open for five minutes will lead to a very strong influx of calcium. This will likely lead to the activation of intracellular pathways that may not be activated during physiological stimulation. Hence, it is very unclear what the increase in Munc13-1 levels in this figure truly reflects.

*Response: In previous studies (PMID: 34668554, PMID: 35650592), we employed 1-minute BDNF stimulation to investigate the local translation of proteins within axonal growth cones. In this study, initial experiments demonstrated that BDNF stimulation results in increased Munc13-1 protein levels in growth cones of wt motoneurons, which is indicative of local translation (verified by using anisomycin and nocodazole controls). However, *Smn* KO and KO^{Rescue} motoneurons exhibited no response to BDNF stimulation (see the graphs below). This lack of response is attributed to impaired TrkB receptor signaling, which was shown in two of our recent papers (PMID: 35650592, PMID: 36607273). Consequently, BDNF triggers reduced activation of downstream pathways in *Smn* KO motoneurons, which likely obscures detection of Munc13-1 local translation (see graphs below).*

*To overcome this issue, we used Roscovitine to induce local translation of Munc13-1, based on previous findings demonstrating Roscovitine's rescue effect on Ca²⁺ transient amplitude in the axonal growth cones of *Smn* KO motoneurons (PMID: 31981925).*

Given the reduced number and impaired clustering of VGCCs in axonal growth cones (our study and PMID: 17923533) and NMJs (PMID: 20089893) of *Smn* KO motoneurons, we decided not to use depolarizing agents such as KCl for stimulation, but a specific drug that opens VGCCs in a comparable manner in wt and *Smn* KO motoneurons, as it appears essential for this experiment. Therefore, using depolarizing agents for stimulation would prevent discrimination between two possible mechanisms: reduced Munc13-1 mRNA levels or reduced $Ca_v2.2$ channel activity.

Along the same line, our Ca^{2+} imaging data revealed a reduced response to KCl stimulation in *Smn* KO motoneurons, see Fig. 6f and 6g. This indicates that the physiological excitability of *Smn* KO axonal growth cones is impaired, making any physiological stimulation that should result in similar intracellular responses in control and *Smn* KO motoneurons difficult.

To address the reviewer's concern and to improve the physiological relevance, we performed additional experiments including the BDNF and KCl stimulation using wt motoneurons. These new experiments complement our Roscovitine data and provide additional support for our conclusions: please see Extended Data Fig. 3b-e and Extended Data Fig. 4a-f.

- Poor data quality. The images corresponding to immunostainings are heavily processed and, in most cases, broadly saturated. Such images cannot be analyzed.

Response: We used 16-bit **unprocessed raw** images for Pearson colocalization analysis or signal intensity measurements. The processed images were used for the representation only. Furthermore, to improve overall image quality and clarity, some of the blurred images have been replaced and all relevant channels were included in the corresponding representative images (see Extended Data Fig. 5a).

- Lack of clarity in the analysis.

Response: We have included a detailed description of our analysis methods in the revised manuscript.

o In Figure 3, the number of Munc13 puncta per growth cone is reported. How are growth cones identified,

Response: Growth cones in our analysis were identified based on their characteristic morphology, including their position at the distal tip of axons, their size, and the presence of distinct filopodia-like structures. Additionally, growth cones were identified as regions devoid of DAPI staining, ensuring that they were not part of the neuronal soma or nuclei. Although no

specific growth cone markers were used in this experiment, we ensured consistency and accuracy by applying well-established morphological criteria. To provide further clarity, we included a detailed description of our identification process in the Method section of the revised manuscript.

and how is their confinement defined, given that no markers are expressed?

Response: For these experiments, we used low-density cultures in which the neurons were largely isolated, thereby minimizing overlap between neighboring processes. Using well-established morphological criteria, we ensured that only puncta located within clearly defined growth cones were included in our analysis, while signals from adjacent axonal or dendritic regions were excluded. To further clarify this approach, we have included an example overview image in this response letter, where three growth cones can be clearly identified based on these criteria (see the example below).

o Similarly, in figure 4, the number of puncta per release site is reported. How are release sites defined here?

Response: Munc13-1 ring-like clusters (originally defined as release sites and now renamed to Munc13-1 clusters in the revised manuscript) in expanded specimens were identified based on two main structural criteria: (I) their size (diameter range between 170-1250 nm) and (II) their characteristic ring-like structure. These criteria were validated using our lattice-SIM data, as shown in Fig. 3c, d, (also see the example picture above) and Extended Data Fig. 3f-j. To address the reviewer's concern, we have included this detailed description in the Methods section of the revised manuscript to ensure clarity and reproducibility.

o The authors provide no comprehensive information about how puncta are detected. Were they manually counted?

Response: The puncta were manually counted within the identified Munc13-1 ring-like clusters (see Fig. 4a, the green squares within the overview images and the corresponding zoom-in images on the right (100 nm bar), in which individual spots can clearly be distinguished within identified Munc13-1 clusters.

o Extended data figure 4: SNAP25 is not an active zone marker. It is a protein that is expressed throughout the axons. Hence, quantifying colocalization between SNAP25 and Munc13 is not meaningful.

Response: We thank the reviewer for pointing to this potential source of confusion. In response, we have repeated these experiments using a more appropriate active zone marker: $Ca_v2.2$, which is known to form clusters specifically at active zones. These updated results are now presented in Extended Data Fig. 4a-f and Extended Data Fig. 5e, and confirm the colocalization between Munc13-1 and $Ca_v2.2$.

• The actual contribution of local translation of Munc13-1 to the phenotypes of SMA is very unclear. The authors show in Figure 2 that deletion of the 3'UTR portion of Munc13-1 mRNA results in decreased levels of this transcript in axons. However, expressing the $\Delta 3'$ UTR Munc13-1 mRNA is sufficient to restore Munc13-1 levels in axons (Extended Data Figure 2), as well as the levels of RIM, Piccolo, Bassoon (Figure 5), and $Ca_v2.2$ (Figure 6). This suggests that local translation is not essential for maintaining presynaptic protein levels in axons. Given that the in-vivo rescue experiments (Figure 8) cannot distinguish between axonal and global translation, and that the stimulation experiment (Figure 3) is not interpretable, it is not possible to assess the role of local translation of Munc13-1.

*Response: Our qRT-PCR data reveal a ~75% reduction in Munc13-1 mRNA levels in axons of *Smn* KO motoneurons, while Munc13-1 protein levels are reduced by approximately 50%. This observation suggests that the residual Munc13-1 protein in *Smn* KO axons may arise from two sources: local translation of the remaining axonal mRNA, and transport of somatically synthesized protein. In the Δ Rescue condition, overexpression of Munc13-1 via a lentivirus construct likely enhances the protein transport from the soma. This leads to the restoration of its protein levels in axonal growth cones in *Smn* KO motoneurons, which results in increased levels of other active zone proteins such as RIM, Bassoon, and $Ca_v2.2$, as shown in Figures 5 and 6. However, our functional assays, including calcium imaging and Synaptotagmin antibody uptake, demonstrate that despite these increased protein levels, presynaptic function is not fully restored. These findings suggest that increasing levels of AZ components alone is insufficient to restore synaptic functions. Rather, we propose that local translation of Munc13-1 is required for the proper assembly and clustering of these AZ proteins, including VGCCs, into functional release sites. While overexpressed, soma-derived Munc13-1 can be transported to axons and facilitate the recruitment of other AZ proteins, it does not restore their precise nanoscale organization or function. This highlights the critical*

role of local translation in establishing functional presynaptic architecture, which cannot be compensated by enhanced somatic protein synthesis and its axonal transport.

Additionally, this overexpression might engage compensatory signaling pathways that could enhance the translation efficiency of the remaining 25% of axonal mRNA. However, these mechanistic possibilities remain to be further investigated.

- Overall, the most likely explanation and the most accurate conclusion based on the data presented in this paper is that axonal degeneration is slowed when Munc13 is overexpressed (regardless of whether translation occurs in axons or soma). This is consistent with the observation that the immunoreactivity of several axonal proteins (RIM, Bass, Munc13, Cav2.2) is restored by Munc13 mRNA (including the $\Delta 3'$ UTR), and with the fact that Munc13 acts downstream of RIM (PMID 21262469). A way to address the concerns of the structural and physiological data would be to focus on the angle of axonal degeneration rescued by Munc13-1 expression and significantly de-emphasize those structural and functional experiments (figures 3 through 6).

Response: As discussed above, In the $\Delta 3'$ Rescue condition, overexpression of Munc13-1 likely results in enhanced protein transport from the soma to the axonal growth cones, leading to its restored protein levels in Smn KO motoneurons. This restoration also leads to increased levels of other AZ proteins such as RIM, Bassoon, and Cav2.2, as shown in Figures 5 and 6. However, our functional assays demonstrate that despite these increased protein levels, presynaptic function is not fully rescued. These findings suggest that increasing levels of AZ components alone is insufficient to restore synaptic functions. Instead, we provide evidence that local translation of Munc13-1 is required for proper assembly and clustering of active zone proteins, including VGCCs, into functional release sites. While soma-derived Munc13-1 can be transported to axons when overexpressed, and facilitate the recruitment of other AZ proteins, it does not restore their precise nanoscale organization or function. This highlights the critical role of local translation in establishing functional presynaptic architecture, a process which cannot be compensated by enhanced somatic protein synthesis and its axonal transport.

- Frequent conceptual errors. The wrong terminology is often used. The authors should carefully review all statements in their manuscript to ensure they are both scientifically and conceptually accurate. I will provide some examples, though there are likely many more.
 - o Line 215. Supramolecular assemblies cannot be identified using imaging. The authors should refer to puncta or clusters.

Response: We have revised the terminology throughout the manuscript, replacing "supramolecular assemblies" with "Munc13-1 clusters".

- o Similarly, the authors often call Munc13 clusters release sites. While there is a relationship between those two, it is unknown whether every Munc13 cluster is a release site. They should refer to them as puncta.

Response: We agree with the reviewer that not every Munc13-1 cluster can be definitively defined as a functional release site. To address this, we have revised the manuscript to refer to these structures as Munc13-1 clusters where appropriate. Moreover, our mCLING uptake assay (see revised Extended Data Fig. 3j), which revealed endocytic activity in close proximity

to *Munc13-1* clusters, supports that these clusters are functionally relevant in vesicle recycling, which appears as an important parameter of the synaptic activity.

o Line 288-290: “At presynaptic AZs, a complex consisting of *Munc13-1*, RIM and RIM-binding protein (RIM-BP) regulates calcium coupling by tethering SVs near voltage-gated Ca^{2+} channels.” This statement is not up to date. Clustering of calcium channels and priming via *Munc13* is instructed by different sub-machineries (PMID 39160372).

Response: We have incorporated the recommended reference and updated the manuscript text accordingly to reflect these new findings.

o Line 292: coupling primed vesicles to a calcium source is the basis for submillisecond neurotransmitter release under basal conditions, not necessarily critical for plasticity.

Response: We have corrected this part in our revised manuscript.

o Caption of figure 5: “*Munc13-1* overexpression restores neurotransmission and AZ assembly in cultured *Smn* KO motoneurons”. Neurotransmission is not measured here, only levels of proteins.

Response: We performed a Synaptotagmin antibody uptake assay (see Fig. 5. a,b), which indicates reduced endocytosis of SVs, which is an indirect indication of their release. However, for transparency, we changed the corresponding capture legend to “Munc13-1 overexpression increases SV release events and AZ assembly in cultured Smn KO motoneurons”.

• The calcium imaging experiment (Figure 6) is not interpretable for two reasons. First, it is unclear what the calcium transients represent. Neuronal networks are not formed in their cultures, as the axons are separated from the soma and dendritic tree. Hence, these transients do not reflect synaptic function. Most likely, the changes measured here are simply indicative of axonal degeneration. Second, the measurements are not ratiometric, making quantifying intensities unreliable (Figures 6f-g).

Response: We have performed control experiments with CTX, a blocker of voltage-gated Ca^{2+} channels. No spontaneous activity is measurable anymore (see Extended data Fig. 6b), providing evidence for specificity and an argument against axonal degeneration as the source of these transients. We respectfully disagree with the assertion that neuronal networks are not formed in our cultures. These neurons are capable of forming functional networks through various types of synaptic contacts, including axon-axon, dendrite-axon, soma-axon, soma-dendrite, and soma-soma interactions. More importantly, these cultured neurons show spontaneous activity, which is critical for neuronal maturation and survival, even in the absence of postsynaptic muscle targets (see PMID: 31981925, 36607273, 29163025). The observed Ca^{2+} transients are consistent with ongoing neuronal activity and are not indicative of axonal degeneration, as demonstrated in the referenced studies.

We acknowledge the limitation that Oregon Green™ 488 BAPTA-1 is not ratiometric, and absolute intensity measurements may be influenced by factors such as dye loading variability, photobleaching, or uneven distribution. However, this dye is widely used for single-wavelength calcium imaging due to its high sensitivity to Ca^{2+} and rapid response time, which appeared essential for our experiments, despite being non-ratiometric.

To address these concerns, we have implemented rigorous controls to minimize variability, including standardized dye loading protocols, $\Delta F/F_0$ normalization, and careful imaging practices to avoid photobleaching.

While absolute quantification may be less reliable compared to ratiometric dyes,

Response: Normalization of the fluorescence signal to baseline ($\Delta F/F_0$) is a reasonable and widely used compensation for differences in absolute dye loading or fluorescence intensities, ensuring that the observed changes reflect relative fluctuations in Ca^{2+} levels with high reliability.

Other points to consider include:

- Line 124. A better description of the mouse model is needed. Non-experts will not know the genetics behind this model

Response: We have now included a more detailed description of this mouse model in the Materials and Methods section of our revised manuscript.

- Figure 1B. Why is there no cell body and axon fraction for the controls? Comparing axonal fractions and cell body fractions in the mutant versus whole cell levels in the control is a misleading comparison.

Response: We thank the reviewer for highlighting this potential misunderstanding. We confirm that the mRNA quantification was performed separately for axonal and somatic compartments in both control and mutant motoneurons. However, we acknowledge that the original presentation may have caused confusion by suggesting that control compartment data were pooled. To address this, we have revised the corresponding graphs and adapted the data presentation in the revised manuscript to clearly distinguish compartment-specific analyses for both control and mutant conditions.

- Quality controls for the FISH experiment are necessary. Given that they have a Munc13-1 KO mouse, the authors should validate their probe using this knockout mouse.

Response: We have performed additional control experiments using Munc13-1 knockdown motoneurons to validate the specificity of the FISH probe. These experiments confirmed that the signal is specifically reduced upon knockdown, supporting the probe's specificity (see Extended Data Fig. 1c,d).

- The total levels of Munc13-1, measured by western blot, are similar between control and mutant neurons. If the axonal levels of this protein are truly decreased in the mutant, this should be reflected in a western blot. The explanation given by the authors (Lines 142-144) is that this is dominated by the levels in the soma. This is probably not right given that Munc13-1 is a presynaptic protein. The authors should investigate this further.

Response: In our original experiments, we performed Western blot analysis of total Munc13-1 protein levels in cultured Smn KO motoneurons, but we had not carried out quantitative analysis at that point. To address this concern, we have now conducted additional quantifications using lysates from multiple independent SMA motoneuron cultures. These new data revealed a significant reduction of approximately 25% in total Munc13-1 protein levels

in Smn KO motoneurons compared to controls (see Extended Data Fig. 1i,j).

Reviewer #3 (Remarks to the Author):

The Survival Motoneurone (SMN) protein is ubiquitously expressed in healthy subjects and reduced throughout the organism in the autosomal recessive disease, spinal muscular atrophy (SMA). Synaptic and, in particular, neuromuscular dysfunction in this paediatric condition therefore remains to be fully explained. In this report, Moradi and colleagues address this conundrum by identifying and implicating a synaptic protein, Munc13-1, in the pathophysiology of SMA. The authors show that whilst total motoneuronal levels of Munc13-1 are not significantly altered, axonal localization and local translation of the gene's transcripts are markedly reduced under conditions of reduced SMN. This affects axonal Munc13-1 protein levels and, since Munc13-1 is critically important in the process of neurotransmission, synaptic activity in SMA is diminished. Identifying and implicating synaptic proteins as mediators of the SMA phenotype is important and would cast substantially new light on neuromuscular dysfunction in this disease. To this extent, the article by Moradi et al is interesting and expansive. Yet, the data in support of the authors' conclusions is sometimes counterintuitive and equivocal. Specific concerns and detailed comments follow:

Lines 87-88: The assertion that "Munc13-1 arrests spontaneous...at NMJs" is not entirely accurate. The study (Ref. 40) clearly found an increase in spontaneous quantal content release in Munc13 knockout NMJs. Please edit accordingly.

Response: We thank the reviewer for this correction and have revised the sentence accordingly in the manuscript.

Figure 1b: This analysis has been incorrectly carried out. If mRNA levels are assessed separately in axons and soma of mutant cells, transcripts should similarly be assessed in the two compartments of motoneurons of control mice. Instead, it appears that transcripts from control neuronal compartments are combined and then compared to axonal and somalocalized Munc13-1 of mutant cells. This is erroneous and ought to be remedied.

Response: We thank the reviewer for highlighting this potential misunderstanding. We confirm that the mRNA quantification was performed separately for axonal and somatic compartments in both control and Smn KO motoneurons. However, we acknowledge that the original presentation may have caused confusion by suggesting that control compartment data were pooled. To address this, we have revised the corresponding graphs and adapted the data presentation to clearly distinguish compartment-specific analyses for both control and mutant conditions in the revised manuscript.

Extended Figure 1a: The assertion made based on data provided here is inconsistent with the data in Fig. 1b wherein it is claimed that Munc13-1 transcripts are INCREASED in mutant soma.

Response: We acknowledge the discrepancy between the data shown in original Extended Data Fig. 1a and original Fig. 1b. We suspect that the initially observed increase in Munc13-1 mRNA levels may be due to the limited sample size in the original qRT-PCR experiment (n = 3 independent cultures). As compartmentalized cultures require a large number of motoneurons (500,000 per chamber), we initially had to pool motoneurons obtained from multiple Smn KO

embryos. This limited the number of independent experiments. To resolve this, we repeated this experiment to obtain data from 5 independent cultures. The updated analysis revealed no significant increase in Munc13-1 mRNA levels in the soma of *Smn* KO motoneurons (see revised Fig. 1b).

On a related note, I am not sure if one should be surprised that there is less Munc13-1 staining in mutant growth cones. This might merely be a consequence of the smaller mutant growth cone. Were Munc13-1 RNA levels normalized to tubulin RNA levels in the respective growth cones?

*Response: While it is true that smaller mutant growth cones could contribute to reduced Munc13-1 staining, we have used average intensity measurements for Munc13-1 signal that are normalized to the size of the growth cone. This method ensures that the intensity measurements are independent of the growth cone size, allowing for a more accurate comparison between the KO and control conditions. Regarding the normalization of Munc13-1 mRNA levels, we normalized the Munc13-1 mRNA (dots) to the axon length or growth cone size. As suggested by the reviewer, normalization to another mRNA, such as tubulin, could be an option. However, we did not decide to normalize to tubulin mRNA, since we do not know whether tubulin mRNA levels are altered in *Smn* KO axons, and this could introduce additional variability that might complicate the interpretation of the data.*

Figures 1j, k: Decreased signal here might also be explained by the smaller mutant growth cones.

Response: To address this, we repeated these experiments and also reanalyzed our data and normalized the number of Munc13-1 PLA-puncta to μm^2 growth cone size. This was done to eliminate any potential bias introduced by size differences between KO and control growth cones (see revised Fig. 1j,k).

Is the local translation of transcripts universally reduced in mutant growth cones or is it more specific to Munc13-1? Quantifying a “control” transcript would be instructive and is recommended.

*Response: We appreciate this valuable suggestion. To address this issue, we performed Puro-PLA for a control transcript, VAMP2, in *Smn* KO motoneurons. We selected VAMP2 because (i) its mRNA levels are not affected by *Smn* depletion, as demonstrated by our qRT-PCR data (see revised Extended Data Fig. 2h,i) and RNA-Seq analysis (Supplementary Table 1), and (ii) a knockout-validated rabbit antibody against its N-terminus was available. Consistent with these findings, the Puro-PLA signal for VAMP2 was comparable between control and *Smn* KO motoneurons (see Extended Data Fig. 2j,k).*

Extended Figure 2a: The data presented here is not convincing or representative of the assertion being made; control IP has more RPL8 than does mutant IP.

Response: We have now repeated this experiment (n = 3 mice/genotype) and updated the representative blot accordingly. Additionally, we quantified the blots to ensure accuracy (see Extended Data Fig. 2a-c).

Lines 168-170: This is most confusing. The authors claim that certain transcripts are not reduced in SMA axons, citing the table (Table 1) from a previously conducted study. They then go on to state that transport of these transcripts, including Syp is not SMN-dependent. Whilst true that Syp does not decrease in mutant axons, on the other hand, the transcript

INCREASES, suggesting that SMN inhibits its transport and that knockdown of the protein (SMN) releases such inhibition. This section needs to be completely re-worded for the sake of clarity.

Response: We agree with this comment. The observed effect could result from the removal of Smn's inhibition on the transport of this transcript or from a compensatory mechanism. Since we cannot distinguish between these two possibilities, we have revised the text to reflect both explanations and discuss them accordingly.

Moreover, why was the 3'UTR of Syp and not the 3'UTR of any other similarly regulated gene transcript used for the study? The authors never explain the specific choice of the Syp 5'UTR.

Response: We appreciate the opportunity to clarify why we chose the Syp 3'UTR for this study. Initially, our RNA-Seq data revealed that while the transcripts of some mRNAs were downregulated in the axonal compartment after Smn knockdown, the transcripts of certain synaptic vesicle proteins (including Syp) were upregulated. This observation suggested that, first, these mRNAs might be Smn-independent, and second, there could be a compensatory mechanism, which enhances the transport of certain mRNAs, such as Syp, into axons in Smn KO motoneurons. We selected the Syp 3'UTR based on its potential involvement in this compensatory mechanism, which could help ensure the enhanced transport of Munc13-1 mRNA into axons in the absence of Smn.

Figure 2b: Comparisons should be made between the same compartments in untransduced and transduced cells. If what is shown in the graphs for untransduced cells is data from the entire cell, the comparison is erroneous.

Response: We thank the reviewer for highlighting this potential misunderstanding. We confirm that the mRNA quantification was performed separately for axonal and somatic compartments in both control and KO motoneurons. However, we acknowledge that the original presentation may have caused confusion by suggesting that control compartment data were pooled. To address this, we have revised the corresponding graphs and adapted the data presentation to clearly distinguish compartment-specific analyses for both control and mutant conditions in the revised manuscript.

Extended Figures 2d-g: Why do the authors “expect” Munc13-1 protein in knockout axons transduced with the RESC Δ 3'UTR construct to be normalized if Munc13-1 RNA in the axons of the cells is not restored and, moreover, is subject to local translation? Would the paucity of axonally localized Munc13-1 RNA not result in reduced Munc13-1 protein?

Response: Our qRT-PCR data reveal a ~75% reduction in Munc13-1 mRNA levels in axons of Smn KO motoneurons, while Munc13-1 protein levels are reduced by approximately 50%. This observation suggests that the remaining 50% of Munc13-1 protein in KO axons may result from local translation of the residual mRNA, in addition to the somatically synthesized protein transported into the axon. In the Δ Rescue condition, overexpression of Munc13-1 via a lentivirus construct may enhance the protein transport from the soma, thereby increasing its levels in the axonal growth cones. Additionally, this overexpression might engage compensatory signaling pathways that enhance the translation efficiency of the remaining 25% of axonal mRNA. However, these mechanistic possibilities remain to be further investigated.

Figures 3a, b: Why does treatment with Ros fail to increase Munc13-1 signal in knockout growth cones? Whilst true that they have less Munc13-1 transcript, the residual transcript should nevertheless be subject to modest increased translation following Ros treatment.

Response: Our data show that only ~25% of Munc13-1 mRNA remains in axons in Smn KO motoneurons. Given this substantial reduction, it is possible that the 5-minute stimulation window we used was insufficient to detect a measurable increase in Munc13-1 local translation in both Smn KO and Rescue Δ 3'UTR. One potential approach to address this limitation would be to extend the stimulation time (e.g., to 20 minutes) using Roscovitine. However, as reviewer 2 correctly noted, prolonged stimulation with Roscovitine, by increasing VGCC activity, could lead to excessive calcium influx and activate calcium-dependent signaling pathways that may be detrimental to the cells. In light of this concern, we have decided not to pursue this control experiment with an unphysiologically long opening of VGCCs.

The same argument could be made for mutant neurones transduced with the RESC Δ 3'UTR construct.

Response: Please see above.

Extended Figure 3f: The overlap referred to is, in reality, quite non-specific; Munc13-1 coverage is much more extensive than is the mCLING dye signal. Please alter the sentence accordingly.

Response: We respectfully disagree with this interpretation. The apparent difference in the spatial extent of the signals is due to the distinct staining patterns of the two markers: mCLING displays a punctate pattern, consistent with endocytosed vesicles, while Munc13-1 exhibits a stronger and broader signal forming ring-like structures that likely represent protein clusters at the AZ. Despite these differences in labeling characteristics, the two signals are in close spatial proximity, supporting the conclusion that Munc13-1 puncta are functionally associated with vesicle recycling sites. We have clarified this explanation in the revised manuscript text.

Figure 3d: Please analyze this data statistically to confirm the significance of the increase in release sites.

Response: The statistical analysis is included in the revised Extended Data Fig. 3f.

Line 256: Figure 3d does not depict the information referenced in this sentence. Please correct.

Response: We believe our conclusion is correct, as the quantification and statistical analysis presented in the original Extended Data Fig. 3b, and again in the revised Extended Data Fig. 3f, support this conclusion.

Extended Figure 3d: Please present statistics to support the stated conclusion.

Response: The statistical information is now added in the corresponding figure panel (see Extended Data Fig. 3h, right panel) in the revised manuscript to support the stated conclusion.

Lines 273-306: This section is simply not intuitive. If transport and local translation of Munc13-1 is essential for synaptic activity and if the RESCΔ3'UTR construct fails to restore these, it is unclear how the construct restores pre-synaptic active zone architecture, i.e., Ca_v2.2, RIM, Bassoon etc., as well as MEPP frequency but not MEPP amplitude or EPPs. Reductions in MEPP amplitude in SMA neurons would suggest aberrant loading of neurotransmitter into synaptic vesicles, a function that is not attributed to Munc13-1 or implicated in this study.

Response: We thank the reviewer for this thoughtful comment. Based on our data with Smn KO motoneurons, we suggest that approximately 60% of Munc13-1 protein is synthesized locally within axonal growth cones, while the remaining 40% is produced in the soma and transported into the presynaptic terminals. We hypothesize that the somatically produced Munc13-1 is sufficient to maintain basic synaptic activity, while the locally translated Munc13-1 plays a crucial role in processes like synaptic plasticity, which we have attempted to demonstrate using super-resolution microscopy and stimulation paradigms. This phenomenon appears to be well correlated with the phenotype exhibited by SMA patients, who show deficits in synaptic plasticity despite having basic neurotransmission intact. These patients exhibit motor dysfunction and fatigue, which further underscores the importance of local protein synthesis in synaptic dynamics. In the ΔRescue condition, overexpression of Munc13-1 via a lentivirus likely results in enhanced protein transport from the soma, leading to the restoration of its protein levels in axonal growth cones in Smn KO motoneurons. This restoration also leads to increased levels of other AZ proteins such as RIM, Bassoon, and Ca_v2.2, as shown in Figures 5 and 6. However, our functional assays, including calcium imaging and Synaptotagmin antibody uptake, demonstrate that despite these increased protein levels, presynaptic function is not fully restored. These findings suggest that increasing levels of AZ components alone is insufficient to restore synaptic functions. Rather, we propose that local translation of Munc13-1 is required for proper assembly and clustering of active zone proteins, including VGCCs, into functional release sites. While overexpressed, soma-derived Munc13-1 can be transported to axons and facilitate the recruitment of other AZ proteins, it does not restore their precise nanoscale organization or function. This highlights the critical role of local translation in establishing functional presynaptic architecture, which cannot be compensated by enhanced somatic protein synthesis and its axonal transport.

Additionally, this overexpression might engage compensatory signaling pathways that could enhance the translation efficiency of the remaining 25% of axonal mRNA. However, these mechanistic possibilities remain to be further investigated.

To address the reviewer's concern, we would like to emphasize that in our study, we did not measure MEP amplitude directly, but instead employed Ca²⁺ imaging to monitor Ca²⁺ transients, which indirectly reflect the function and clustering of VGCCs.

Figure 5b: It is unclear why the RESCΔ3'UTR construct is deemed to partially correct the phenotype in the SV recycling assay. Indeed, the result is inconsistent with the failure of this construct to restore levels of axonal Munc13-1 (e.g., see Fig. 2b). I am also puzzled by the assertion that the RESCΔ3'UTR construct provides partial benefit, as this is not depicted in Fig. 5b. I am also quite confused by the allegation that the RESC construct restores neurotransmission – based on assay outcomes – and the subsequent statement in lines 280-282. The two sentences seem to contradict one another, as the inference made in the second sentence does not logically follow from observations made in the first.

Response: While the RESCΔ3'UTR construct fails to restore Munc13-1 mRNA levels and hence its local translation, it results in increased levels of Munc13-1 protein in axonal growth cones, via enhanced protein transport from the soma. This enhancement contributes to partial

rescue due to the restoration of Munc13-1 total protein levels in axonal growth cones.

Figure 6h: Please provide statistical analyses.

Response: We thank the reviewer for this comment. However, it is not feasible to perform standard statistical analysis on the type of data presented in Fig. 6h, as it represents a binary outcome (failure vs. no failure) plotted across individual events, rather than a continuous variable. This format is commonly used to illustrate response variability across trials, and applying conventional statistical tests to such distributions would not be meaningful.

Lines 346-347: The authors' results do not support this conclusion. They restricted their observations to a single time point (P10) at which both denervated NMJs and reduced motoneurons were observed. To conclusively assert what is being expressed in the sentence, mutants would have to be analyzed at multiple time points. Else, please alter the sentence.

Response: We have deleted this sentence accordingly.

Lines 356-357: The assertion that no overt phenotype was seen in "rescued" mice is difficult to reconcile with a mere 40% increase in survival and 100% mortality of all such mutants by P26.

Response: This sentence refers to the comparison between control and control^{Rescue} (WT^{Rescue}) mice, but not KO and KO^{Rescue} mice.

What were the weights of the 4 cohorts of mice?

Response: We initially monitored the body weight of the 4 cohorts up to P14; however, due to the heterogeneity within litter groups, we did not observe a significant improvement in the weight of Smn KO^{Rescue} mice.

What did the "rescued" mutants succumb to?

Response: Regarding the "rescued" mutants, while the exact cause of their death is unknown, we hypothesize that it is likely due to respiratory dysfunction or heart failure. Both conditions are common complications in models of spinal muscular atrophy. Respiratory dysfunction, in particular, is a hallmark of severe SMA phenotypes, often stemming from weakness in respiratory muscles.

What did muscle morphology in the mutants look like?

Response: We did not assess muscle morphology in the mutants as part of this study. Our focus was primarily on neural mechanisms and synaptic functionality. Future studies could explore muscle morphology to provide a more comprehensive understanding of how motoneuron dysfunction translates to muscular phenotypes.

Was neurotransmission at NMJs of mice truly restored? In this regard, parallel analysis of a transgenic line expressing a RESCA3'UTR construct would have been very useful.

Response: We thank the reviewer for this suggestion. However, developing a knock-in mouse expressing the RESCA3'UTR construct would require generating and validating the model, followed by crossing with SMA mice for at least six generations. While this approach is valuable, it is beyond the scope of our current study. Electrophysiological analyses were not conducted in the present study, as they are difficult and hard to interpret at early postnatal

stages, when the period of removal of supernumerary NMJs that cause polysynaptic innervation in skeletal muscles are not removed yet. This would require additional controls and analyses that are beyond the scope of our original aims and remain outside the scope of this current revision.

Extended Figure 6e: Please include photomicrographs of controls and untransduced mutant cells.

Response: These photomicrographs are included in the Fig. 9C.

Lines 468-471: What purpose would intra-axonal translation of Munc13-1 be expected to serve? Can the authors comment?

Response: We thank the reviewer for pointing out the potential misunderstanding in terminology. To clarify, our use of the term "intra-axonal translation" specifically refers to local protein synthesis within axonal growth cones, which are specialized structures that include presynaptic terminals. However, since we detect Munc13-1 mRNAs in multiple axonal sub-compartments, including both proximal axons and growth cones, using FISH and PLA, we have revised the terminology throughout the manuscript from "intra-axonal" to "local translation" for consistency and clarity.

Reviewer #1 (Remarks to the Author):

The authors have adequately addressed my concerns and I have no further issues with the manuscript.

Reviewer #2 (Remarks to the Author):

Moradi and colleagues have made clear improvements to the previous version of the manuscript. The clarity of the text has increased, and conclusions are now more accurately framed. The main message remains that increasing the axonal levels of Munc13 in a model of spinal muscular atrophy improves motor coordination and increases the survival of mutant mice, which is a sufficiently significant finding.

In some parts of the manuscript, the authors keep attempting to establish a broader connection between neuronal activity, active zone protein levels, local translation and active zone assembly that continues to be insufficiently supported by the data. I appreciate the effort to address these complex mechanisms, but this is still based on assumptions and overinterpretation. I believe these claims need to be interpreted more factually. Here are the three main concern in this regard:

- Reliance on incubation with Roscovitine for 5 minutes as their means to establish changes driven by neuronal activity. As pointed out, this approach does not constitute physiological activation and therefore does not support conclusions about activity dependence. The revised version now includes evidence that other forms of non-physiological excitation (such as application of KCl for 5 minutes) also result in increased protein levels in growth cones, which is not sufficient. I appreciate the effort to validate their pharmacological approach, and I understand that it may be hard to bypass the confounding of the different levels of Cav2 across genotypes. However, not being able to present a way to address neuronal stimulation physiologically does not allow taking conclusions at this level. Furthermore, if Roscovitine operates through opening Cav2 channels -as the authors state in line 215- it is unclear how this pharmacological treatment would bypass the reduced Cav2 levels in Smn null motoneurons. The authors should replace every claim regarding activity-dependence for “pharmacological treatment with Roscovitine for 5 minutes” in the abstract, introduction and results sections. Any interpretation about how this pharmacological treatment may reflect a plasticity pathway triggered by neuronal activity should only be present in the discussion.

In order to deal with the specific criticism of the reviewer, we have now replaced “activity-dependent” with pharmacological stimulation or stimulus-dependent throughout the text.

We respectfully would like to argue against the reviewer’s view that the incubation with Roscovitine for 5 minutes is not an adequate means for studying changes driven by enhanced neuronal activity. It has to be kept in mind that continuous muscle contraction, which is highly relevant for the proximal tonic muscle groups that are severely affected in spinal muscular atrophy, requires high-frequency continuous firing of motoneurons at about 50 Hz, leading to repeated opening of presynaptic VGCCs that causes a continuous influx of Ca²⁺ into the presynapse. This is undoubtedly induced by treatment with Roscovitine, a drug that

prolongs the opening of VGCCs. The pathophysiological relevance of this pharmacological mechanism has been demonstrated by several independent lines of evidence from multiple groups besides ours, working on the role of VGCCs for diseases of neuromuscular transmission.

*The best example is Lambert-Eaton Syndrome, a disease caused by an autoimmune-mediated reduction in the number of VGCCs at motor nerve terminals. This autoimmune-mediated reduction in presynaptic VGCCs causes a decrease in calcium influx in response to presynaptic action potentials, which decreases chemical neurotransmission, leading to severe neuromuscular weakness that resembles spinal muscular atrophy in many pathophysiological aspects. It needs to be pointed out that repeated stimulation of motor nerves, which one would consider the most physiological way of inducing neuromuscular transmission, is without relevance for the treatment of this disease. Instead, the therapy with K⁺-channel inhibitors such as 3,4 diaminopyridine (3,4-DAP) (Verschuuren et al., *Expert Opin Pharmacother.* 2006 DOI: [10.1517/14656566.7.10.1323](https://doi.org/10.1517/14656566.7.10.1323)) is effective. Pharmacological inhibition of potassium channels in the presynaptic nerve terminal broadens the action potential and increases the duration of membrane depolarization. This prolonged period of depolarization activates a greater number of presynaptic VGCCs and increases calcium influx into the nerve terminal, as shown in a clinical trial (Wirtz et al., *Clin Pharmacol Ther.* 2009 DOI: [10.1038/clpt.2009.35](https://doi.org/10.1038/clpt.2009.35)). This just underlines the pathophysiological relevance of prolonged and continuous opening of VGCCs for therapy of neuromuscular diseases that involve dysfunction of presynaptic acetylcholine release.*

*Pharmacological use of Roscovitine and Roscovitine derivatives is then generally considered as the next logical step for therapy development. Indeed, Roscovitine and Roscovitine derivatives such as MF-06 and GV-58 have been introduced in therapy development for Lambert-Eaton Syndrome and other neuromuscular diseases, and the corresponding studies have shown high potential of using Roscovitine and its derivatives for restoring neuromuscular transmission (Wu et al., *Neuropharmacology.* 2018, DOI: [10.1016/j.neuropharm.2017.12.022](https://doi.org/10.1016/j.neuropharm.2017.12.022), Tarr et al., *J Neurosci.* 2013, DOI: [10.1523/JNEUROSCI.4629-12.2013](https://doi.org/10.1523/JNEUROSCI.4629-12.2013))*

The context of our study is therapy development for spinal muscular atrophy, and for this reason, we have followed this pharmacological means of Cav2 channel opening, which appears highly relevant in our context.

- Suggestion that Munc13 mediates assembly of release sites. They should better incorporate studies that concluded that Munc13 is downstream of active zone assembly (for instance PMIDs 16052212, 21262469, 31530643, or 27537483) and discuss how those studies impact their findings. Similarly, remove any explicit claims that Munc13 drives assembly in motoneurons (for instance, line 82).

We have changed the text accordingly in line 82 and incorporated these references in the introduction and discussion. We do not claim that Munc13-1 mediates the assembly of the AZ, but rather show that locally translated Munc13-1 is recruited into newly assembled Munc13-1 clusters, which may represent SV release sites.

- Interpretation of calcium imaging experiments. I appreciate the inclusion of controls, such as calcium channel blockers, in the revised version, which addresses some of my previous concerns. However, the data remain insufficient to support conclusions about active zone assembly or Cav2 clustering. This is because the observed effects could stem from upstream factors affecting presynaptic excitability rather than local protein assembly. For instance, a reduced ability of the neurons to fire action potentials or decreased synaptic input from the network could also lead to diminished calcium signals, independent of changes at the active zone. The authors should limit their conclusion to a 'reduction in calcium influx' and clearly state that the underlying cause—whether upstream or local— cannot be determined from these experiments.

We have changed our conclusion and the text accordingly to address these possibilities outlined by the reviewer.

In order to dissect the relevance of network effects, altered upstream firing of action potential in motoneurons and altered Ca²⁺ influx through VGCCs, the efficacy of Roscovitine in rescuing the disease phenotype in a spinal atrophy mouse model has previously been described (Tejero R et al., iScience. 2020 DOI: 10.1016/j.isci.2020.100826.)

This previous study shows that pharmacologically induced prolonged opening of Ca²⁺ channels improves neuromuscular transmission and has beneficial therapeutic effects in SMA. We agree with the reviewer that these previously observed effects of altered Ca²⁺ influx might be independent of changes at the active zone and that our present study cannot give a final answer on whether additional pharmacological actions at other sites than the presynaptic sites contribute to the observed effects.

- Also, in the revised version, the authors added several assumptions regarding the amplitude of Cv2 signals and its connection to action potentials (lines 342-351). This seems very speculative as it is not backed by literature or any measurement shown. This should be removed

The complex relationship between amplitude of calcium signals and action potentials has been discussed in detail in previous studies, for example by Catterall WA, (Cold Spring Harb Perspect Biol. 2011 DOI: 10.1101/cshperspect.a003947) and Dolphin AC and Lee A (Nat Rev Neurosci. 2020, DOI: 10.1038/s41583-020-0278-2). These articles did not address the potential relevance and pathophysiological changes in the context of spinal muscular atrophy. Since the relation between action potentials and the amplitude of Calcium signals is not the theme of our study, we have now removed these sentences in line 342-351.

Furthermore, three additional issues that remain are:

- The representative microscopy images remain below standards: This revised version still includes many examples that are saturated. See Figures 3c, 5c, or Extended Data Figure 5, as examples, although there may be more. Given that the correlation between intensities are measured, non-saturated images should be shown as examples.

We have now replaced these images with non-saturated representative images (please see revised Fig. 3c, Fig. 5c, and Extended Data Fig. 5).

Furthermore, some experiments lack representative examples (for instance, measurements of Cav2 in Extended Data Figure 5). All measurement should be accompanied by example images.

We have now added representative images. Please see Extended Data Fig. 6.

- Conceptual inaccuracies remain:
 - o Line 331: Munc13 does not cluster Cav2

We have now corrected this sentence.

- o Line 23: I would revisit the statement that loss of Munc13-1 arrest spontaneous release. I would say decrease. Arrest may sound like full stop

We have now changed this phrase accordingly.

- o Line 272: Munc13-1 clusters are not found on dendrites, but on axons that make synapses onto dendrites. Important distinction

We have now changed this phrase accordingly.

- o Furthermore, the statements regarding image resolution (lines 271 and 283) seems to be based on the assumption that they are getting close to the best resolution that the microscopes can provide. Unless they have conducted precise measurements of the XY resolution that they can get in their samples, such statements should be removed.

We have now removed this phrase accordingly.

- Potential bias in their structural analysis. Since the ring-like structures are selected based on the arrangement of Munc13 and Munc13 clusters are counted manually, it is essential that the authors clearly explain how they minimized bias. Ideally, such measurements should be performed using a marker different from the protein of interest and analyzed using automated methods to eliminate bias.

We have now added the following section in Materials and Methods to explain how we minimized bias.

“To minimize potential bias in the identification and quantification of Munc13-1 ring-like structures in expansion specimens, we employed several strategies. First, ROIs were defined using objective structural criteria such as size, intensity, and their characteristic ring-like structure rather than subjective selection. These criteria were validated using our lattice-SIM data, as shown in Fig. 3c,d, and Extended Data Fig. 3f-j. The number of Munc13-1 nanoassemblies within each ring-like cluster was determined manually, with an average of 6-8 nanoassemblies per cluster. The center-to-center distance between neighboring nanoassemblies was measured using the freehand line tool in Fiji. Although ring-like arrangements were identified based on the Munc13-1 signal, we used a consistent

thresholding approach across all images, applied uniformly using Fiji. Two independent observers performed manual counting to ensure reproducibility. We acknowledge the limitation that Munc13-1 was both the marker of interest and the basis for structural selection.”